# $\sigma$-**PCA: a building block for neural learning of identifiable linear transformations**

**Fahdi Kanavati**        *fkanavati@invertibleai.com*
*Invertible AI, London, U.K.*

**Lucy Katsnith**        *lkatsnith@invertibleai.com*
*Invertible AI, London, U.K.*

**Masayuki Tsuneki**        *mtsuneki@invertibleai.com*
*Invertible AI, London, U.K.*

**Reviewed on OpenReview:** *https: // openreview. net/ forum? id=KpVJ6CGnwI*

## Abstract

Linear principal component analysis (PCA) learns (semi-)orthogonal transformations by orienting the axes to maximize variance. Consequently, it can only identify orthogonal axes whose variances are clearly distinct, but it cannot identify the subsets of axes whose variances are roughly equal. It cannot eliminate the subspace rotational indeterminacy: it fails to disentangle components with equal variances (eigenvalues), resulting, in each eigen subspace, in randomly rotated axes. In this paper, we propose $\sigma$-PCA, a method that (1) formulates a unified model for linear and nonlinear PCA, the latter being a special case of linear independent component analysis (ICA), and (2) introduces a missing piece into nonlinear PCA that allows it to eliminate, from the canonical linear PCA solution, the subspace rotational indeterminacy – without whitening the inputs. Whitening, a preprocessing step which converts the inputs into unit-variance inputs, has generally been a prerequisite step for linear ICA methods, which meant that conventional nonlinear PCA could not necessarily preserve the orthogonality of the overall transformation, could not directly reduce dimensionality, and could not intrinsically order by variances. We offer insights on the relationship between linear PCA, nonlinear PCA, and linear ICA – three methods with autoencoder formulations for learning special linear transformations from data, transformations that are (semi-)orthogonal for PCA, and arbitrary unit-variance for ICA. As part of our formulation, nonlinear PCA can be seen as a method that maximizes both variance and statistical independence, lying in the middle between linear PCA and linear ICA, serving as a building block for learning linear transformations that are identifiable.

## 1 Introduction

Principal component analysis (PCA) (Pearson, 1901; Hotelling, 1933) needs no introduction. It is classical, ubiquitous, perennial. It is an unsupervised learning method that can be arrived at from three paradigms of representation learning (Bengio et al., 2013): neural, manifold, and probabilistic. This paper is about learning **linear transformations** from data using the first paradigm: the **neural**, in the form of a single-layer autoencoder – a model which encodes the data into a new representation and then decodes that representation back into a reconstruction of the original data.

From the data, PCA learns a linear **(semi-)orthogonal** transformation $\mathbf{W}$ that transforms a given data point $\mathbf{x}$ into a new representation $\mathbf{y} = \mathbf{xW}$, a representation with reduced dimensionality and minimal loss of information (Diamantaras & Kung, 1996; Jolliffe & Cadima, 2016). This representation has components that are uncorrelated, i.e. have no linear associations, or, in other words, the covariance between two distinct

components is zero. When the variances of the components are clearly distinct, PCA has a unique solution, up to sign indeterminacies, that consists in a set of principal eigenvectors (or axes) ordered by their variances. This solution can usually be obtained from the singular value decomposition (SVD) of the data matrix (or the eigendecomposition of the data covariance matrix). Such a solution not only maximizes the amount of variance but also minimizes the error of reconstruction. And it is the latter – in the form of the squared difference between the data and its reconstruction – that is also the loss function of a single-layer linear autoencoder. However, minimizing such a loss will result in a solution that lies in the PCA **subspace** (Baldi & Hornik, 1989; Bourlard & Kamp, 1988; Oja, 1992b), i.e. a linear combination of the principal eigenvectors, rather than the actual principal eigenvectors. And so, a linear autoencoder performs PCA – but not in its expected axis-aligned form.

What happens when we insert a nonlinearity in the single-layer autoencoder? By natural extension, it must be a form of nonlinear PCA (Xu, 1993; Karhunen & Joutsensalo, 1994; 1995; Karhunen et al., 1998). However, to yield an interesting output, it was observed that the input data needs to be **whitened**, i.e. rotated and scaled into uncorrelated components of **unit** variance. And after the input is whitened, applying nonlinear PCA results in a similar output (Karhunen & Joutsensalo, 1994; 1995; Karhunen et al., 1998; Hyvärinen & Oja, 2000) to linear independent component analysis (ICA) (Jutten & Herault, 1991; Comon, 1994; Bell & Sejnowski, 1997; Hyvarinen, 1999). [1]

Linear ICA, too with a long history in unsupervised learning (Hyvärinen & Oja, 2000), yields, as the name implies, not just uncorrelated but independent components. Linear ICA seeks to find a linear transformation $\mathbf{B}$ – not necessarily orthogonal – such that $\mathbf{y} = \mathbf{xB}$ has its components as statistically independent as possible. It is based on the idea that $\mathbf{x}$ is an observed linear mixing of the hidden sources $\mathbf{y}$, and it aims to recover these sources solely based on the assumptions that they are non-Gaussian and independent. Generally, unlike PCA, we cannot determine the variances of the components nor their order (Hyvarinen, 1999; Hyvärinen & Oja, 2000; Hyvärinen, 2015); they are impossible to recover when we have no prior knowledge about how the data was generated. For if $\mathbf{B}$ is a transformation that recovers the sources, then so is $\mathbf{B\Lambda}$, with $\mathbf{\Lambda}$ an arbitrary diagonal scaling matrix. And so, the best that can be done, without any priors, is to assume the sources have **unit** variance. Linear ICA guarantees **identifiability** when its assumptions are met (Comon, 1994), meaning that it can recover the true sources up to trivial sign, scale, and permutation indeterminacies.

We thus have three methods for learning linear transformations from data: linear PCA can learn rotations that orient axes into directions that maximize variance, but, by solely relying on variance, it cannot eliminate the **subspace rotational indeterminacy**: it cannot separate components that have the same variance, i.e. it cannot identify the true principal eigenvectors in a given eigenspace, resulting in randomly rotated eigenvectors in the eigenspaces. Nonlinear PCA (Karhunen & Joutsensalo, 1994) can learn rotations that orient axes into directions that maximize statistical independence, reducing the subspace indeterminacy from rotational to a trivial permutational, but it cannot be applied directly to data without preprocessing – whitening. Linear ICA learns a linear transformation – not just a rotation – that points the axes into directions that maximize statistical independence under the assumption of unit variance. The relationship between all three can be best understood by the SVD of the linear ICA transformation into a sequence of rotation, scale, rotation – stated formally, $\mathbf{W\Sigma}^{-1}\mathbf{V}$, with $\mathbf{W}$ (semi-)orthogonal, $\mathbf{V}$ orthogonal, and $\mathbf{\Sigma}$ diagonal. In one formulation of linear ICA, linear PCA learns the first rotation, $\mathbf{W}$, and nonlinear PCA learns the second, $\mathbf{V}$. The scale, $\mathbf{\Sigma}^{-1}$, is simply the inverse of the standard deviations.

The problem is that, in contrast to linear PCA, conventional nonlinear PCA cannot be used directly on the data to learn the first rotation, the first being special as it can reduce dimensionality and order by variances. The process of first applying the linear PCA rotation and then dividing by the standard deviations is the whitening preprocessing step, the prerequisite for nonlinear PCA. And so, nonlinear PCA, rather than being on an equal footing, is dependent on linear PCA. Conventional nonlinear PCA, by the mere introduction of a nonlinear function, loses the ability for dimensionality reduction and ordering by variances – a jarring disparity between both linear and nonlinear PCA.

---

[1]In the literature, for ICA, the terms linear and nonlinear refer to the transformation, while, for PCA, they refer to the function. Both linear ICA and nonlinear PCA use nonlinear functions, but their resulting transformations are still linear.

As part of our contributions in this paper, we have identified the reason why conventional nonlinear PCA has been unable to recover the first rotation, $\mathbf{W}$: it has been missing, in the reconstruction loss, $\boldsymbol{\Sigma}$. This means that the nonlinear PCA model should not just be $\mathbf{W}$ but $\mathbf{W}\boldsymbol{\Sigma}^{-1}$. The reason why $\boldsymbol{\Sigma}$ is needed is simply because it standardizes the components to unit variance before applying the nonlinearity – while still allowing us to compute the variances. A consequence of introducing $\boldsymbol{\Sigma}$ is that nonlinear PCA can now learn not just the second, but also the first rotation, i.e. it can also be applied directly to the data without whitening. Another key observation for nonlinear PCA to work for dimensionality reduction is that it should put an emphasis not on the decoder contribution, but on the encoder contribution – in contrast to conventional linear and nonlinear PCA. In light of the introduction of $\boldsymbol{\Sigma}$, and to distinguish our proposed formulation from the conventional PCA model, the conventional being simply a special case with $\boldsymbol{\Sigma} = \mathbf{I}$, we call our formulation $\sigma$-PCA. [2]

Thus, our primary contribution is a reformulated nonlinear PCA method, which naturally derives from our $\sigma$-PCA formulation, that can eliminate, from the canonical linear PCA solution, the subspace rotational indeterminacy – without whitening the inputs. And so, like linear PCA, nonlinear PCA can learn a semi-orthogonal transformation that reduces dimensionality and orders by variances, but, unlike linear PCA, it can also eliminate the subspace rotational indeterminacy – it can disentangle components with equal variances. With its ability to learn both the first and second rotations of the SVD of the linear ICA transformation, nonlinear PCA serves as a building block for learning linear transformations that are identifiable.

## 2 Preliminaries

### 2.1 Learning linear transformations from data and the notion of identifiability

Given a set of observations, the goal is to learn a linear transformation that uncovers the true independent sources that generated the observations. It is as if the observations are the result of the sources becoming linearly **entangled** (Bengio et al., 2013) and what we want to do is **disentangle** the observations and **identify** the sources (Comon, 1994; Hyvärinen & Oja, 2000). [3]

Let $\mathbf{Y}_* \in \mathbb{R}^{n \times k}$ be the data matrix of the true independent sources, with $\mathbf{y}_* \in \mathbb{R}^{1 \times k}$ a row vector representing a source sample, and $\mathbf{X} \in \mathbb{R}^{n \times p}$ be the data matrix of the observations, with $\mathbf{x} \in \mathbb{R}^{1 \times p}$ a row vector representing an observation sample. As the observations can have a higher dimension than the sources, we have $p \geq k$. Without loss of generality, we assume the sources are centred. The observations were generated by transforming the sources with a true transformation $\mathbf{B}_{*(L)}^{-1} \in \mathbb{R}^{k \times p}$, the left inverse of $\mathbf{B}_*$, to obtain the observations $\mathbf{X} = \mathbf{Y}_* \mathbf{B}_{*(L)}^{-1}$.

To learn from data is to ask the following: without knowing what the true transformation was, to what extent can we recover the sources $\mathbf{Y}_*$ from the observations $\mathbf{X}$? Or in other words, to what extent can we recover the left invertible transformation $\mathbf{B}_*$? The problem is then to identify $\mathbf{B}$ such that

$$\mathbf{y} = \mathbf{x}\mathbf{B}, \tag{2.1}$$

with $\mathbf{y}$ as close as possible to $\mathbf{y}_*$. In the ideal case, we want $\mathbf{B} = \mathbf{B}_*$, allowing us to recover the true source as $\mathbf{y} = \mathbf{x}\mathbf{B}_* = \mathbf{y}_* \mathbf{B}_{*(L)}^{-1} \mathbf{B}_* = \mathbf{y}_*$. We can note that

$$\mathbf{x}\mathbf{B}_* \mathbf{B}_{*(L)}^{-1} = \mathbf{y}_* \mathbf{B}_{*(L)}^{-1} \mathbf{B}_* \mathbf{B}_{*(L)}^{-1} = \mathbf{y}_* \mathbf{B}_{*(L)}^{-1} = \mathbf{x}, \tag{2.2}$$

implying that $\mathbf{B}_*$ satisfies an **autoencoder** formulation, with $\mathbf{B}_*$ representing the encoder and $\mathbf{B}_{*(L)}^{-1}$ representing the decoder. This means that, in theory, the minimum of the reconstruction loss

---

[2]Arguably, $\sigma$-PCA should be the general PCA model, for it is common to divide by the standard deviations when performing the PCA transformation. Therefore, instead of using the names linear $\sigma$-PCA and nonlinear $\sigma$-PCA, we shall simply refer to the model with $\boldsymbol{\Sigma} = \mathbf{I}$ as the conventional PCA model.

[3]To disentangle means to recover the sources but not necessarily their distributions, while to identify means to recover not just the sources but also their distributions. Identifiability implies disentanglement, but disentanglement does not imply identifiability.

$$\mathbb{E}(||\mathbf{x} - \mathbf{x}\mathbf{B}\mathbf{B}_{(L)}^{-1}||_2^2) \tag{2.3}$$

can be reached when $\mathbf{B} = \mathbf{B}_*$, giving us a potential way to recover $\mathbf{B}_*$. Alas, $\mathbf{B}_*$ is not the only minimizer of the reconstruction loss. Equally minimizing the loss is $\mathbf{B}_*\mathbf{D}$, where $\mathbf{D} \in \mathbb{R}^{k \times k}$ is an invertible matrix with orthogonal columns (Baldi & Hornik, 1989). This is because we can write $\mathbf{B}\mathbf{B}_{(L)}^{-1} = \mathbf{B}\mathbf{D}\mathbf{D}^{-1}\mathbf{B}_{(L)}^{-1} = (\mathbf{B}\mathbf{D})(\mathbf{D}\mathbf{B}_{(L)})^{-1}$. The matrix $\mathbf{D}$ is an indeterminacy matrix, and it can take the form of a combination of certain special indeterminacies:

$$\text{Scale} \quad \mathbf{\Lambda} = \text{diag}([\lambda_1, ..., \lambda_k]) \in \mathbb{R}_+^{k \times k}, \tag{2.4}$$
$$\text{Sign} \quad \mathbf{I}_\pm = \text{diag}([\pm 1, ..., \pm 1]) \in O(k), \tag{2.5}$$
$$\text{Permutation} \quad \mathbf{P} \in O(k) \text{ a permutation matrix}, \tag{2.6}$$
$$\text{Rotation} \quad \mathbf{R} \in SO(k) \text{ an orthogonal matrix with } \det(\mathbf{R}) = 1. \tag{2.7}$$

Any orthogonal matrix $\mathbf{Q}$ can be expressed as an arbitrary product of other orthogonal matrices, and, more specifically, as the product of a rotation (determinant $+1$) and a reflection (determinant $-1$).[4] Both the sign and the permutation matrices can have determinant $+1$ or $-1$. Consequently, we can express an orthogonal indeterminacy matrix $\mathbf{Q}$ as $\mathbf{I}_\pm\mathbf{R}$, but it can equally be an arbitrary product of $\mathbf{I}_\pm$, $\mathbf{P}$, and $\mathbf{R}$, e.g.

$$\mathbf{Q} = \mathbf{I}_\pm\mathbf{R} = \mathbf{P}'\mathbf{R}'\mathbf{I}_\pm' = \mathbf{R}''\mathbf{I}_\pm'' = \mathbf{R}'''\mathbf{P}''. \tag{2.8}$$

An indeterminacy can be over the entire space or only a **subspace**, i.e. affecting only a subset of the columns. If it affects a subspace, then we shall denote it with the subscript $s$, e.g. a subspace rotational indeterminacy, which takes the form of a block orthogonal matrix, will be denoted as $\mathbf{R}_s$.

The worst of the four indeterminacies is the rotational indeterminacy. The other three are trivial in comparison. This is because a rotational indeterminacy does a linear combination of the sources, i.e. it entangles the sources, making them impossible to identify – it is the primary reason why representations are not reproducible.

A linear transformation is said to be **identifiable** if it contains **no rotational indeterminacy** – only sign, scale, and permutation indeterminacies. In other words, a transformation is identifiable if for any two transformations $\mathbf{A}$ and $\mathbf{B}$ satisfying $\mathbf{y} = \mathbf{x}\mathbf{A}$ and $\mathbf{y} = \mathbf{x}\mathbf{B}$, then $\mathbf{A} = \mathbf{B}\mathbf{\Lambda}\mathbf{P}\mathbf{I}_\pm$.

## 2.2 Singular value decomposition (SVD)

Let $\mathbf{B} \in \mathbb{R}^{p \times k}$ represent a linear transformation, then its reduced SVD (Trefethen & Bau III, 1997) is

$$\mathbf{B} = \mathbf{U}\mathbf{S}\mathbf{V}^T, \tag{2.9}$$

where $\mathbf{U} \in \mathbb{R}^{p \times k}$ a semi-orthogonal matrix with each column a left eigenvector, $\mathbf{V} \in O(k)$ an orthogonal matrix with each column a right eigenvector, and $\mathbf{S} \in \mathbb{R}^{k \times k}$ a diagonal matrix of ordered eigenvalues.

**Sign indeterminacy** If all the eigenvectors have distinct eigenvalues, then this decomposition is, up to sign indeterminacies, unique. This is because the diagonal sign matrix can commute with $\mathbf{S}$, so we can write

$$\mathbf{B} = \mathbf{U}\mathbf{S}\mathbf{V}^T = \mathbf{U}\mathbf{I}_\pm\mathbf{S}\mathbf{I}_\pm\mathbf{V}^T = \mathbf{U}'\mathbf{S}\mathbf{V}'^T. \tag{2.10}$$

---

[4]This is because $O(k) \cong SO(k) \rtimes O(1)$. This implies that an orthogonal matrix can be mapped onto a rotation and a trivial reflection, such as a diagonal matrix of all ones except $-1$ in the first row.

**Subspace rotational indeterminacy**   If multiple eigenvectors share the same eigenvalue, then, in addition to the sign indeterminacy, there is a subspace rotational indeterminacy. The eigenvectors in each eigen subspace are randomly rotated or reflected. This means that any decomposition with an arbitrary rotation or reflection of those eigenvectors would remain valid. [5] To see why, let $\mathbf{Q}_s = \mathbf{I}_\pm \mathbf{R}_s$ be a block orthogonal matrix where each block affects the subset of eigenvectors that share the same eigenvalue, then, given that it can commute with $\mathbf{S}$, i.e. $\mathbf{S}\mathbf{Q}_s = \mathbf{Q}_s\mathbf{S}$ (see Appendix I), we can write

$$\mathbf{U}\mathbf{S}\mathbf{V}^T = \mathbf{U}\mathbf{I}_\pm \mathbf{R}_s \mathbf{S}\mathbf{R}_s^T \mathbf{I}_\pm^T \mathbf{V}^T = \mathbf{U}'\mathbf{S}\mathbf{V}'^T. \tag{2.11}$$

## 2.3   Linear PCA via SVD

Linear PCA can be obtained by the SVD of the data matrix. Let $\mathbf{X} \in \mathbb{R}^{n \times p}$ be the data matrix, where each row vector $\mathbf{x}$ is a data sample, then we can decompose $\mathbf{X}$ into

$$\mathbf{X} = \mathbf{U}\mathbf{S}\mathbf{V}^T = \mathbf{U}\mathbf{R}_s \mathbf{I}_\pm \mathbf{S}\mathbf{I}_\pm^T \mathbf{R}_s^T \mathbf{V}^T. \tag{2.12}$$

Letting $\mathbf{Y} = \frac{1}{\sqrt{n}}\mathbf{U}\mathbf{R}_s\mathbf{I}_\pm$ and $\mathbf{W} = \mathbf{V}\mathbf{R}_s\mathbf{I}_\pm$ and $\mathbf{\Sigma} = \frac{1}{\sqrt{n}}\mathbf{S}$, we can write

$$\mathbf{X} = \mathbf{Y}\mathbf{\Sigma}\mathbf{W}^T, \tag{2.13}$$
$$\mathbf{Y} = \mathbf{X}\mathbf{W}\mathbf{\Sigma}^{-1}. \tag{2.14}$$

The columns of $\mathbf{W}$ are the principal eigenvectors, the columns of $\mathbf{Y}$ are the principal components, and $\mathbf{\Sigma}$ is an ordered diagonal matrix of the standard deviations.

Because of the presence of $\mathbf{R}_s$, linear PCA has a subspace rotational indeterminacy, which means that it cannot separate, cannot disentangle, components that share the same variance.

## 2.4   Degrees of identifiability via variance and independence

From the SVD, we know that any linear transformation can be decomposed into a sequence of rotation, scale, rotation. A rotation – an orthogonal transformation – is thus a building block. The first rotation, in particular, can be used for reducing dimensionality, so it can be semi-orthogonal. There are two ways for finding rotations from data: maximizing variance and maximizing independence. The former fails for components with equal variance, and the latter, although succeeding for components with equal variance, fails for components with Gaussian distributions. The latter fails because of two things: a unit-variance Gaussian has spherical symmetry and maximizing statistical independence is akin to maximizing non-Gaussianity.

Non-Gaussianity is an essential assumption of linear ICA (Comon, 1994) to guarantee identifiability. With a linear transformation, identifiability theory guarantees that independent sources with unit variance can be recovered if at least all but one of the components have non-Gaussian distributions (Comon, 1994; Hyvärinen & Oja, 2000). Figure 1 illustrates the identifiability of rotations using maximization of variance or independence. Although maximizing independence cannot identify sources with Gaussian distributions, maximizing variance can still identify the sources for one particular case: when the true transformation is orthogonal and the variances are all distinct (the ellipse in Fig. 1; see Appendix D.3).

Let us now consider how and to what degree we can identify the linear transformation based on whether it is (semi-)orthogonal or non-orthogonal. Although we will be expressing what is identified in terms of the true transformations, in reality, what we end up recovering, due to noise and absence of infinite data, are but approximations of the true transformations.

---

[5]In theory, this is the case and the rotation is random, but in practice, the eigenvalues are not exactly equal so an SVD solver could still numerically reproduce the same fixed rotational indeterminacy (Hyvärinen et al., 2009).

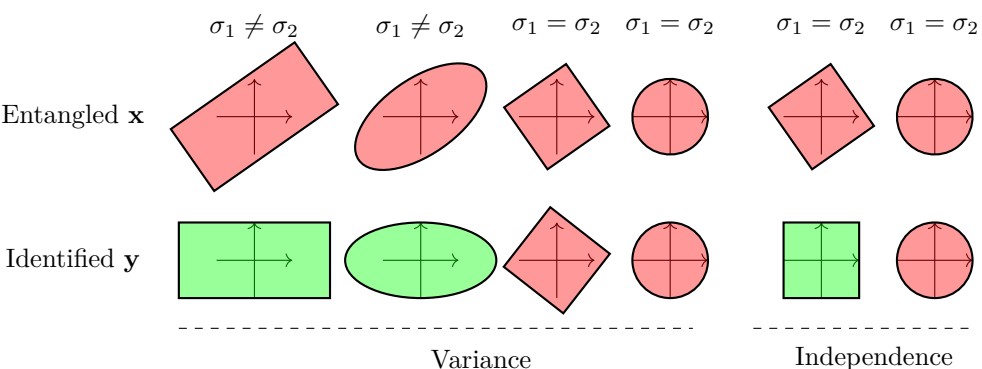

Figure 1: Identifiability of rotations using variance or independence maximization. The aim is to find a rotation that transforms $\mathbf{x}$ into $\mathbf{y}$, aligning the axes. Red indicates unaligned; green, aligned. For the rectangle and the ellipse, representing distinct variances, we can figure out a rotation, except that we do not know if we should flip up or down, left or right – a sign indeterminacy. For the square and the circle, representing equal variance, we cannot figure out a rotation only from the variance. For the square, however, we can use independence to figure out a rotation, except that we do not know if we should flip and/or permute the vertical and horizontal axes – sign and permutational indeterminacies. For the circle, nothing can be done. There are no favoured directions; even if we apply a rotation, it would remain the same – a rotational indeterminacy. A Gaussian is like a circle.

### 2.4.1 Semi-orthogonal transformation

Suppose that $\mathbf{B}_*$ is semi-orthogonal, i.e. $\mathbf{B}_{(L)}^{-1} = \mathbf{B}_*^T$ with $\mathbf{B}_*^T\mathbf{B}_* = \mathbf{I}$. Let $\mathbf{E}_*$ denote the matrix of independent orthonormal eigenvectors ordered by their variances, so that true transformation is $\mathbf{B}_* = \mathbf{E}_*$.

The primary method for learning semi-orthogonal transformations is linear PCA. The extent of how much it can identify the true transformation depends on the variances of the components:

$$\mathbf{B} = \begin{cases} \mathbf{E}_*\mathbf{I}_\pm, & \text{if all the variances are distinct – identifiable;} \\ \mathbf{E}_*\mathbf{I}_\pm\mathbf{R}_s, & \text{if a subset of the variances are equal – partially identifiable;} \\ \mathbf{E}_*\mathbf{I}_\pm\mathbf{R}, & \text{if all the variances are equal – unidentifiable.} \end{cases} \tag{2.15}$$

If $\mathbf{B}_*$ had orthogonal but non-orthonormal columns, then there is also a scale indeterminacy, arising because we can introduce a diagonal matrix $\mathbf{\Lambda}$ such that $\mathbf{B}_{(L)}^{-1}\mathbf{B} = \mathbf{B}^T\mathbf{\Lambda}^{-1}\mathbf{\Lambda}\mathbf{B} = \mathbf{B}'^{-1}_{(L)}\mathbf{B}'$.

Although conventional nonlinear PCA can resolve the rotational indeterminacy (Karhunen & Joutsensalo, 1994; 1995; Karhunen et al., 1998), it currently can only work after linearly transforming the inputs into decorrelated components with unit variance – after a preprocessing step called whitening. This can be better understood when we consider the non-orthogonal transformation.

### 2.4.2 Non-orthogonal transformation

Suppose that $\mathbf{B}_*$ is non-orthogonal. With a non-orthogonal matrix, the columns do not necessarily have unit norm, and so, without any prior information, there is a scale indeterminacy, arising because we can introduce a diagonal matrix $\mathbf{\Lambda}$ such that $\mathbf{B}_{(L)}^{-1}\mathbf{B} = \mathbf{B}_{(L)}^{-1}\mathbf{\Lambda}^{-1}\mathbf{\Lambda}\mathbf{B} = \mathbf{B}'^{-1}_{(L)}\mathbf{B}'$ – there is no way to tell what $\mathbf{\Lambda}$ is. Let us suppose that the true transformation is $\mathbf{B}_* = \mathbf{U}_*\mathbf{\Lambda}_*$, where $\mathbf{\Lambda}_*$ represents the standard deviations of the independent components, and $\mathbf{U}_*$ represents the linear transformation that results in unit variance components. The scale $\mathbf{\Lambda}_*$ can only ever be recovered if prior knowledge is available about the norm of the columns of $\mathbf{B}_*$ or $\mathbf{B}_{*(L)}^{-1}$, but that is rarely the case.

Linear ICA allows us to recover $\mathbf{U}_*$ up to sign and permutation indeterminacies. It exploits the fact that once the data has been whitened – rotated and scaled into uncorrelated components of **unit** variance – then any

other subsequent rotation will maintain unit variance. What this means is that if $\mathbf{A} \in \mathbb{R}^{p \times k}$ is a whitening matrix, then $\mathbb{E}((\mathbf{xA})^T \mathbf{xA}) = \mathbf{I}$, and if we attempt to apply another rotation $\mathbf{V} \in O(k)$, then

$$\mathbb{E}(\mathbf{V}^T \mathbf{A}^T \mathbf{x}^T \mathbf{xAV}) = \mathbf{V}^T \mathbb{E}((\mathbf{xA})^T \mathbf{xA}) \mathbf{V} = \mathbf{V}^T \mathbf{V} = \mathbf{I}. \tag{2.16}$$

Therefore, $\mathbf{U}$ can be recovered in two stages by first finding a whitening transformation $\mathbf{A}$ and then finding a rotation $\mathbf{V}$ that maximizes independence, resulting in $\mathbf{U} = \mathbf{AV}$. We already have a method for finding a whitening matrix: linear PCA, resulting in $\mathbf{A} = \mathbf{E}_* \mathbf{I}_{\pm} \mathbf{R}_s \boldsymbol{\Sigma}^{-1}$. What is left now is to find $\mathbf{V}$. For that, one method is the fast fixed-point algorithm of FastICA (Hyvarinen, 1999); an alternative is conventional nonlinear PCA (Karhunen & Joutsensalo, 1994). If $\mathbf{U}_* = \mathbf{E}_* \boldsymbol{\Sigma}^{-1} \mathbf{V}_*$, then

$$\mathbf{V} = \mathbf{R}_s^T \mathbf{I}_{\pm} \mathbf{V}_* \mathbf{I}'_{\pm} \mathbf{P}, \tag{2.17}$$

resulting in

$$\mathbf{U} = \mathbf{AV} = \mathbf{E}_* \mathbf{I}_{\pm} \mathbf{R}_s \boldsymbol{\Sigma}^{-1} \mathbf{R}_s^T \mathbf{I}_{\pm} \mathbf{V}_* \mathbf{I}'_{\pm} \mathbf{P} = \mathbf{E}_* \boldsymbol{\Sigma}^{-1} \mathbf{V}_* \mathbf{I}'_{\pm} \mathbf{P} = \mathbf{U}_* \mathbf{I}'_{\pm} \mathbf{P}. \tag{2.18}$$

## 2.5 Conventional neural linear PCA

A single-layer tied linear autoencoder has a solution that lies in the PCA subspace (Baldi & Hornik, 1989; Bourlard & Kamp, 1988; Oja, 1992b). Let $\mathbf{X} \in \mathbb{R}^{n \times p}$ be the **centred** data matrix, with $\mathbf{x}$ a row vector of $\mathbf{X}$, and $\mathbf{W} \in \mathbb{R}^{p \times k}$, with $k \leq p$, then we can write the autoencoder reconstruction loss as

$$\mathcal{L}(\mathbf{W}) = \mathbb{E}(\ell(\mathbf{W})) = \mathbb{E}(||\mathbf{x} - \mathbf{xWW}^T||_2^2). \tag{2.19}$$

From this we see that the autoencoder transforms the data with $\mathbf{W}$ (the encoder) into a set of components $\mathbf{y} = \mathbf{xW}$, and then it reconstructs the original data with $\mathbf{W}^T$ (the decoder) to obtain the reconstruction $\hat{\mathbf{x}} = \mathbf{yW}^T$. If $k < p$ then it performs dimensionality reduction. The loss naturally enforces $\mathbf{W}^T \mathbf{W} = \mathbf{I}$, and when it is enforced, the contribution to the gradient originating from the encoder is zero (see Appendix A.2.1). Given that, the linear PCA loss can be equally expressed in a form that results in the subspace learning algorithm (Oja, 1989) weight update (see Appendix A.2.2):

$$\mathcal{L}(\mathbf{W}) = \mathbb{E}(||\mathbf{x} - \mathbf{x}[\mathbf{W}]_{sg} \mathbf{W}^T||_2^2), \tag{2.20}$$

where $[\ \ ]_{sg}$ is the stop gradient operator, an operator that marks the value of the weight as a constant – i.e. the weight is not treated as a variable that contributes to the gradient.

Let $\mathbf{E}_* \in \mathbb{R}^{p \times k}$ be the matrix whose columns are ordered orthonormal independent eigenvectors, then the minimum of Eq. 2.19 or 2.20 is obtained by **any** orthonormal basis of the PCA subspace spanned by the columns of $\mathbf{E}_*$. That is, the optimal $\mathbf{W}$ has the form

$$\mathbf{E}_* \mathbf{I}_{\pm} \mathbf{R}_s \mathbf{R}, \tag{2.21}$$

where $\mathbf{R} \in O(k)$ is an indeterminacy over the entire space, resulting in a combined indeterminacy $\mathbf{R}_s \mathbf{R}$, which does not affect the loss, given that $\mathbf{E}_* \mathbf{I}_{\pm} \mathbf{R}_s \mathbf{R} \mathbf{R}^T \mathbf{R}_s^T \mathbf{I}_{\pm}^T \mathbf{E}_*^T = \mathbf{E}_* \mathbf{E}_*^T$. If $\mathbf{R} = \mathbf{I}$, then the linear PCA solution $\mathbf{E}_* \mathbf{I}_{\pm} \mathbf{R}_s$ can be recovered. The indeterminacy $\mathbf{R}$ appears is because the loss is symmetric: no component is favoured over the other. To eliminate $\mathbf{R}$, we simply need to break the symmetry (see Appendix A.3).

### 2.6 Conventional neural nonlinear PCA

A straightforward extension (Xu, 1993; Karhunen & Joutsensalo, 1994; 1995; Karhunen et al., 1998) from linear to nonlinear PCA is to simply introduce in Eq. 2.19 a nonlinearity $h$, such as tanh to obtain

$$\mathcal{L}(\mathbf{W}) = \mathbb{E}(||\mathbf{x} - h(\mathbf{xW})\mathbf{W}^T||_2^2). \tag{2.22}$$

This loss tends to work only when the input is already whitened. This means that, when the input has unit variance, this loss can recover a square orthogonal matrix that maximizes statistical independence. It can be shown that it maximizes statistical independence as exact relationships can be established between the nonlinear PCA loss and several ICA methods (Karhunen et al., 1998). In particular, these relationships become more apparent when noting that, under the constraint $\mathbf{W}^T\mathbf{W} = \mathbf{I}$, Eq. 2.22 can also be written as

$$\mathcal{L}(\mathbf{W}) = \mathbb{E}(||\mathbf{y} - h(\mathbf{y})||_2^2), \tag{2.23}$$

which is similar to the Bussgang criterion used in blind source separation (Lambert, 1996; Haykin, 1996).

Unlike linear PCA, conventional nonlinear PCA cannot recover a useful semi-orthogonal matrix that reduces dimensionality. With the current formulation, the mere introduction of a nonlinear function creates a jarring disparity between both linear and nonlinear PCA. Ideally, we would like to be able to apply nonlinear PCA directly on the input to reduce dimensionality while maximizing both variance and statistical independence, allowing us to order by variances and to eliminate the subspace rotational indeterminacy.

## 3 $\sigma$-PCA: a unified neural model for linear and nonlinear PCA

Instead of generalizing directly from the linear reconstruction loss from Eq. 2.19, let us take a step back and consider the loss with $\mathbf{B} \in \mathbb{R}^{p \times k}$, a left invertible transformation:

$$\mathbb{E}(||\mathbf{x} - h(\mathbf{xB})\mathbf{B}_{(L)}^{-1}||_2^2). \tag{3.1}$$

Conventional nonlinear PCA sets $\mathbf{B} = \mathbf{W}$, with $\mathbf{W}$ orthogonal. But if we consider that (1) the conventional nonlinear PCA loss requires unit variance input, and (2) the linear PCA transformation, whether from the SVD decomposition (Eq. 2.14) or from the whitening transformation, takes the form $\mathbf{W}\mathbf{\Sigma}^{-1}$, then it seems natural that $\mathbf{B}$ should not be just $\mathbf{W}$ but $\mathbf{W}\mathbf{\Sigma}^{-1}$, with $\mathbf{\Sigma}$ a diagonal matrix of the standard deviations. The transformation $\mathbf{W}\mathbf{\Sigma}^{-1}$ standardizes the components to unit variance before the nonlinearity $h$. We thus propose a **general** form of the unified linear and nonlinear PCA model: $\sigma$-PCA. Its reconstruction loss is

$$\mathcal{L}(\mathbf{W}, \mathbf{\Sigma}) = \mathbb{E}(||\mathbf{x} - h(\mathbf{xW}\mathbf{\Sigma}^{-1})\mathbf{\Sigma}\mathbf{W}^T||_2^2), \tag{3.2}$$

with $\mathbf{W}^T\mathbf{W} = \mathbf{I}$. Formulating it this way allows us to unify linear and nonlinear PCA. Indeed, if $h$ is linear, then we recover the linear PCA loss: $\mathbb{E}(||\mathbf{x} - \mathbf{xW}\mathbf{\Sigma}^{-1}\mathbf{\Sigma}\mathbf{W}^T||_2^2) = \mathbb{E}(||\mathbf{x} - \mathbf{xW}\mathbf{W}^T||_2^2)$.

From this **general** form of the loss, we will see that we can derive **specific** losses for linear PCA, nonlinear PCA, and linear ICA in equations 3.19, 3.20, and 3.21, respectively.

### 3.1 A closer look at the gradient

To derive the new nonlinear PCA loss from the general form in Eq. 3.2, we need to make one modification: we need to omit, from the weight update, the decoder contribution. To understand why, we need to have a

closer look at the individual contributions of both the encoder and the decoder to the update of $\mathbf{W}$. Given that gradient contributions are additive, we will be computing each contribution separately then adding them up. This is as if we untie the weights of the encoder ($\mathbf{W}_e$, $\mathbf{\Sigma}_e$) and the decoder ($\mathbf{W}_d$, $\mathbf{\Sigma}_d$), compute their respective gradients, and then tie them back up, summing the contributions. Without loss of generality, we will look at the stochastic gradient descent update (i.e. the loss $\ell$ of a single sample), and multiply by $\frac{1}{2}$ for convenience, so as to compute the gradients of

$$\ell(\mathbf{W}_e, \mathbf{W}_d, \mathbf{\Sigma}_e, \mathbf{\Sigma}_d) = \frac{1}{2}||\mathbf{x} - h(\mathbf{x}\mathbf{W}_e\mathbf{\Sigma}_e^{-1})\mathbf{\Sigma}_d\mathbf{W}_d^T||_2^2. \tag{3.3}$$

Let $\mathbf{y} = \mathbf{x}\mathbf{W}_e$, $\mathbf{z} = \mathbf{y}\mathbf{\Sigma}_e^{-1}$, $\hat{\mathbf{x}} = h(\mathbf{z})\mathbf{\Sigma}_d\mathbf{W}_d^T$, and $\hat{\mathbf{y}} = \hat{\mathbf{x}}\mathbf{W}_e$. By computing the gradients, we get

$$\frac{\partial\ell}{\partial\mathbf{W}_e} = \mathbf{x}^T(\hat{\mathbf{x}} - \mathbf{x})\mathbf{W}_d\mathbf{\Sigma}_d \odot h'(\mathbf{z})\mathbf{\Sigma}_e^{-1} \tag{3.4}$$

$$\frac{\partial\ell}{\partial\mathbf{W}_d} = (\hat{\mathbf{x}} - \mathbf{x})^T h(\mathbf{z})\mathbf{\Sigma}_d \tag{3.5}$$

Now that we have traced the origin of each contribution, we can easily compute the gradient when the weights are tied, i.e. $\mathbf{W} = \mathbf{W}_e = \mathbf{W}_d$ and $\mathbf{\Sigma} = \mathbf{\Sigma}_e = \mathbf{\Sigma}_d$, but we can still inspect each contribution separately:

$$\frac{\partial\ell}{\partial\mathbf{W}_e} = \mathbf{x}^T(\hat{\mathbf{y}} - \mathbf{y}) \odot h'(\mathbf{z}) \tag{3.6}$$

$$\frac{\partial\ell}{\partial\mathbf{W}_d} = (\hat{\mathbf{x}} - \mathbf{x})^T h(\mathbf{z})\mathbf{\Sigma} \tag{3.7}$$

$$\frac{\partial\ell}{\partial\mathbf{W}} = \frac{\partial\ell}{\partial\mathbf{W}_e} + \frac{\partial\ell}{\partial\mathbf{W}_d}. \tag{3.8}$$

From this, we can see the following: the contribution of the decoder, $\frac{\partial\ell}{\partial\mathbf{W}_d}$, will be zero when $\hat{\mathbf{x}} \approx \mathbf{x}$, and that of the encoder, $\frac{\partial\ell}{\partial\mathbf{W}_e}$, when $\hat{\mathbf{y}} \approx \mathbf{y}$. Thus, unsurprisingly, the decoder puts the emphasis on the input reconstruction, while the encoder puts the emphasis on the latent reconstruction.

In the **linear** case, if $\mathbf{W}^T\mathbf{W} = \mathbf{I}$, the contribution of the encoder is zero. This is because we have $\frac{\partial\ell}{\partial\mathbf{W}_e} = \mathbf{x}^T\mathbf{y}(\mathbf{W}^T\mathbf{W} - \mathbf{I}) = \mathbf{0}$. In fact, apart from acting as an additional orthonormalization term, its removal does not change the solution. We can remove it from the gradient by using the stop gradient operator $[\quad]_{sg}$:

$$\ell(\mathbf{W}) = \frac{1}{2}||\mathbf{x} - \mathbf{x}[\mathbf{W}]_{sg}\mathbf{W}^T||_2^2. \tag{3.9}$$

This results in

$$\frac{\partial\ell}{\partial\mathbf{W}} = -\mathbf{x}^T\mathbf{y} + \mathbf{W}\mathbf{y}^T\mathbf{y}, \tag{3.10}$$

which is simply the subspace learning algorithm (Oja, 1989), obtained from combining variance maximization with an orthonormalization constraint (see Appendix A.2.2).

In the conventional extension from linear to nonlinear PCA, the contribution of the encoder was also removed (Karhunen & Joutsensalo, 1994), as it was deemed appropriate to simply generalize the subspace learning algorithm (Oja, 1989; Karhunen & Joutsensalo, 1994). But we find that it should be the other way around for the nonlinear case: what is essential for unwhitened inputs is not the contribution of the decoder, but the contribution of the encoder.

Indeed, in the **nonlinear** case, $\frac{\partial\ell}{\partial\mathbf{W}_e}$ is no longer zero. The problem is that it is overpowered by the decoder contribution, in particular when performing dimensionality reduction (see Appendix F.6). Arguably it would

be desirable that the latent of the reconstruction is as close as possible to the latent of the input, i.e. put more emphasis on latent reconstruction. This means that for the nonlinear case we want to remove $\frac{\partial \ell}{\partial \mathbf{W}_d}$ from the update and only keep $\frac{\partial \ell}{\partial \mathbf{W}_e}$ – which is opposite of the linear case. We thus arrive at what the loss should be in the nonlinear case:

$$\mathcal{L}_{\text{NLPCA}}(\mathbf{W}; \boldsymbol{\Sigma}) = \boxed{\mathbb{E}(||\mathbf{x} - h(\mathbf{x}\mathbf{W}\boldsymbol{\Sigma}^{-1})\boldsymbol{\Sigma}[\mathbf{W}^T]_{sg}||_2^2), \quad \text{subject to } ||\mathbf{W}[:,j]||_2 = 1, \forall j}, \qquad (3.11)$$

where $\mathbf{W}[:,j]$ is the $j$th column of $\mathbf{W}$. The corresponding gradient is

$$\frac{\partial \mathcal{L}_{\text{NLPCA}}}{\partial \mathbf{W}} = 2\mathbb{E}((\mathbf{x}^T h(\mathbf{y}\boldsymbol{\Sigma}^{-1})\boldsymbol{\Sigma}\mathbf{W}^T\mathbf{W} - \mathbf{x}^T\mathbf{y}) \odot h'(\mathbf{y}\boldsymbol{\Sigma}^{-1})). \qquad (3.12)$$

Although this loss naturally enforces the orthogonality of the columns of $\mathbf{W}$ by the presence of the multiplicative term $\mathbf{W}^T\mathbf{W}$, it does not maintain their unit norm because $\mathbf{W}$ and $\boldsymbol{\Sigma}$ can both be scaled by the same amount and $\mathbf{W}\boldsymbol{\Sigma}^{-1}$ would remain the same, justifying the need for a projective unit norm constraint, $\mathbf{W}[:,j] \leftarrow \frac{\mathbf{W}[:,j]}{||\mathbf{W}[:,j]||_2}$.

Though $\boldsymbol{\Sigma}$ could be trainable, without any constraints, it could increase without limit. For instance, if $h = \tanh$, we have $\lim_{\sigma \to \infty} \sigma \tanh(z/\sigma) = z$. One option is to add an $L_2$ regularizer, but $\boldsymbol{\Sigma}$ need not be trainable, as we already have a natural choice for it: the estimated standard deviation of $\mathbf{y}$. Indeed, we can set $\boldsymbol{\Sigma} = [\text{diag}(\sqrt{\text{var}(\mathbf{y})})]_{sg}$ (see Appendix F.2). We discuss what $h$ needs to be in Section 3.4.

## 3.2 Nonlinear PCA: an ICA method that maximizes both variance and independence

In the gradient update of nonlinear PCA (Eq. 3.12), we can see two terms, one corresponding to maximizing variance and the other to maximizing independence – both under the constraint $\mathbf{W}^T\mathbf{W} = \mathbf{I}$. These can be clearly seen in the stochastic weight update (omitting $h'$ as it is undone by unit normalization):

$$\Delta\mathbf{W} \propto \overbrace{\mathbf{x}^T\mathbf{y}}^{\text{variance}} - \underbrace{\mathbf{x}^T h(\mathbf{y}\boldsymbol{\Sigma}^{-1})\boldsymbol{\Sigma}\mathbf{W}^T\mathbf{W}}_{\text{independence + orthogonality}}. \qquad (3.13)$$

The variance maximizing term, $\mathbf{x}^T\mathbf{y}$, can be obtained in isolation by minimizing the loss $-||\mathbf{y}||_2^2$, which is the negative of the variance if taken in expectation. The independence maximizing term, $-\mathbf{x}^T h(\mathbf{y}\boldsymbol{\Sigma}^{-1})\boldsymbol{\Sigma}\mathbf{W}^T\mathbf{W}$, can be seen as a generalization of the linear ICA update to non-unit variance input; indeed, assuming a square $\mathbf{W} \in O(k)$ and unit variance, the variance term becomes constant in expectation, and the independence term reduces to $-\mathbf{x}^T h(\mathbf{y})$. The linear ICA update of Hyvärinen & Oja (1998) for pre-whitened inputs takes the form $\Delta\mathbf{W} \propto \pm\mathbf{x}^T f(\mathbf{y})$, with the sign mostly dependent on whether the distribution is sub- or super-Gaussian, determined by the sign of $\mathbb{E}(yf(y) - f'(y))$. If we set $f(y) = y - a\tanh(y/a)$, then we have $\mathbf{x}^T\mathbf{y} - \mathbf{x}^T h(\mathbf{y}) = \mathbf{x}^T f(\mathbf{y})$, thus recovering the exact same update, with the setting of the scalar $a$ similarly dependent on whether the distribution is sub- or super-Gaussian (see Section 3.4). Both variance and independence terms can be seen as Hebbian update rules, the former being linear, the latter nonlinear.

As linear ICA is based on maximizing non-Gaussianity, a general formulation (Comon, 1994; Hyvärinen & Oja, 1998) of the linear ICA loss is

$$\mathcal{J}(\mathbf{W}) = |\mathbb{E}(F(\mathbf{x}\mathbf{W})) - \mathbb{E}(F(\mathbf{n}))|, \qquad (3.14)$$

under the constraint $\mathbf{W}^T\mathbf{W} = \mathbf{I}$, where $\mathbf{n} \sim \mathcal{N}(\mathbf{0}, \mathbf{1})$ and $F$ is a smooth nonlinear function. This, however, assumes that the inputs are pre-whitened. For nonlinear PCA, we can make a generalization of this loss that takes the variances into account:

$$\mathcal{J}_{\boldsymbol{\Sigma}}(\mathbf{W}) = |\mathbb{E}(F(\mathbf{x}\mathbf{W}\boldsymbol{\Sigma}^{-1})\boldsymbol{\Sigma}) - \mathbb{E}(F(\mathbf{n})\boldsymbol{\Sigma}))|. \qquad (3.15)$$

And so, instead of comparing with a Gaussian with unit variance, we are now comparing with a Gaussian with different variances. As the term involving the Gaussian is constant (Hyvärinen & Oja, 1998), the linear ICA problem reduces to finding all the extrema points of the first term $\mathbb{E}(F(\mathbf{x}\mathbf{W}\boldsymbol{\Sigma}^{-1})\boldsymbol{\Sigma})$. Now, in a similar way that conventional nonlinear PCA (Karhunen et al., 1998) can be put in relation to linear ICA, we can put nonlinear PCA in relation to linear ICA by noting that if $\mathbf{W}^T\mathbf{W} = \mathbf{I}$ is enforced, then the nonlinear PCA loss can be re-expressed as

$$\mathbb{E}(||h(\mathbf{y}\boldsymbol{\Sigma}^{-1})\boldsymbol{\Sigma} - [\mathbf{y}]_{sg}||_2^2). \tag{3.16}$$

If we chose $F(z) = (h(z) - [z]_{sg})^2$, then the nonlinear PCA loss becomes equivalent to the loss in Eq. 3.15. Given that, it can be shown that the loss will lead the columns of $\mathbf{W}$ to converge to the independent eigenvectors if the independent components $z_i$ satisfy $\mathbb{E}(z_i F'(z_i) - F''(z_i)) \neq 0$, because, if the condition is satisfied, then the independent eigenvectors are stable stationary points of the loss. This guarantees **local convergence** to the extrema points – the proof of this is given in Hyvärinen & Oja (1998). Global convergence, however, cannot be easily shown, but numerical simulations tend to show that the linear ICA loss tends to converge globally (Hyvärinen & Oja, 1998; Karhunen et al., 1998).

Nonlinear PCA can be put in relation to other methods related to linear ICA, such as blind deconvolution and $L_1$ sparsity constraints – in particular a linear autoencoder with an $L_1$ regularization term. Both connections are discussed in Appendix F.3 and F.4.

### 3.3 Relationship between linear PCA, nonlinear PCA, and linear ICA

Linear PCA, nonlinear PCA, and linear ICA seek to find a matrix $\mathbf{B} \in \mathbb{R}^{p \times k}$ such that $\mathbf{y} = \mathbf{x}\mathbf{B}$. Linear PCA puts an emphasis on maximizing variance, linear ICA on maximizing statistical independence, and nonlinear PCA on both. All three assume that the resulting components of $\mathbf{y}$ have unit variance. This $\mathbf{B}$ is found by being decomposed into $\mathbf{W}\boldsymbol{\Sigma}^{-1}\mathbf{V}$, with $\mathbf{W} \in \mathbb{R}^{p \times k}$ semi-orthogonal, $\boldsymbol{\Sigma} \in \mathbb{R}^{k \times k}$ diagonal, and $\mathbf{V} \in O(k)$ orthogonal.

Orthogonality is a defining aspect of PCA, so in the case of both linear and nonlinear PCA, we have $\mathbf{V} = \mathbf{I}$, i.e. the restriction is for $\mathbf{B}$ to have orthogonal columns; there is no such restriction with ICA. Simply stated,

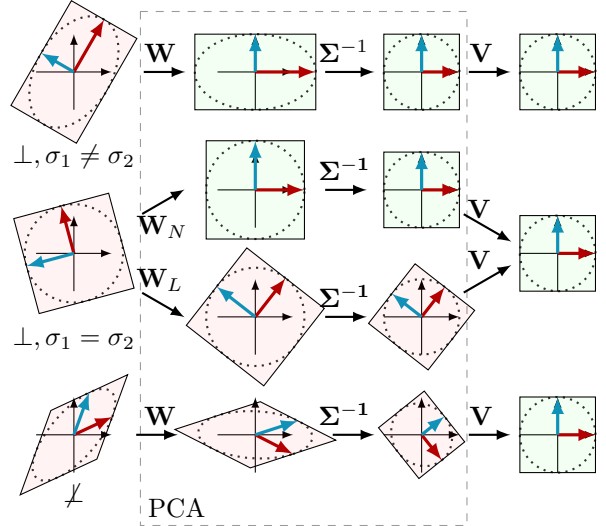

Figure 2: Linear PCA, nonlinear PCA, and linear ICA. Green indicates alignment of axes.

$$\text{(Non)linear PCA} \quad \mathbf{W}\boldsymbol{\Sigma}^{-1} \tag{3.17}$$
$$\text{Linear ICA} \quad \mathbf{W}\boldsymbol{\Sigma}^{-1}\mathbf{V}. \tag{3.18}$$

As part of the linear ICA transformation, linear PCA can only find the first rotation, $\mathbf{W}$, while nonlinear PCA can find both rotations, $\mathbf{W}$ and $\mathbf{V}$. Figure 2 summarizes the differences between all three methods based on the orthogonality of the overall transformation and the variances of the independent components.

The reconstruction losses of linear PCA (LPCA), nonlinear PCA (NLPCA), and linear ICA (LICA) follow the same pattern of $\mathbb{E}(||\mathbf{x} - h(\mathbf{x}\mathbf{B})\mathbf{B}_{(L)}^{-1}||_2^2)$, but with a few subtle differences, which we can write as

$$\mathcal{L}_{\text{LPCA}}(\mathbf{W}) = \mathbb{E}(||\mathbf{x} - \mathbf{x}\mathbf{W}\mathbf{W}^T||_2^2) \quad \text{or} \quad \mathbb{E}(||\mathbf{x} - \mathbf{x}[\mathbf{W}]_{sg}\mathbf{W}^T||_2^2), \tag{3.19}$$

$$\mathcal{L}_{\text{NLPCA}}(\mathbf{W}; \boldsymbol{\Sigma}) = \mathbb{E}(||\mathbf{x} - h(\mathbf{x}\mathbf{W}\boldsymbol{\Sigma}^{-1})\boldsymbol{\Sigma}[\mathbf{W}^T]_{sg}||_2^2), \tag{3.20}$$

$$\mathcal{L}_{\text{LICA}}(\mathbf{W}, \mathbf{V}; \boldsymbol{\Sigma}) = \mathbb{E}(||\mathbf{x}[\mathbf{W}\boldsymbol{\Sigma}^{-1}]_{sg} - h(\mathbf{x}[\mathbf{W}\boldsymbol{\Sigma}^{-1}]_{sg}\mathbf{V})\mathbf{V}^T||_2^2) + \mathcal{L}_{\text{NLPCA}}(\mathbf{W}; \boldsymbol{\Sigma}). \tag{3.21}$$

The value of $\mathbf{\Sigma}$ is set as $[\mathrm{diag}(\boldsymbol{\sigma})]_{sg}$, where $\boldsymbol{\sigma}^2 = \mathrm{var}(\mathbf{xW})$ (see Section 3.5). The nonlinear PCA loss, in particular, requires a unit norm constraint.

Linear PCA emphasizes input reconstruction, while nonlinear PCA emphasizes latent reconstruction. We can see this clearly from their losses and their respective weight updates:

$$\text{Linear PCA} \quad \mathbb{E}(||\mathbf{x} - \mathbf{x}[\mathbf{W}]_{sg}\mathbf{W}^T||_2^2), \tag{3.22}$$

$$\text{Nonlinear PCA} \quad \mathbb{E}(||\mathbf{x} - h(\mathbf{xW\Sigma}^{-1})\mathbf{\Sigma}[\mathbf{W}^T]_{sg}||_2^2) \tag{3.23}$$

$$\text{Linear PCA} \quad \Delta\mathbf{W} \propto \mathbf{x}^T\mathbf{y} - \mathbf{W}\mathbf{y}^T\mathbf{y} \qquad = \boxed{(\mathbf{x} - \hat{\mathbf{x}})^T\mathbf{y}}, \tag{3.24}$$

$$\text{Nonlinear PCA} \quad \Delta\mathbf{W} \propto (\mathbf{x}^T\mathbf{y} - \mathbf{x}^T\hat{\mathbf{x}}\mathbf{W}) \odot h'(\mathbf{y\Sigma}^{-1}) = \boxed{\mathbf{x}^T(\mathbf{y} - \hat{\mathbf{y}})} \odot h'(\mathbf{y\Sigma}^{-1}). \tag{3.25}$$

Unlike the nonlinear PCA and linear ICA losses, the linear PCA loss in Eq. 3.19 does not allow us to obtain a solution without a rotational indeterminacy, in addition to the subspace rotational indeterminacy (see Section 2.5), but there are multiple ways to obtain one by breaking the symmetry. We list many existing methods in Appendix A.3.

Omitting $h'$ from the stochastic update, as it is simply a scaling that is undone by unit normalization, in the weight update of linear and nonlinear PCA, we can see the variance, independence, and orthogonality terms:

$$\text{Linear PCA} \quad \Delta\mathbf{W} \propto \overbrace{\mathbf{x}^T\mathbf{y}}^{\text{variance}} - \overbrace{\mathbf{W}\mathbf{y}^T\mathbf{y}}^{\text{orthogonality}} \tag{3.26}$$

$$\text{Nonlinear PCA} \quad \Delta\mathbf{W} \propto \underbrace{\mathbf{x}^T\mathbf{y}}_{\text{variance}} - \underbrace{\mathbf{x}^T h(\mathbf{y\Sigma}^{-1})\mathbf{\Sigma}\mathbf{W}^T\mathbf{W}}_{\text{independence + orthogonality}} \tag{3.27}$$

Suppose that $\mathbf{E}_*$ is the matrix whose columns are the true ordered orthonormal eigenvectors, and let $\mathbf{W}_{\mathrm{L}}$ be the axis-aligned linear PCA solution, $\mathbf{W}_{\mathrm{N}}$ the nonlinear PCA solution, then we have

$$\text{Linear PCA} \qquad \mathbf{W}_{\mathrm{L}} = \begin{cases} \mathbf{E}_*\mathbf{I}_\pm, & \text{if all the variances are distinct -- identifiable;} \\ \mathbf{E}_*\mathbf{I}_\pm\mathbf{R}_s, & \text{if a subset of the variances are equal -- partially identifiable;} \\ \mathbf{E}_*\mathbf{I}_\pm\mathbf{R}, & \text{if all the variances are equal -- unidentifiable.} \end{cases} \tag{3.28}$$

$$\text{Nonlinear PCA} \qquad \mathbf{W}_{\mathrm{N}} = \begin{cases} \mathbf{E}_*\mathbf{I}_\pm, & \text{if all the variances are distinct -- identifiable;} \\ \mathbf{E}_*\mathbf{I}_\pm\mathbf{P}_s, & \text{if a subset of the variances are equal -- identifiable;} \\ \mathbf{E}_*\mathbf{I}_\pm\mathbf{P}, & \text{if all the variances are equal -- identifiable.} \end{cases} \tag{3.29}$$

Now suppose that the true transformation is non-orthogonal of the form $\mathbf{E}_*\mathbf{\Sigma}^{-1}\mathbf{V}_*$. Linear ICA can recover it in two ways, each consisting of two steps, akin to a two-layer model: (1) linear PCA followed by nonlinear PCA (with unit variance), and (2) nonlinear PCA followed by nonlinear PCA (with unit variance):

$$\text{Linear ICA} \quad \left.\begin{array}{ll} (1) & \mathbf{W}_{\mathrm{L}}\mathbf{\Sigma}^{-1}\mathbf{V}_{\mathrm{N}} = \mathbf{E}_*\mathbf{I}_\pm\mathbf{R}_s\mathbf{\Sigma}^{-1}\mathbf{R}_s^T\mathbf{I}_\pm\mathbf{V}_*\mathbf{I}_\pm'\mathbf{P} \\ (2) & \mathbf{W}_{\mathrm{N}}\mathbf{\Sigma}^{-1}\mathbf{V}_{\mathrm{N}}' = \mathbf{E}_*\mathbf{I}_\pm\mathbf{P}_s\mathbf{\Sigma}^{-1}\mathbf{P}_s^T\mathbf{I}_\pm\mathbf{V}_*\mathbf{I}_\pm'\mathbf{P} \end{array}\right\} = \mathbf{E}_*\mathbf{\Sigma}^{-1}\mathbf{V}_*\mathbf{I}_\pm'\mathbf{P} \tag{3.30}$$

The nonlinear PCA solution, $\mathbf{W}_{\mathrm{N}}$, exists within the linear ICA solution space. This is because if we consider the transformation $\mathbf{B} = \mathbf{W}\mathbf{\Sigma}^{-1}\mathbf{V}$, the nonlinear PCA solution space is restricted to block orthogonal matrices

$\mathbf{V}$ that can commute with $\mathbf{\Sigma}^{-1}$, i.e. $\mathbf{V}\mathbf{\Sigma}^{-1} = \mathbf{\Sigma}^{-1}\mathbf{V}$; whereas the linear ICA solution space includes not just the matrices $\mathbf{V}$ that commute, but also the ones that do not, i.e. $\mathbf{\Sigma}^{-1}\mathbf{V} \neq \mathbf{V}\mathbf{\Sigma}^{-1}$. When $\mathbf{V}$ does not commute the transformation $\mathbf{B}$ does not necessarily have orthogonal columns. Therefore, in theory – but not necessarily in practice as the linear ICA model is over-parametrized with a larger solution space[6] – the nonlinear PCA solution can be recovered using linear ICA (as linear PCA followed by conventional nonlinear PCA). If $\mathbf{V}_* = \mathbf{I}$, then when we apply linear ICA, we obtain

$$\mathbf{W}_{\mathrm{L}}\mathbf{\Sigma}^{-1}\mathbf{V} = \mathbf{E}_*\mathbf{I}_{\pm}\mathbf{R}_s\mathbf{\Sigma}^{-1}\mathbf{R}_s^T\mathbf{I}'_{\pm}\mathbf{P} = \mathbf{E}_*\mathbf{\Sigma}^{-1}\mathbf{I}''_{\pm}\mathbf{P}. \tag{3.31}$$

To reduce the whole space permutation indeterminacy, we can simply renormalize $\mathbf{E}_*\mathbf{\Sigma}^{-1}\mathbf{I}''_{\pm}\mathbf{P}$ to unit norm columns and reorder by the estimated variances to obtain $\mathbf{W}_{\mathrm{N}}$.

### 3.4 Choice of nonlinearity for sub- and super-Gaussian input distributions

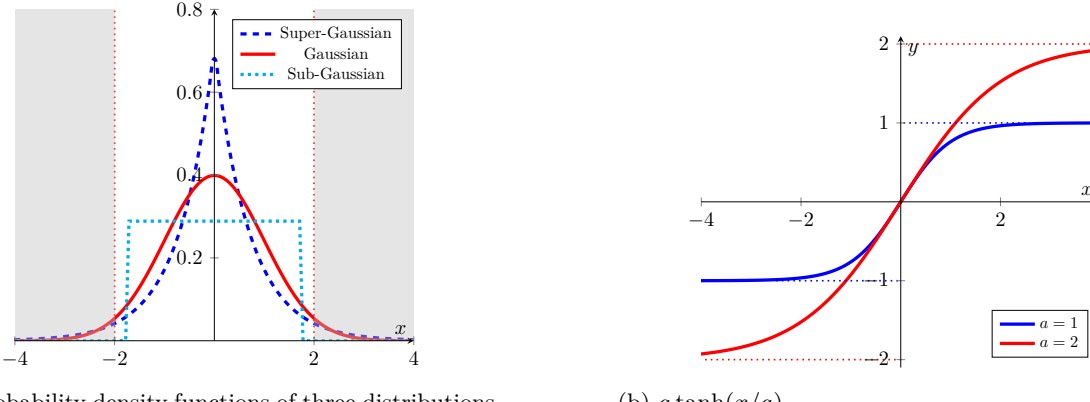

(a) Probability density functions of three distributions

(b) $a\tanh(x/a)$

Figure 3: Three distributions with unit variance (a): uniform distribution (sub-Gaussian), Gaussian distribution, and Laplace distribution (super-Gaussian). Shaded in grey is any $|x| \geq 2\sigma$. When using tanh without any scale adjustment, we can see that for the super-Gaussian distribution (or for any heavy-tailed distribution), values beyond $2\sigma$ might have their reconstruction impaired because of the squashing by tanh (b). A remedy, in this case, would be to use $a\tanh(x/a)$ with $a \geq 1$. For a sub-Gaussian distribution, it is more suited to use $a \leq 1$ as the values are within $2\sigma$.

The choice of nonlinearity generally depends on whether the distribution is sub- or super-Gaussian (Hyvärinen & Oja, 2000; Bingham et al., 2015). A typical choice is tanh, but to make tanh compatible with the distribution type, we need to use

$$h(z) = a\tanh(z/a), \tag{3.32}$$

with $a > 0$ instead of $\tanh(z)$. The choice of $a$ will generally depend on whether the distribution is sub- or super-Gaussian, with $a \leq 1$ for a sub-Gaussian and $a \geq 1$ for a super-Gaussian. Fundamentally, the scalar $a$ allows controlling the sign of $\mathbb{E}(zf(z) - f'(z))$, where $f(z) = z - a\tanh(z/a)$ (see Section 3.2 and Theorem 1 in Hyvärinen & Oja (1998)). From a reconstruction point of view, the reason why a super-Gaussian requires $a \geq 1$ is that it is more likely to be tail-heavy, so, given that $z$ is standardized, there will be more values greater $2\sigma$, which would result in tanh squashing them close 1 (see Fig. 3), making it unlikely to satisfy $\mathbb{E}(a\tanh(z/a)) \approx \mathbb{E}(z)$ for the stationary point (see Appendix F.1). Therefore, $a$ needs to be just large enough to allow good reconstruction but not too large that the function becomes close to linear. If $\mathbf{\Sigma}$ were trainable, a regularized $\mathbf{\Sigma}$ is more likely to work with super-Gaussian than sub-Gaussian inputs, since it will tend to

---

[6]If there are $m$ eigenspaces, then, by virtue of taking the variances into account, nonlinear PCA will tend to maximize independence within each eigenspace $\mathcal{S}_j$, whereas, by reducing all the eigenvectors to have the same eigenvalue, linear ICA would maximize independence over the entire space $\mathcal{S} = \mathcal{S}_1 \oplus ... \oplus \mathcal{S}_m$, which might not necessarily lead to the same result.

increase. Other options include setting $a$ to be trainable, using $\max(-a, \min(z, a))$ as an approximation of tanh, or using an asymmetric function – we briefly explored these during experiments in Appendix G.

## 3.5 Mean and variance of data

Mean centring is an important part of PCA (Diamantaras & Kung, 1996; Hyvärinen & Oja, 2000; Jolliffe & Cadima, 2016). So far, we have assumed that the input data is centred; if it is not, then it can easily be centred as a pre-processing step or during training. If the latter, we can explicitly write the loss function in Eq. 3.11 as

$$\mathcal{L}(\mathbf{W}) = \mathbb{E}(||\mathbf{x} - \boldsymbol{\mu}_x - (h(\frac{(\mathbf{x} - \boldsymbol{\mu}_x)\mathbf{W}}{\boldsymbol{\sigma}})\boldsymbol{\sigma})[\mathbf{W}^T]_{sg}||_2^2), \tag{3.33}$$

where $\boldsymbol{\mu}_x = \mathbb{E}(\mathbf{x})$ and $\boldsymbol{\sigma}^2 = [\text{var}(\mathbf{x}\mathbf{W})]_{sg}$. The mean and variance can be estimated on a batch of data, or, as is done with batch normalization (Ioffe & Szegedy, 2015), estimated using exponential moving averages (Appendix J). See Appendix F.8 for a non-centred variant.

## 3.6 Ordering of the components

After minimizing the nonlinear PCA loss, we can order the components based on $\boldsymbol{\Sigma}$. It is also possible to automatically induce the order based on index position. One such method is to introduce the projective deflation-like term $P(\mathbf{W}) = I - \unlhd(\mathbf{W}^T[\mathbf{W}]_{sg})$ into Eq. 3.11 as follows

$$\mathcal{L}(\mathbf{W}) = \mathbb{E}(||\mathbf{x} - h(\mathbf{x}\mathbf{W}P(\mathbf{W})\boldsymbol{\Sigma}^{-1})\boldsymbol{\Sigma}[\mathbf{W}^T]_{sg}||_2^2), \tag{3.34}$$

where $\unlhd$ is lower triangular without the diagonal. See Appendix F.7 for details and other methods.

# 4 Experiments

We applied the neural PCA models on image patches and time signals (see Appendix G for additional experiments); we optimized using gradient descent (see Appendix H for training details).

**Image patches** We extracted $11 \times 11$px overlapping patches, with zero padding, from a random subset of 500 images from CIFAR-10 (Krizhevsky et al., 2009), resulting in a total of 512K patches. We estimated $\boldsymbol{\Sigma}$ from the data and we used $a\tanh(z/a)$ with $a = 4$, though $a \in [3, 14]$ also worked well (see Appendix Fig. 12). We used $a > 1$ as a lot of natural images tend to have components with super-Gaussian distributions with similar variances (Hyvärinen et al., 2009). We applied linear PCA, nonlinear PCA, and FastICA (Hyvarinen, 1999), and summarized the results in Fig. 4. We see that the PCA filters (Fig. 4a) appear like blurred combinations of the filters found by nonlinear PCA (Fig. 4d). This stems from the subspace rotational indeterminacy of linear PCA for components with similar variances. We see that $a = 1$ does not work (Fig. 4d) in comparison to $a = 4$ (Fig. 4d). By slowly increasing $a$, we can have a smooth transition passing from linear PCA filters up to nonlinear PCA filters (see Appendix Fig. 12). We also see that conventional nonlinear PCA (Fig. 4g) failed to recover any meaningful filters. This is equally the case when the contribution of the decoder is not removed (Fig. 4f), highlighting that the presence of the decoder contribution is in fact detrimental.

**Time signals** We adapted a classic linear ICA example of signal separation (Comon, 1994) from scikit-learn (Pedregosa et al., 2011). First, we considered an orthogonal transformation. We generated three noisy signals: sinusoidal, square, and sawtooth. The last two had the same variance, and we mixed all three using a random **orthogonal** mixing matrix. We estimated the variance from data and used $a\tanh(z/a)$ with $a = 0.8$ since the distributions are sub-Gaussian. Figure 5 shows the result of applying linear and nonlinear PCA. As expected, linear PCA separated the sinusoidal signal, but not the square and sawtooth signals because they had the same variance. Nonlinear PCA recovered the three signals and their variances, up to sign

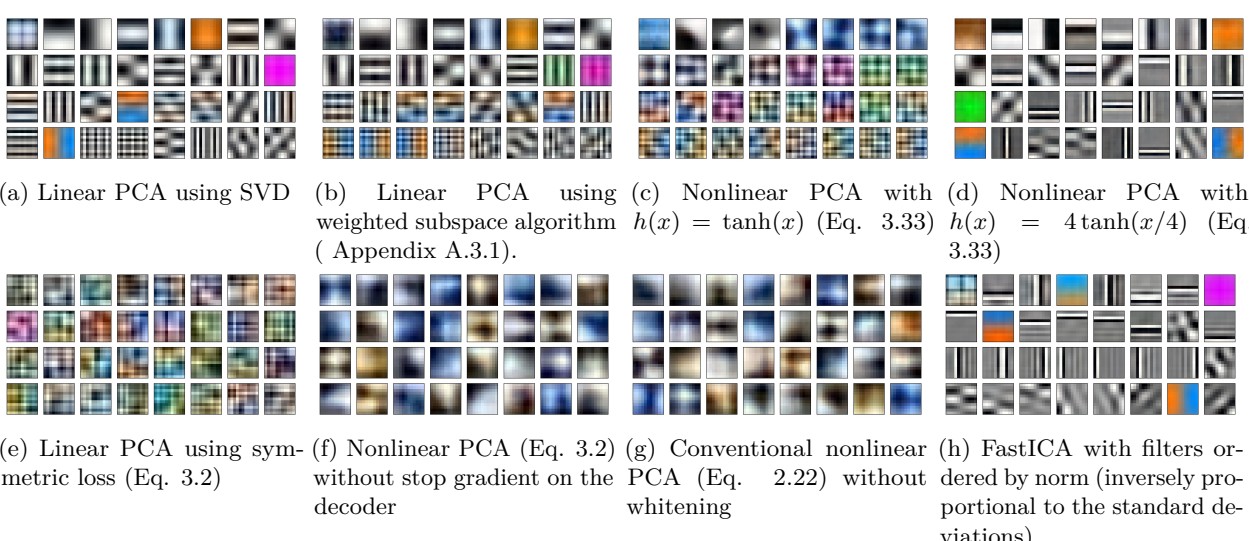

(a) Linear PCA using SVD

(b) Linear PCA using weighted subspace algorithm ( Appendix A.3.1).

(c) Nonlinear PCA with $h(x) = \tanh(x)$ (Eq. 3.33)

(d) Nonlinear PCA with $h(x) = 4\tanh(x/4)$ (Eq. 3.33)

(e) Linear PCA using symmetric loss (Eq. 3.2)

(f) Nonlinear PCA (Eq. 3.2) without stop gradient on the decoder

(g) Conventional nonlinear PCA (Eq. 2.22) without whitening

(h) FastICA with filters ordered by norm (inversely proportional to the standard deviations)

Figure 4: A set of 32 11x11px filters obtained on patches from the CIFAR-10 dataset. We obtained similar filters with the proposed unit-norm-preserving linear PCA loss (b) as the ones obtained via SVD (a). The filters obtained by nonlinear PCA (c, d) seem to have further separated the mixed filters from linear PCA. In particular, this is obvious by looking at the vertical and horizontal line filters at different positions that have been unmixed with nonlinear PCA. We see that setting $a = 1$ is not enough to separate (c), requiring a large $a$ such as $a = 4$ (d). We obtained meaningless filters in the PCA subspace with the symmetric linear PCA loss (e). We also obtained meaningless filters when the contribution of the decoder was included (f). This is similarly the case with conventional nonlinear PCA without whitening (g). FastICA (h) relaxes the orthogonality assumption of the overall transformation, so we see filters that are not necessarily orthogonal; however, there is some overlap with nonlinear PCA filters.

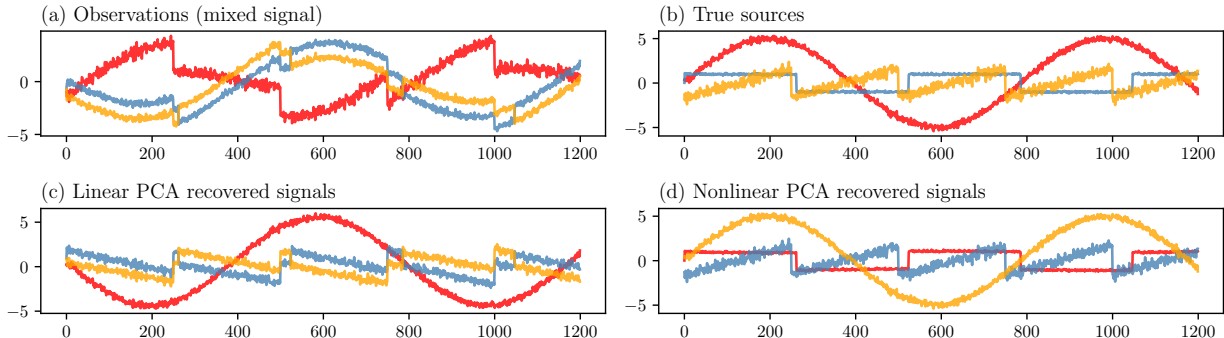

Figure 5: Three signals (sinusoidal, square, and sawtooth) that were mixed with an **orthogonal** mixing matrix. Linear PCA separated the sinusoidal signal as it had a distinct variance, but did not separate the square and the sawtooth signals as they had the same variance. Nonlinear PCA separated the signals and recovered their variances.

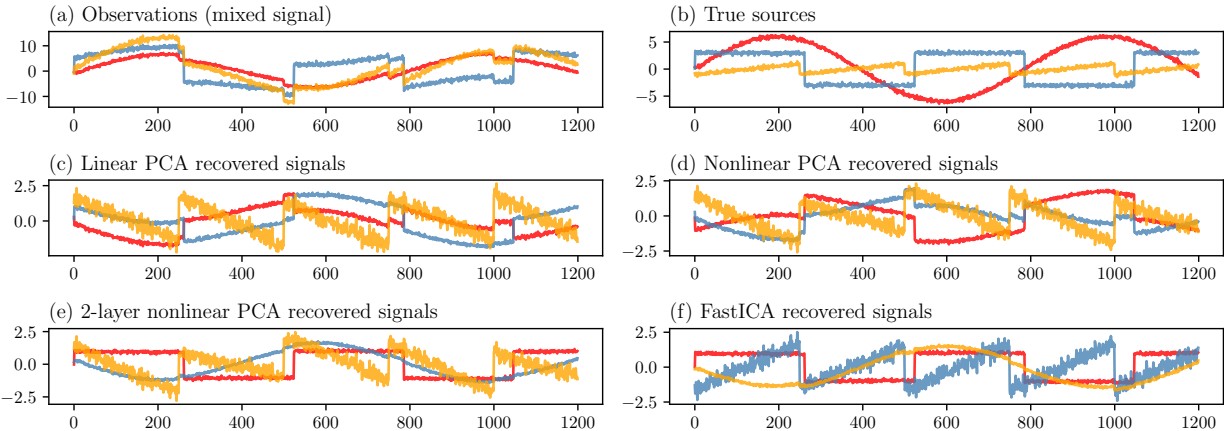

Figure 6: Three signals (sinusoidal, square, and sawtooth) that were mixed with a **non-orthogonal** mixing matrix. As expected, both linear (c) and nonlinear PCA (d) were not able to recover the signals. Only FastICA (f) and the 2-layer nonlinear PCA model (e) recovered the signals, but without their variances.

indeterminacies. Second, we considered a non-orthogonal transformation. We mixed the three signals, this time with distinct variances, using a **non-orthogonal** mixing matrix. We summarized the results in Fig. 6. As expected, both linear and nonlinear PCA did not recover the signals – they simple recovered the closest orthogonal transformation in terms of reconstruction loss. Both FastICA (linear PCA followed by the fast fixed-point algorithm) and the two-layer nonlinear PCA model (Eq. 3.21) recovered the signals, but without their variances due to the scale indeterminacy.

## 5 Related work

Hyvärinen (2015) previously proposed a unified model of linear PCA and linear ICA as a theoretical framework from the probabilistic paradigm, where the variances of the components are modelled as separate parameters but are eventually integrated out. Our key idea is the same: we model the variances as separate parameters, except that we do not integrate them out, and we flesh out the key role they play, from the neural paradigm, for nonlinear PCA that allows it to learn semi-orthogonal transformations that reduce dimensionality rather than just orthogonal transformations.

Neural PCA was started with the seminal work of Oja (1982) which showed that the first principal component can be extracted with a Hebbian (Hebb, 1949) learning rule. This spurred a lot of interest in PCA neural networks in the 80s and 90s, resulting in different variants and extensions for the extraction of multiple components (Foldiak, 1989; Oja, 1989; Rubner & Schulten, 1990; Kung & Diamantaras, 1990; Oja, 1992b; Bourlard & Kamp, 1988; Baldi & Hornik, 1989; Oja, 1992a; 1995) (see Diamantaras & Kung (1996) for an overview), some of which extracted the axis-aligned solution while others the subspace solution. A linear autoencoder with a mean squared error reconstruction loss was shown to extract the subspace solution (Bourlard & Kamp, 1988; Baldi & Hornik, 1989).

Unlike in the linear case, the term nonlinear PCA has been applied more broadly (Hyvärinen & Oja, 2000) to single-layer (Xu, 1993; Karhunen & Joutsensalo, 1994; Oja, 1995), multi-layer (Kramer, 1991; Oja, 1991; Scholz & Vigário, 2002), and kernel-based (Schölkopf et al., 1998) variants. As a single-layer autoencoder, the main emphasis has been its close connection to linear ICA, especially with whitened data (Karhunen et al., 1997; 1998; Hyvärinen & Oja, 2000). Linear ICA is a more powerful extension to PCA in that it seeks to find components that are non-Gaussian and statistically independent, with the overall transformation not necessarily orthogonal. Many algorithms have been proposed (see Hyvärinen & Oja (2000); Choi et al. (2005); Bingham et al. (2015) for an exhaustive overview), with the most popular being FastICA (Hyvarinen, 1999). Reconstruction ICA (RICA) (Le et al., 2011) is an autoencoder formulation which combines a linear reconstruction loss with $L_1$ sparsity; although it was proposed as a method for learning overcomplete ICA features, we show that RICA, in fact, is performing nonlinear PCA, except that it does not preserve unit norm (see Appendix F.4).

## 6 Discussion and conclusion

We have proposed $\sigma$-PCA, a unified neural model for linear and nonlinear PCA. This model allows nonlinear PCA to be on equal footing with linear PCA: just like linear PCA, nonlinear PCA can now learn a semi-orthogonal transformation that reduces dimensionality and orders by variances. But, unlike linear PCA, nonlinear PCA does not suffer from a subspace rotational indeterminacy: it can identify, disentangle, non-Gaussian components that have the same variance.

The mechanism by which nonlinear PCA eliminates the subspace rotational indeterminacy from the canonical linear PCA solution can be seen directly in its resulting weight update, where there are two terms that appear: one that maximizes variance, and another that maximizes independence.

Previously, it was only possible to apply conventional nonlinear PCA after whitening the input. This meant that when we consider the linear ICA transformation as a sequence of rotation, scale, rotation, then conventional nonlinear PCA could only be applied to learn the second rotation on unit variance input, but was not able to learn the first rotation. With our model, nonlinear PCA can now be applied to learn not just the second but also the first rotation. Although the nonlinear PCA solution implicitly exists in the linear ICA solution space, it is not necessarily explicitly recovered, because linear ICA has a far larger solution space, attempting to fit a linear transformation of the form $\mathbf{W}\mathbf{\Sigma}^{-1}\mathbf{V}$, whereas nonlinear PCA attempts to fit $\mathbf{W}\mathbf{\Sigma}^{-1}$, with $\mathbf{W}$ semi-orthogonal and $\mathbf{V}$ orthogonal.

An interesting aspect that emerged is that there is an elegant mirroring between linear and nonlinear PCA: the former puts an emphasis on input reconstruction, while the latter on latent reconstruction; the former relies on the decoder contribution, while the latter relies on the encoder contribution. With the $\sigma$-PCA model, we can have a smooth transition between both, rather than a harsh dichotomy. The advantage of nonlinear PCA is that it can be applied wherever linear PCA tends to be applied to learn (semi-)orthogonal transformations – with the added benefit that it can eliminate the subspace rotational indeterminacy from the canonical linear PCA solution, making it fully identifiable.

When applied on non-Gaussian inputs once, nonlinear PCA can learn identifiable semi-orthogonal transformations that reduce dimensionality, and, when applied twice, it overall can perform linear ICA to learn arbitrary unit-variance identifiable linear transformations. Nonlinear PCA, as part of our proposed $\sigma$-PCA formulation, thus serves as a building block for learning linear transformations that are identifiable.

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

# Appendix

## A    Linear PCA

In this section we go through different methods related to PCA and highlight their loss functions and weight updates where relevant. There are a variety of other neural methods for PCA; see (Diamantaras & Kung, 1996) for an in-depth review.

### A.1 PCA as eigendecomposition of covariance matrix or SVD of data matrix

Let $\mathbf{X} \in \mathbb{R}^{n \times p}$ be the **centred** data matrix and let $\mathbf{C} = \frac{1}{n}\mathbf{X}^T\mathbf{X}$ be the symmetric covariance matrix, then we can write the SVD of $\mathbf{X}$ as

$$\mathbf{X} = \mathbf{U}\mathbf{S}\mathbf{V}^T, \tag{A.1}$$

where $\mathbf{U} \in \mathbb{R}^{n \times p}$ is a semi-orthogonal matrix, $\mathbf{S} \in \mathbb{R}^{p \times p}$ is a diagonal matrix of singular values, and $\mathbf{V} \in \mathbb{R}^{p \times p}$ is an orthogonal matrix; and we can write the eigendecomposition of $\mathbf{C}$ as

$$\mathbf{C} = \mathbf{E}\mathbf{\Lambda}\mathbf{E}^T, \tag{A.2}$$

where $\mathbf{E} \in \mathbb{R}^{p \times p}$ is an orthogonal matrix of eigenvectors corresponding to the principal axes, and $\mathbf{\Lambda} \in \mathbb{R}^{p \times p}$ is a diagonal matrix corresponding to the variances.

We can relate both by noting that we have

$$\mathbf{C} = \frac{1}{n}\mathbf{X}^T\mathbf{X} = \frac{1}{n}\mathbf{V}\mathbf{S}^T\mathbf{U}^T\mathbf{U}\mathbf{S}\mathbf{V}^T \tag{A.3}$$

$$\mathbf{E}\mathbf{\Lambda}\mathbf{E}^T = \mathbf{V}\frac{1}{n}\mathbf{S}^2\mathbf{V}^T. \tag{A.4}$$

From this we can see that $\mathbf{E} = \mathbf{V}$ and $\mathbf{\Lambda} = \frac{1}{n}\mathbf{S}^2 = \mathrm{diag}(\boldsymbol{\sigma}^2)$.

### A.2 PCA subspace solutions

#### A.2.1 Linear autoencoder

Let $\mathbf{X}$ be the centred data matrix and $\mathbf{x}$ a row of $\mathbf{X}$, then we can write the reconstruction loss of a linear autoencoder as

$$\mathcal{L}(\mathbf{W}_e, \mathbf{W}_d) = \mathbb{E}(\ell(\mathbf{W}_e, \mathbf{W}_d)) = \frac{1}{2}\mathbb{E}(||\mathbf{x}\mathbf{W}_e\mathbf{W}_d^T - \mathbf{x}||_2^2), \tag{A.5}$$

where $\mathbf{W}_e \in \mathbb{R}^{p \times k}$ is the encoder and $\mathbf{W}_d^T \in \mathbb{R}^{p \times k}$ is the decoder. The stochastic loss yields the following gradients:

$$\frac{\partial \ell}{\partial \mathbf{W}_e} = \mathbf{x}^T(\mathbf{x}\mathbf{W}_e\mathbf{W}_d^T - \mathbf{x})\mathbf{W}_d \tag{A.6}$$

$$\frac{\partial \ell}{\partial \mathbf{W}_d} = (\mathbf{x}\mathbf{W}_e\mathbf{W}_d^T - \mathbf{x})^T\mathbf{x}\mathbf{W}_e. \tag{A.7}$$

Let us consider the tied weights case, i.e. $\mathbf{W}_e = \mathbf{W}_d = \mathbf{W}$, but keep the contributions separate. Let $\mathbf{y} = \mathbf{x}\mathbf{W}$, we can write

$$\frac{\partial \ell}{\partial \mathbf{W}_e} = \mathbf{x}^T(\mathbf{x}\mathbf{W}\mathbf{W}^T - \mathbf{x})\mathbf{W} \tag{A.8}$$

$$= \mathbf{x}^T\mathbf{x}\mathbf{W}(\mathbf{W}^T\mathbf{W} - \mathbf{I}) \tag{A.9}$$

$$= \mathbf{x}^T\mathbf{y}(\mathbf{W}^T\mathbf{W} - \mathbf{I}) \tag{A.10}$$

$$\frac{\partial \ell}{\partial \mathbf{W}_d} = (\mathbf{x}\mathbf{W}\mathbf{W}^T - \mathbf{x})^T\mathbf{x}\mathbf{W} \tag{A.11}$$

$$= (\mathbf{x}\mathbf{W}\mathbf{W}^T)^T\mathbf{x}\mathbf{W} - \mathbf{x}^T\mathbf{x}\mathbf{W} \tag{A.12}$$

$$= \mathbf{W}\mathbf{W}^T\mathbf{x}^T\mathbf{x}\mathbf{W} - \mathbf{x}^T\mathbf{x}\mathbf{W} \tag{A.13}$$

$$= (\mathbf{W}\mathbf{W}^T - \mathbf{I})\mathbf{x}^T\mathbf{x}\mathbf{W} \tag{A.14}$$

$$= (\mathbf{W}\mathbf{W}^T - \mathbf{I})\mathbf{x}^T\mathbf{y}. \tag{A.15}$$

We can note that the term $-\mathbf{x}^T\mathbf{y}$ appears in both the contributions of the encoder and the decoder, and this term is a variance maximization term. Indeed, given that the data is centred, we can write the variance maximization loss as

$$\mathcal{J}(\mathbf{W}) = -\frac{1}{2}\mathbb{E}(||\mathbf{x}\mathbf{W}||_2^2) = -\frac{1}{2}\mathbb{E}(||\mathbf{y}||_2^2), \tag{A.16}$$

which has the following gradient (omitting the expectation)

$$\frac{\partial \mathcal{J}}{\partial \mathbf{W}} = -\mathbf{x}^T\mathbf{x}\mathbf{W} = -\mathbf{x}^T\mathbf{y}. \tag{A.17}$$

Thus, the encoder multiplies the variance maximization term with $(\mathbf{W}^T\mathbf{W} - \mathbf{I})$ and the decoder with $(\mathbf{W}\mathbf{W}^T - \mathbf{I})$. If $\mathbf{W}$ is constrained to be orthogonal, i.e. $\mathbf{W}^T\mathbf{W} = \mathbf{I}$, then the contribution from the encoder is zero, and the contribution from the decoder is dominate. We can also note that the variance maximization of the latent results in a Hebbian update rule $\Delta \mathbf{W} \propto \mathbf{x}^T\mathbf{y}$ (Oja, 1982).

Let $\hat{\mathbf{x}} = \mathbf{y}\mathbf{W}^T$ and $\hat{\mathbf{y}} = \mathbf{x}\mathbf{W}$, then we can rewrite the contributions as

$$\frac{\partial \ell}{\partial \mathbf{W}_e} = \mathbf{x}^T(\hat{\mathbf{y}} - \mathbf{y}) \tag{A.18}$$

$$\frac{\partial \ell}{\partial \mathbf{W}_d} = (\hat{\mathbf{x}} - \mathbf{x})^T\mathbf{y}, \tag{A.19}$$

highlighting that the encoder puts an emphasis on latent reconstruction while the decoder puts an emphasis on input reconstruction. Rewritten as weight updates, we have

$$\Delta \mathbf{W}_e \propto \mathbf{x}^T\mathbf{y} - \mathbf{x}^T\mathbf{x}\mathbf{W} \tag{A.20}$$

$$\Delta \mathbf{W}_d \propto \mathbf{x}^T\mathbf{y} - \mathbf{W}\mathbf{y}^T\mathbf{y}. \tag{A.21}$$

### A.2.2   Subspace learning algorithm

The subspace learning algorithm (Oja, 1983; Williams & University of California, 1985; Oja, 1989; Hyvärinen & Oja, 2000) was proposed as a generalization of Oja's single component neural learning algorithm (Oja, 1982). The weight update has the following form:

$$\Delta\mathbf{W} = \mathbf{x}^T\mathbf{x}\mathbf{W} - \mathbf{W}(\mathbf{W}^T\mathbf{x}^T\mathbf{x}\mathbf{W}) \tag{A.22}$$

$$= \mathbf{x}^T\mathbf{y} - \mathbf{W}\mathbf{y}^T\mathbf{y}. \tag{A.23}$$

This update is obtained by maximizing the variance (minimizing the negative of the variance) under an orthonormality constraint:

$$\mathcal{L}(\mathbf{W}) = \mathbb{E}(\ell(\mathbf{W})) = -\frac{1}{2}\mathbb{E}(||\mathbf{x}\mathbf{W}||_2^2) \tag{A.24}$$

$$\text{subject to } \mathbf{W}^T\mathbf{W} = \mathbf{I}.$$

The variance maximizing term results in the gradient

$$\frac{\partial\ell}{\partial\mathbf{W}} = -\mathbf{x}^T\mathbf{y}. \tag{A.25}$$

Once combined with an orthonormalization update, this results in the following algorithm:

$$\mathbf{W}_{(i+1)} = \mathbf{W}_{(i)} + \eta\mathbf{x}^T\mathbf{x}\mathbf{W}_{(i)} \tag{A.26}$$

$$\mathbf{W}_{(i+1)} \leftarrow \mathbf{W}_{(i+1)}(\mathbf{W}_{(i+1)}^T\mathbf{W}_{(i+1)})^{-1/2}, \tag{A.27}$$

where $\eta$ is the learning rate. After expanding both equations as a power series of $\eta$ up to first order, and assuming $\mathbf{W}_{(i)}^T\mathbf{W}_{(i)} = \mathbf{I}$, we obtain

$$\mathbf{W}_{(i+1)}(\mathbf{W}_{(i+1)}^T\mathbf{W}_{(i+1)})^{-1/2} \approx \mathbf{W}_{(i+1)}(\mathbf{I} - \frac{1}{2}(\mathbf{W}_{(i+1)}^T\mathbf{W}_{(i+1)} - \mathbf{I})) \tag{A.28}$$

$$= (\mathbf{W}_{(i)} + \eta\mathbf{x}^T\mathbf{x}\mathbf{W}_{(i)})(\mathbf{I} - \eta\mathbf{W}_{(i)}^T\mathbf{x}^T\mathbf{x}\mathbf{W}_{(i)} + \mathcal{O}(\eta^2)) \tag{A.29}$$

$$= \mathbf{W}_{(i)} + \eta\mathbf{x}^T\mathbf{x}\mathbf{W}_{(i)} - \eta\mathbf{W}_{(i)}\mathbf{W}_{(i)}^T\mathbf{x}^T\mathbf{x}\mathbf{W}_{(i)} + \mathcal{O}(\eta^2). \tag{A.30}$$

Finally, this results in

$$\Delta\mathbf{W} = \mathbf{x}^T\mathbf{y} - \mathbf{W}\mathbf{y}^T\mathbf{y}. \tag{A.31}$$

The additional term $-\mathbf{W}\mathbf{y}^T\mathbf{y}$ thus serves as an orthonormalization term.

We see that this update is exactly the same as Eq. A.21: it is simply the decoder contribution from a linear reconstruction loss. As we have seen from Eq. A.9, the contribution of the encoder is zero if $\mathbf{W}$ is semi-orthogonal, so in the subspace learning algorithm the encoder contribution is simply omitted. The subspace learning algorithm can be obtained as a reconstruction loss with the stop gradient operator placed on the encoder:

$$\ell(\mathbf{W}) = \frac{1}{2}||\mathbf{x}[\mathbf{W}]_{sg}\mathbf{W}^T - \mathbf{x}||_2^2. \tag{A.32}$$

### A.3 Finding axis-aligned principal vectors

Here we look at the main idea for obtaining axis-aligned solutions: symmetry breaking. We saw in the previous section that the weight updates in the linear case are symmetric – no particular component is favoured over the other. Therefore, any method that breaks the symmetry will tend to lead to the axis-aligned PCA solution.

### A.3.1 Weighted subspace algorithm

One straightforward way to break the symmetry is to simply weigh each component differently. This has been shown to converge (Oja, 1992a; Oja et al., 1992; Xu, 1993; Hyvärinen & Oja, 2000) to the PCA eigenvectors.

The weighted subspace algorithm does exactly that, and it modifies the update rule of the subspace learning algorithm into

$$\Delta \mathbf{W} = \mathbf{x}^T \mathbf{y} - \mathbf{W} \mathbf{y}^T \mathbf{y} \boldsymbol{\Lambda}^{-1}, \tag{A.33}$$

$$\text{or } \Delta \mathbf{W} = \mathbf{x}^T \mathbf{y} \boldsymbol{\Lambda} - \mathbf{W} \mathbf{y}^T \mathbf{y}. \tag{A.34}$$

where $\boldsymbol{\Lambda} = \text{diag}(\lambda_1, ..., \lambda_k)$ such that $\lambda_1 > ... > \lambda_k > 0$.

The second form can be written as a loss function:

$$\ell(\mathbf{W}) = \frac{1}{2} \mathbb{E}(||\mathbf{x}[\mathbf{W}]_{sg} \mathbf{W}^T - \mathbf{x}||_2^2 - ||\mathbf{x} \mathbf{W} \boldsymbol{\Lambda}^{\frac{1}{2}}||_2^2 + ||\mathbf{x} \mathbf{W}||_2^2). \tag{A.35}$$

### A.3.2 Weighted subspace algorithm with unit norm

Although the above updates (Eq. A.33 and A.34) do converge to the eigenvectors, they no longer maintain unit norm columns. To see why, we can look at the stationary point, where we know that $\Delta \mathbf{W}$ should be $\mathbf{0}$ and that the components of $\mathbf{y}$ should be uncorrelated, i.e. $\mathbb{E}(\mathbf{y}^T \mathbf{y}) = \hat{\boldsymbol{\Sigma}}$ is diagonal. We can write:

$$\Delta \mathbf{W} = \mathbf{0} \tag{A.36}$$

$$\mathbb{E}(\mathbf{x}^T \mathbf{y} - \mathbf{W} \mathbf{y}^T \mathbf{y} \boldsymbol{\Lambda}^{-1}) = \mathbf{0} \tag{A.37}$$

$$\mathbb{E}(\mathbf{W}^T \mathbf{x}^T \mathbf{y} - \mathbf{W}^T \mathbf{W} \mathbf{y}^T \mathbf{y} \boldsymbol{\Lambda}^{-1}) = \mathbf{0} \tag{A.38}$$

$$\mathbb{E}(\mathbf{y}^T \mathbf{y}) - \mathbf{W}^T \mathbf{W} \mathbb{E}(\mathbf{y}^T \mathbf{y}) \boldsymbol{\Lambda}^{-1} = \mathbf{0} \tag{A.39}$$

$$\hat{\boldsymbol{\Sigma}}^2 - \mathbf{W}^T \mathbf{W} \hat{\boldsymbol{\Sigma}}^2 \boldsymbol{\Lambda}^{-1} = \mathbf{0} \tag{A.40}$$

$$\mathbf{W}^T \mathbf{W} \boldsymbol{\Lambda}^{-1} \hat{\boldsymbol{\Sigma}}^2 = \hat{\boldsymbol{\Sigma}}^2 \tag{A.41}$$

$$\mathbf{W}^T \mathbf{W} = \boldsymbol{\Lambda}. \tag{A.42}$$

As we know $\mathbf{W}$ has orthogonal columns, $\mathbf{W}^T \mathbf{W}$ must be diagonal, and so the norm of each $i$th column becomes equal to $\sqrt{\lambda_i}$. If we want unit norm columns, a straightforward remedy is to normalize at the end of training. A closer look allows us to see that the main contributor to the norm is the diagonal part of $\mathbf{y}^T \mathbf{y} \boldsymbol{\Lambda}^{-1}$ – this suggests to us two options that could maintain unit norm columns.

As a first option, we can simply counteract the effect of $\boldsymbol{\Lambda}^{-1}$ on the diagonal by multiplying by its inverse on the other side of $\mathbf{y}^T \mathbf{y}$ to obtain

$$\Delta \mathbf{W} = \mathbf{x}^T \mathbf{y} - \mathbf{W} (\mathbf{y} \boldsymbol{\Lambda})^T \mathbf{y} \boldsymbol{\Lambda}^{-1}, \tag{A.43}$$

$$\text{or } \Delta \mathbf{W} = \mathbf{x}^T \mathbf{y} \boldsymbol{\Lambda} - \mathbf{W} (\mathbf{y} \boldsymbol{\Lambda})^T \mathbf{y}, \tag{A.44}$$

for which we now have

$$\Delta \mathbf{W} = \mathbf{0} \tag{A.45}$$

$$\hat{\boldsymbol{\Sigma}}^2 - \mathbf{W}^T \mathbf{W} \boldsymbol{\Lambda} \hat{\boldsymbol{\Sigma}}^2 \boldsymbol{\Lambda}^{-1} = \mathbf{0} \tag{A.46}$$

$$\mathbf{W}^T \mathbf{W} \boldsymbol{\Lambda} \boldsymbol{\Lambda}^{-1} \hat{\boldsymbol{\Sigma}}^2 = \hat{\boldsymbol{\Sigma}}^2 \tag{A.47}$$

$$\mathbf{W}^T \mathbf{W} = \mathbf{I}. \tag{A.48}$$

As a second option, we can remove the diagonal part of $\mathbf{y}^T\mathbf{y}\boldsymbol{\Lambda}^{-1}$. This consists in adding

$$-\mathbf{W}(\text{diag}(\text{diag}^{-1}(\mathbf{y}^T\mathbf{y})) - \text{diag}(\text{diag}^{-1}(\mathbf{y}^T\mathbf{y}\boldsymbol{\Lambda}^{-1}))) \tag{A.49}$$

to Eq. A.33, or

$$-\mathbf{W}(\text{diag}(\text{diag}^{-1}(\mathbf{y}^T\mathbf{y}\boldsymbol{\Lambda})) - \text{diag}(\text{diag}^{-1}(\mathbf{y}^T\mathbf{y}))) \tag{A.50}$$

to Eq. A.34. The latter can be derived from the gradient of

$$\frac{1}{2}(||\mathbf{W}[\hat{\boldsymbol{\Sigma}}]_{sg}\boldsymbol{\Lambda}^{\frac{1}{2}}||_2^2 - ||\mathbf{W}[\hat{\boldsymbol{\Sigma}}]_{sg}||_2^2). \tag{A.51}$$

We thus arrive at a total loss that maintains unit norm columns:

$$\mathcal{L}(\mathbf{W}) = \frac{1}{2}\mathbb{E}(||\mathbf{x}[\mathbf{W}]_{sg}\mathbf{W}^T - \mathbf{x}||_2^2 + ||\mathbf{W}[\hat{\boldsymbol{\Sigma}}]_{sg}\boldsymbol{\Lambda}^{\frac{1}{2}}||_2^2 - ||\mathbf{W}[\hat{\boldsymbol{\Sigma}}]_{sg}||_2^2 - ||\mathbf{x}\mathbf{W}\boldsymbol{\Lambda}^{\frac{1}{2}}||_2^2 + ||\mathbf{x}\mathbf{W}||_2^2) \tag{A.52}$$

We see that this combines reconstruction, weighted regularization, and weighted variance maximization. We can also note from this that $\lambda_i$ should be $\leq 1$ to avoid the variance maximization term overpowering the other terms.

### A.3.3 Generalized Hebbian Algorithm (GHA)

The generalized Hebbian algorithm (GHA) (Sanger, 1989) learns multiple PCA components by combining Oja's update rule (Oja, 1982) with a Gram-Schmidt-like orthogonalisation term. It breaks the symmetry in the subspace learning algorithm by taking the lower triangular part of $\mathbf{y}^T\mathbf{y}$, allowing the weights to converge to the true PCA eigenvectors.

$$\mathbf{y} = \mathbf{x}\mathbf{W} \tag{A.53}$$

$$\Delta\mathbf{W} \propto \mathbf{x}^T\mathbf{y} - \mathbf{W}_{\unlhd}(\mathbf{y}^T\mathbf{y}) \tag{A.54}$$

One way to write this as a loss function is to use variance maximization and the stop gradient operator to get the following:

$$\ell(\mathbf{W}) = -\frac{1}{2}||\mathbf{x}\mathbf{W}||_2^2 + \mathbf{1}\mathbf{W}\odot[\mathbf{W}_{\unlhd}(\mathbf{y}^T\mathbf{y})]_{sg}\mathbf{1}^T. \tag{A.55}$$

### A.3.4 Autoencoder with GHA update

We can also consider combining the orthogonalisation term of GHA with the autoencoder reconstruction to get

$$\ell(\mathbf{W}) = ||\mathbf{x} - \mathbf{x}\mathbf{W}\mathbf{W}^T||_2^2 + \mathbf{1}\mathbf{W}\odot[\mathbf{W}_{\unlhd}(\mathbf{y}^T\mathbf{y})]_{sg}\mathbf{1}^T, \tag{A.56}$$

where $\unlhd$ is the operation that takes the lower triangular without the diagonal; otherwise, the term $-\mathbf{W}\text{diag}(\text{diag}^{-1}(\mathbf{y}^T\mathbf{y}))$ would be doubled unless we also include another $\mathbf{x}^T\mathbf{y}$ to counteract it. The double term might still work in practice, but it becomes detrimental when combined with a projective unit norm

constraint. If $\mathbf{W}$ is orthogonal, then this is similar to combining the subspace update rule (Eq. A.23) with the orthogonalisation term of GHA to result in

$$\Delta\mathbf{W} = \mathbf{x}^T\mathbf{y} - \mathbf{W}\mathbf{y}^T\mathbf{y} - \mathbf{W}_{\triangle}(\mathbf{y}^T\mathbf{y}), \tag{A.57}$$

which can be obtained from the following loss

$$\ell(\mathbf{W}) = ||\mathbf{x} - \mathbf{x}[\mathbf{W}]_{sg}\mathbf{W}^T||_2^2 + \mathbf{1}\mathbf{W} \odot [\mathbf{W}_{\triangle}(\mathbf{y}^T\mathbf{y})]_{sg}\mathbf{1}^T. \tag{A.58}$$

Nonetheless, keeping the encoder contribution can still help maintain the orthogonality of $\mathbf{W}$, as the term $\mathbf{W}^T\mathbf{W} - \mathbf{I}$ is similar to the one derived from a symmetric orthogonalisation regularizer (see Appendix B).

Another variant we can consider is to make triangular the full gradient of the linear autoencoder update, which includes not just the contribution of the decoder but also that of the encoder, i.e. by taking

$$\frac{\partial\ell}{\partial\mathbf{W}} = \mathbf{x}^T\mathbf{y}(\mathbf{W}^T\mathbf{W} - \mathbf{I}) - \mathbf{x}^T\mathbf{y} + \mathbf{W}\mathbf{y}^T\mathbf{y} \tag{A.59}$$

and changing it into

$$\frac{\partial\ell}{\partial\mathbf{W}} = \mathbf{x}^T\mathbf{y}(\triangle(\mathbf{W}^T\mathbf{W}) - \mathbf{I}) - \mathbf{x}^T\mathbf{y} + \mathbf{W}_{\triangle}(\mathbf{y}^T\mathbf{y}). \tag{A.60}$$

Though the contribution from the encoder is negligible when $\mathbf{W}$ is close to orthogonal, the term $\triangle\mathbf{W}^T\mathbf{W} - \mathbf{I}$ also acts as an approximation to Gram-Schmidt orthogonalisation (see Appendix B), and so when $\mathbf{W}$ deviates from being orthogonal, it can help bring it back more quickly.

### A.3.5 Nested dropout

Another way to break the symmetry and enforce an ordering is nested dropout (Rippel et al., 2014), a procedure for randomly removing nested sets of components. It can be shown (Rippel et al., 2014) that it leads to an exact equivalence with PCA in the case of a single-layer linear autoencoder. An idea similar in spirit was the previously proposed hierarchical PCA method (Scholz & Vigário, 2002) for ordering of the principal components; however, it was limited in number of dimensions and was not stochastic. Nested dropout works by assigning a prior distribution $p(.)$ over the component indices $1...k$, and then using it to sample an index $\mathcal{L}$ and drop the units $\mathcal{L} + 1, ..., k$. A typical choice of distribution is a geometric distribution with $p(j) = \rho^{j-1}(1 - \rho), 0 < \rho < 1$.

Let $\mathbf{m}_{1|j} \in \{0, 1\}^k$ be a vector such that $m_i = \begin{cases} 1 & \text{if } i < j \\ 0 & \text{otherwise} \end{cases}$, then the reconstruction loss becomes

$$\mathcal{L}(\mathbf{W}) = \frac{1}{2}\mathbb{E}(||(\mathbf{x}\mathbf{W} \odot \mathbf{m}_{1|j})\mathbf{W}^T - \mathbf{x}||_2^2), \tag{A.61}$$

where the mask $\mathbf{m}_{1|j}$ is randomly sampled during each update step. One issue with nested dropout is that the higher the index the smaller the gradient update. This means that training has to run much longer for them to converge. One suggested remedy for this (Rippel et al., 2014) is to perform gradient sweeping, where once an earlier index has converged, it can be frozen and the nested dropout index can be incremented.

A non-stochastic version, which is an extension of the hierarchical PCA method (Scholz & Vigário, 2002), sums all the reconstruction loss to result in

$$\mathcal{L}(\mathbf{W}) = \frac{1}{2} \sum_j \mathbb{E}(||\mathbf{x}\mathbf{W}_{1|j}\mathbf{W}_{1|j}^T - \mathbf{x}||_2^2), \tag{A.62}$$

where $\mathbf{W}_{1|j}$ is the truncated matrix that contains the first $j$ columns of $\mathbf{W}$. However, this can get more computationally demanding when there is a large number of components compared to the stochastic version.

### A.3.6 Weighted variance maximization

We can formulate a variant of the weighted subspace algorithm more explicitly as a reconstruction loss combined with a weighted variance maximization term:

$$\mathcal{L}(\mathbf{W}) = \frac{1}{2}\mathbb{E}(||\mathbf{x} - \mathbf{x}\mathbf{W}\mathbf{W}^T||_2^2 - \alpha||\mathbf{x}\mathbf{W}\mathbf{\Lambda}^{\frac{1}{2}}||_2^2), \tag{A.63}$$

where $\mathbf{\Lambda} = \text{diag}(\lambda_1, ..., \lambda_k)$ consists of linearly spaced values between 0 and 1, and $\alpha \in \{-1, 1\}$. We can also pick $\mathbf{\Lambda}$ to be proportional to the variances, but the term should not be trainable – we can do that with the stop gradient operator: $\mathbf{\Lambda} = [\frac{\mathbb{E}(\mathbf{y}^2)}{\max_i(\mathbb{E}(y_i^2))}]_{sg}$.

The associated gradient of the loss is

$$\frac{\partial \ell}{\partial \mathbf{W}} = -\mathbf{x}^T\mathbf{y} + \mathbf{W}\mathbf{y}^T\mathbf{y} + \mathbf{x}^T\mathbf{y}(\mathbf{W}^T\mathbf{W} - \mathbf{I}) - \alpha\mathbf{x}^T\mathbf{y}\mathbf{\Lambda} \tag{A.64}$$

$$= -\mathbf{x}^T\mathbf{y}(\mathbf{I} + \alpha\mathbf{\Lambda}) + \mathbf{W}\mathbf{y}^T\mathbf{y} + \mathbf{x}^T\mathbf{y}(\mathbf{W}^T\mathbf{W} - \mathbf{I}). \tag{A.65}$$

Following a similar analysis as in Appendix A.3.2, it can be shown that at the stationary point we have $\mathbf{W}^T\mathbf{W} = \mathbf{I} + \frac{1}{2}\alpha\mathbf{\Lambda}$.

Alternative, we can use a stochastic weighting by applying nested dropout on the variance regularizer:

$$\mathcal{L}(\mathbf{W}) = \mathbb{E}(||\mathbf{x} - \mathbf{x}\mathbf{W}\mathbf{W}^T||_2^2 - \alpha||\mathbf{x}\mathbf{W} \odot \mathbf{m}_{1|j}||_2^2), \tag{A.66}$$

with a randomly sampled mask per data sample.

## B    Enforcing orthogonality

There are a few ways to enforce orthogonality either via a regularizer or a projective constraint, and this can be symmetric or asymmetric.

### B.1    Symmetric

A symmetric orthogonalisation scheme can be used when there are no favoured components.

**Regularizer**    A straightforward regularizer, which also enforces unit norm, is

$$J(\mathbf{W}) = \alpha||\mathbf{I} - \mathbf{W}^T\mathbf{W}||_F^2, \tag{B.1}$$

and it has the gradient

$$\frac{\partial J}{\partial \mathbf{W}} = 4\alpha\mathbf{W}(\mathbf{W}^T\mathbf{W} - \mathbf{I}). \tag{B.2}$$

Combining the above with PCA or ICA updates has been previously referred to as the bigradient rule (Wang et al., 1995; Wang & Karhunen, 1996; Karhunen et al., 1997), where $\alpha$ is at most $\frac{1}{8}$, for which the justification can be derived from its corresponding projective constraint.

**Projective constraint** An iterative algorithm, as used in FastICA (Hyvarinen, 1999), is the following:

1. $\mathbf{w}_i \leftarrow \dfrac{\mathbf{w}_i}{||\mathbf{w}_i||}$

2. Repeat until convergence:

$$\mathbf{W} \leftarrow \frac{3}{2}\mathbf{W} - \frac{1}{2}\mathbf{W}\mathbf{W}^T\mathbf{W}.$$

Where the update $\frac{3}{2}\mathbf{W} - \frac{1}{2}\mathbf{W}\mathbf{W}^T\mathbf{W}$ stems from the Taylor expansion of $\mathbf{W}(\mathbf{W}^T\mathbf{W})^{-1/2}$ around $\mathbf{W}^T\mathbf{W} = \mathbf{I}$. We can note that

$$\frac{3}{2}\mathbf{W} - \frac{1}{2}\mathbf{W}\mathbf{W}^T\mathbf{W} = \mathbf{W} - \frac{1}{2}\mathbf{W}(\mathbf{W}^T\mathbf{W} - \mathbf{I}) \tag{B.3}$$

$$= \mathbf{W} - \frac{1}{8\alpha}\frac{\partial J}{\partial \mathbf{W}}, \tag{B.4}$$

which allows to to see that we can set $\alpha = \frac{1}{8}$ in Eq. B.2 to derive the projective constraint from the stochastic gradient descent update.

The iterative algorithm can be shown to converge (Hyvarinen, 1999; Hyvärinen & Oja, 2000) by analysing the evolution of the eigenvalues of $\mathbf{W}^T\mathbf{W}$. The initial normalization guarantees that, prior to any iterations, the eigenvalues of $\mathbf{W}^T\mathbf{W}$ are $\leq 1$. Let $\mathbf{E}^T\mathbf{\Lambda}\mathbf{E}$ be the eigendecomposition of $\mathbf{W}^T\mathbf{W}$, and let us consider the general case

$$\hat{\mathbf{W}} = \mathbf{W} - \beta\mathbf{W}(\mathbf{W}^T\mathbf{W} - \mathbf{I}) \tag{B.5}$$

$$= (1+\beta)\mathbf{W} - \beta\mathbf{W}\mathbf{W}^T\mathbf{W}. \tag{B.6}$$

After one iteration we have

$$\hat{\mathbf{W}}^T\hat{\mathbf{W}} = (1+\beta)^2\mathbf{W}^T\mathbf{W} - 2(1+\beta)\beta(\mathbf{W}^T\mathbf{W})^2 + \beta^2(\mathbf{W}^T\mathbf{W})^3 \tag{B.7}$$

$$= (1+\beta)^2\mathbf{E}^T\mathbf{\Lambda}\mathbf{E} - 2(1+\beta)\beta(\mathbf{E}^T\mathbf{\Lambda}\mathbf{E})^2 + \beta^2(\mathbf{E}^T\mathbf{\Lambda}\mathbf{E})^3 \tag{B.8}$$

$$= (1+\beta)^2\mathbf{E}^T\mathbf{\Lambda}\mathbf{E} - 2(1+\beta)\beta\mathbf{E}^T\mathbf{\Lambda}^2\mathbf{E} + \beta^2\mathbf{E}^T\mathbf{\Lambda}^3\mathbf{E} \tag{B.9}$$

$$= \mathbf{E}^T((1+\beta)^2\mathbf{\Lambda} - 2(1+\beta)\beta\mathbf{\Lambda}^2 + \beta^2\mathbf{\Lambda}^3)\mathbf{E}. \tag{B.10}$$

We now have the eigenvalues after one iteration as a function of the eigenvalues of the previous iteration:

$$f_\beta(\lambda) = (1+\beta)^2\lambda - 2(1+\beta)\beta\lambda^2 + \beta^2\lambda^3 \tag{B.11}$$

$$f_\beta'(\lambda) = (1+\beta)^2 - 4(1+\beta)\beta\lambda + 3\beta^2\lambda^2. \tag{B.12}$$

We know that the eigenvalues are in the interval $]0, 1]$. For $\lambda = 1$ we have

$$f_\beta(1) = 1 \tag{B.13}$$

$$f_\beta'(1) = (1+\beta)^2 - 4(1+\beta)\beta + 3\beta^2 = 1 - 2\beta. \tag{B.14}$$

We see that there is a stationary point at $\beta = \frac{1}{2}$. Noting that $f_0(\lambda) = \lambda$, we can plot $f_\beta$:

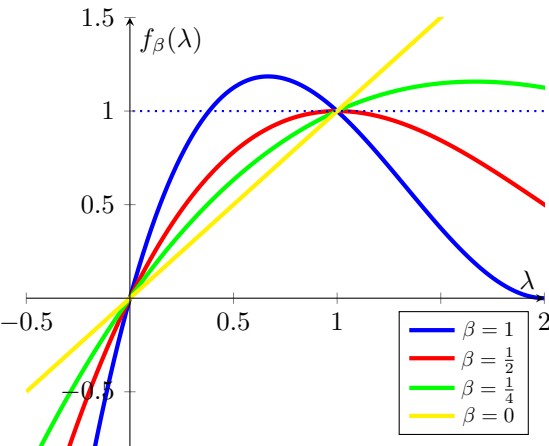

We see that if $\lambda \in ]0,1]$ and $\beta \in ]0, \frac{1}{2}]$, then the eigenvalues will always be in $]0,1]$; and given that $f(\lambda) > \lambda$, all the eigenvalues will eventually converge to 1.

Thus, for $\beta = \frac{1}{2}$ we get the fastest convergence where $\lambda$ does not exceed $]0,1]$. This means that when using stochastic gradient descent with a learning rate of 1 and with a symmetric regularizer (Eq. B.1), an optimal value for $\alpha$ is $\frac{1}{8}$. This analysis does not consider potential other interactions from the use of additional losses that could affect $\mathbf{W}$ or the use of an adaptive optimizer like Adam Kingma & Ba (2014).

### B.2 Asymmetric

**Regularizer**   From the symmetric regularizer we can derive an asymmetric version that performs a Gram-Schmidt-like orthogonalisation. We simply need to use the lower (or upper) triangular part of $\mathbf{W}^T\mathbf{W}$ while using a stop gradient on either $\mathbf{W}$ or $\mathbf{W}^T$. This means that the loss, which also maintains unit norm, is

$$J(\mathbf{W}) = \frac{1}{2}||\triangle(\mathbf{W}^T[\mathbf{W}]_{sg}) - \mathbf{I}||_F^2 \tag{B.15}$$

or without the diagonal (to remove the unit norm regularizer):

$$J(\mathbf{W}) = \frac{1}{2}||\triangle(\mathbf{W}^T[\mathbf{W}]_{sg})||_F^2, \tag{B.16}$$

where $\triangle$ refers to the lower triangular matrix without the diagonal. From this, we have

$$\frac{\partial J}{\partial \mathbf{W}} = \mathbf{W}\triangle(\mathbf{W}^T\mathbf{W} - \mathbf{I}), \tag{B.17}$$

or

$$\frac{\partial J}{\partial \mathbf{W}} = \mathbf{W}\triangle(\mathbf{W}^T\mathbf{W}). \tag{B.18}$$

The use of such an update has been previously referred to as the hierarchic version of the bigradient algorithm (Wang et al., 1995).

Recall that in the weight update of GHA (Appendix A.3.3) we have the term $\mathbf{W}\triangle(\mathbf{y}^T\mathbf{y})$; the asymmetric update can benefit from being at a similar scale in order to be effective, especially when the variances are large. To do this, we can weight the loss by the non-trainable standard deviations to obtain

$$J(\mathbf{W}) = \frac{1}{2}||\triangle(\mathbf{W}^T[\mathbf{W}\hat{\boldsymbol{\Sigma}}]_{sg})||_F^2. \tag{B.19}$$

**Gram-Schmidt projective constraint**   After each gradient update step, we can apply the Gram-Schmidt orthogonalisation procedure (Schmidt, 1907).

### B.3 Encoder contribution of linear reconstruction

As noted in Appendix A.2.1, in a linear autoencoder the contribution of the encoder to the gradient is zero when $\mathbf{W}$ is orthogonal, and so if used in isolation it can serve as an implicit orthonormality regularizer:

$$J(\mathbf{W}) = \mathbb{E}(||\mathbf{x} - \mathbf{x}\mathbf{W}[\mathbf{W}^T]_{sg}||_2^2), \tag{B.20}$$

having

$$\frac{\partial J}{\partial \mathbf{W}} = \mathbf{x}^T \mathbf{y}(\mathbf{W}^T\mathbf{W} - \mathbf{I}) \tag{B.21}$$

as gradient. We see that it simply replaces in the symmetric regularizer the first $\mathbf{W}$ with $\mathbf{x^T y}$.

## C Enforcing unit norm

If not already taken care of by the orthogonality constraint, there are a few ways to enforce unit norm.

### C.1 Projective constraint

The columns $\mathbf{w}_i^T$ of $\mathbf{W}$ can be normalized to unit norm after each update step:

$$\mathbf{w}_i^T \leftarrow \frac{\mathbf{w}_i^T}{||\mathbf{w}_i^T||}. \tag{C.1}$$

### C.2 Regularization

We can enforce unit norm by adding a regularizer on the norm of the column vectors $\mathbf{w}_i^T$. This results in the loss

$$J(\mathbf{w}_i^T) = \frac{1}{2}(\mathbf{1} - ||\mathbf{w}_i^T||_2)^2 \tag{C.2}$$

which has the following gradient

$$\frac{\partial J}{\partial \mathbf{w}_i^T} = (1 - \frac{1}{||\mathbf{w}_i^T||})\mathbf{w}_i^T \tag{C.3}$$

### C.3 Differentiable weight normalization

A third way to do this is to use differentiable weight normalization (Salimans & Kingma, 2016), except without the scale parameter. This consists in replacing the column vectors with

$$\hat{\mathbf{w}}_i^T = \frac{\mathbf{w}_i^T}{||\mathbf{w}_i^T||}. \tag{C.4}$$

This has the following Jacobian

$$\frac{\partial \hat{\mathbf{w}}_i^T}{\partial \mathbf{w}_i^T} = \frac{1}{||\mathbf{w}_i^T||}(\mathbf{I} - \frac{\mathbf{w}_i^T\mathbf{w}_i}{||\mathbf{w}_i^T||^2}), \tag{C.5}$$

which results for a given loss $\mathcal{L}$ in

$$\frac{\partial \mathcal{L}}{\partial \mathbf{w}_i^T} = \frac{\partial \hat{\mathbf{w}}_i^T}{\partial \mathbf{w}_i^T} \frac{\partial \mathcal{L}}{\partial \hat{\mathbf{w}}_i^T}. \tag{C.6}$$

## D    Linear ICA

### D.1    Overview

The goal of ICA (Jutten & Herault, 1991; Comon, 1994; Bell & Sejnowski, 1997; Hyvarinen, 1999) is to linearly transform the data into a set of components that are as statistically independent as possible. That is, if $\mathbf{x} \in \mathbb{R}^{1 \times k}$ is a row vector, the goal is to find an unmixing matrix $\mathbf{B} \in \mathbb{R}^{k \times k}$ such that

$$\mathbf{y} = \mathbf{x}\mathbf{B} \tag{D.1}$$

has its components $y_1, ..., y_k$ as independent as possible.

In an alternative equivalent formulation, we assume that the observed $x_1, ..., x_k$ were generated by mixing $k$ independent sources using a mixing matrix $\mathbf{A}$, i.e. we have $\mathbf{x} = \mathbf{s}\mathbf{A}$ with $\mathbf{A} = \mathbf{B}_{(L)}^{-1}$ and $\mathbf{s} = \mathbf{y}$, and the goal is to recover the mixing matrix.

To be able to estimate $\mathbf{B}$, at least $k-1$ of the independent components **must** have non-Gaussian distributions. Otherwise, if the independent components have Gaussian distributions, then the model is **not** identifiable.

Without any other assumptions about how the data was mixed, ICA has two ambiguities: it is not possible to determine the variances nor the order of the independent sources.

There are a few different algorithms for performing ICA (Hyvärinen & Oja, 2000; Bingham et al., 2015), the most popular being FastICA (Hyvarinen, 1999). In the ICA model, the mixing matrix can be arbitrary, meaning that it is not necessarily orthogonal; however, to go about finding it, ICA algorithms generally decompose it into a sequence of transformations that include orthogonal matrices, for there is a guarantee that any matrix has an SVD that encodes the sequence: rotation (orthogonal), scaling (diagonal), rotation (orthogonal). FastICA uses PCA as a preprocessing step to whiten the data and reduce dimensionality. The whitening operation performs the initial rotation and scaling. What remains after that is to find the last rotation that maximizes non-Gaussianity. If $\mathbf{U}\mathbf{S}\mathbf{V}^{\mathbf{T}}$ is the SVD of the covariance matrix, then the whitening transformation is $\mathbf{V}\mathbf{S}^{-1}$, and the resulting overall ICA transformation has the following form

$$\mathbf{B} = \mathbf{V}\mathbf{S}^{-1}\mathbf{W}, \tag{D.2}$$

with $\mathbf{V}$ and $\mathbf{W}$ orthogonal matrices and $\mathbf{S}$ diagonal.

**Ordering of components**    As ICA makes no assumption about the transformation, all the ICs are assumed to have unit variance. And so there is no order implied. Without prior knowledge about how the mixing of the sources occurred, it is impossible to resolve the ICA ambiguity. However, if we have reason to assume that the unmixing matrix is close to orthogonal, or simply has unit norm columns (akin to relaxing the orthogonality constraint in PCA while maintaining unit norm directions) then we can in fact order the components by their variances. In this case, the norm of the columns of the unmixing matrix obtained by FastICA is inversely proportional to the standard deviations of the components. On the other hand, it is also possible to assume that it is the mixing matrix, rather than the unmixing matrix, that has unit norm columns (Hyvärinen, 1999). It is also possible to base the ordering on the measure of non-Gaussianity (Hyvärinen, 1999).

### D.2    Equivariant adaptive separation via independence

Equivariant adaptive separation via independence (EASI) (Cardoso & Laheld, 1996) is a serial updating algorithm for source separation. It combines both a whitening term

$$\mathbf{W}(\mathbf{y}^T\mathbf{y} - \mathbf{I}) \tag{D.3}$$

and a skew-symmetric term

$$\mathbf{W}(\mathbf{y}^T h(\mathbf{y}) - h(\mathbf{y})^T\mathbf{y}) \tag{D.4}$$

to obtain the following global relative gradient update rule

$$\Delta\mathbf{W} = -\eta\mathbf{W}(\mathbf{y}^T\mathbf{y} - \mathbf{I} + \mathbf{y}^T h(\mathbf{y}) - h(\mathbf{y})^T\mathbf{y}), \tag{D.5}$$

with $\eta$ the learning rate. The skew-symmetric term originates from skew-symmetrising $\mathbf{W}h(\mathbf{y})^T\mathbf{y}$ in order to roughly preserve orthogonality with each update. To see why, suppose we have $\mathbf{W}^T\mathbf{W} = \mathbf{I}$, and we modify it into $\mathbf{W} + \mathbf{W}\boldsymbol{\mathcal{E}}$, then we can expand it as

$$(\mathbf{W} + \mathbf{W}\boldsymbol{\mathcal{E}})^T(\mathbf{W} + \mathbf{W}\boldsymbol{\mathcal{E}}) = \mathbf{I} + \boldsymbol{\mathcal{E}}^T + \boldsymbol{\mathcal{E}} + \boldsymbol{\mathcal{E}}^T\boldsymbol{\mathcal{E}}. \tag{D.6}$$

If we want $\mathbf{W} + \mathbf{W}\boldsymbol{\mathcal{E}}$ to remain orthogonal up to first-order, we must also have $(\mathbf{W} + \mathbf{W}\boldsymbol{\mathcal{E}})^T(\mathbf{W} + \mathbf{W}\boldsymbol{\mathcal{E}}) = \mathbf{I} + o(\boldsymbol{\mathcal{E}})$. This implies that $\boldsymbol{\mathcal{E}}$ must be skew-symmetric with $\boldsymbol{\mathcal{E}}^T = -\boldsymbol{\mathcal{E}}$.

### D.3 Non-identifiability of a Gaussian distribution

When the criterion used depends on the variance and/or the independence of components, the only case where it is possible to identify a Gaussian distribution is when the transformation is orthogonal and all the variances are clearly distinct. This means that linear PCA can identify the sources if they have distinct variances and if they were transformed by a rotation. It is impossible to identify in any other case.

Given $\mathbf{y} \sim \mathcal{N}(\mathbf{0}, \boldsymbol{\Sigma}^2)$ and $\mathbf{x} = \mathbf{y}\mathbf{A}$, with $\boldsymbol{\Sigma}^2 = \text{diag}(\boldsymbol{\sigma}^2)$, then, from the affine property of the multivariate Gaussian, we have

$$\mathbf{x} \sim \mathcal{N}(\mathbf{0}, \mathbf{A}^T\boldsymbol{\Sigma}^2\mathbf{A}). \tag{D.7}$$

If $\mathbf{A}$ is orthogonal, then if all the variances are distinct then it is possible to identify up to sign indeterminacy. But if components have equal variances, then they are not possible to identify.

If $\mathbf{A}$ is non-orthogonal, then given the symmetry of $\mathbf{A}^T\boldsymbol{\Sigma}^2\mathbf{A}$, the spectral theorem allows us to write $\mathbf{A}^T\boldsymbol{\Sigma}^2\mathbf{A} = \mathbf{Q}^T\boldsymbol{\Lambda}\mathbf{Q}$, where $\mathbf{Q}$ is orthogonal and $\boldsymbol{\Lambda}$ is diagonal. Therefore, there is no way to recover the non-orthogonal $\mathbf{A}$ even if the sources have distinct variances because it could have been equally transformed by the orthogonal $\mathbf{Q}$.

## E  Probabilistic PCA and Factor Analysis

Probabilistic PCA (pPCA) (Tipping & Bishop, 1999) and Factor Analysis (FA) (Harman, 1976) are two other methods for learning linear transformations. The former is related to linear PCA whereas the latter – in a loose sense – to linear ICA. The primary difference is that, unlike the standard linear PCA and ICA models, the pPCA and FA models include an additional term for modelling the noise.

Without loss of generality, we will assume that the inputs have zero mean. Let $\mathbf{x} \in \mathbb{R}^{1\times p}$ be an input sample and $\mathbf{y} \in \mathbb{R}^{1\times k}$ its corresponding latent. The probabilistic PCA model is of the form

$$\mathbf{x} = \mathbf{y}\mathbf{A} + \lambda\boldsymbol{\epsilon}, \tag{E.1}$$

where $\boldsymbol{\epsilon} \sim \mathcal{N}(\mathbf{0}, \mathbf{I})$ and $\lambda$ is a scalar. The covariance of the input can be expressed as $\mathbf{C} = \frac{1}{n}\mathbb{E}(\mathbf{x}^T\mathbf{x}) = \mathbf{A}^T\mathbf{A} + \lambda^2\mathbf{I}$. The pPCA solution takes the form

$$\mathbf{A} = \boldsymbol{\Sigma}\mathbf{U}^T, \tag{E.2}$$

where $\boldsymbol{\Sigma} = (\max(\mathbf{0}, \mathbf{S} - \lambda^2\mathbf{I}))^{1/2}$, $\mathbf{S}$ is a diagonal matrix of the eigenvalues of $\mathbf{C}$, and $\mathbf{U}$ is an orthogonal matrix of the eigenvectors of $\mathbf{C}$. We can express the latents as a function of the input using

$$\mathbf{B} = \mathbf{A}^{-1} = \mathbf{U}\boldsymbol{\Sigma}^{-1}. \tag{E.3}$$

The solution is thus no different from that of linear PCA, except that the standard deviations are truncated by that of the noise.

FA further generalizes probabilistic PCA by modelling the scale of each component of the noise independently, and so its data model is

$$\mathbf{x} = \mathbf{y}\mathbf{A} + \boldsymbol{\epsilon}\boldsymbol{\Lambda}, \tag{E.4}$$

where $\boldsymbol{\Lambda}$ is a diagonal matrix. FA goes about finding both $\boldsymbol{\Lambda}$ and $\mathbf{A}$ in an iterative fashion (Barber, 2012). This originates from the fact that if we start from an initial guess of $\boldsymbol{\Lambda}$, estimated from the variances of the input data, then we can reduce the FA model to that of pPCA by dividing by $\boldsymbol{\Lambda}$ to obtain

$$\mathbf{x}' = \mathbf{x}\boldsymbol{\Lambda}^{-1} = \mathbf{y}\mathbf{A}\boldsymbol{\Lambda}^{-1} + \boldsymbol{\epsilon}. \tag{E.5}$$

This can now be solved in the same way as pPCA. This process is repeated a number of times until the estimate of the log likelihood of the data, under the Gaussian assumption, no longer changes. The obtained solution ends up being of the form

$$\mathbf{A} = \boldsymbol{\Sigma}\mathbf{U}^T\boldsymbol{\Lambda}, \tag{E.6}$$

where $\boldsymbol{\Sigma} = (\max(\mathbf{0}, \mathbf{S} - \mathbf{I}))^{1/2}$, and $\mathbf{S}$ and $\mathbf{U}$ correspond, respectively, to the eigenvalues and eigenvectors of $\boldsymbol{\Lambda}^{-1}\mathbf{C}\boldsymbol{\Lambda}^{-1}$. This means that, unlike PCA, the matrix $\mathbf{A}$ does not necessarily have orthogonal rows because $\mathbf{A}\mathbf{A}^T = \boldsymbol{\Sigma}\mathbf{V}^T\boldsymbol{\Lambda}^2\mathbf{V}\boldsymbol{\Sigma}$. Some FA methods proceed with doing an additional rotation by multiplying with another orthogonal matrix $\mathbf{V} \in O(k)$ to obtain

$$\mathbf{A} = \mathbf{V}^T\boldsymbol{\Sigma}\mathbf{U}^T\boldsymbol{\Lambda}. \tag{E.7}$$

This means that the overall transformation of from the inputs to the latents is

$$\mathbf{B} = \boldsymbol{\Lambda}^{-1}\mathbf{U}\boldsymbol{\Sigma}^{-1}\mathbf{V}. \tag{E.8}$$

We this see that, similarly to linear ICA, FA estimates two orthogonal transformations, and so the resulting overall transformation is not necessarily orthogonal. Where they differ is in the presence of an additional scaling of the inputs, and notably, in the process of finding the second rotation, linear ICA maximizes independence, whereas FA maximizes variance (Hyvärinen & Oja, 2000).

From the point of view of learning a linear transformation $\mathbf{B}$ such that $\mathbf{y} = \mathbf{x}\mathbf{B}$, we can summarize the models learnt by the different methods in Tab. 1.

| | $\mathbf{B}$ | Criterion |
|---|---|---|
| Linear PCA | $\mathbf{W\Sigma}^{-1}$ | variance |
| Probabilistic PCA | $\mathbf{W\Sigma}^{-1}$ | variance |
| Nonlinear PCA | $\mathbf{W\Sigma}^{-1}$ | variance and independence |
| Linear ICA | $\mathbf{W\Sigma}^{-1}\mathbf{V}$ | variance ($\mathbf{W}$); independence ($\mathbf{V}$) |
| Factor Analysis | $\mathbf{\Lambda}^{-1}\mathbf{W\Sigma}^{-1}\mathbf{V}$ | variance |

Table 1: Linear transformation models learnt by the different methods. Both $\mathbf{W}$ and $\mathbf{V}$ are orthogonal, whereas both $\mathbf{\Lambda}$ and $\mathbf{\Sigma}$ are diagonal. The optimal values found for each of the variables are not necessarily the same. We can note that FA has the most parameters.

# F Nonlinear PCA

## F.1 Stationary point

### F.1.1 Modified derivative

Let us consider the case where $h'(x) = 1$. This approximation does not seem to affect the learned filters when $\mathbf{\Sigma}$ is estimated from the data (see Appendix G.4). We can write our gradient as

$$\frac{\partial \ell}{\partial \mathbf{W}} \approx \mathbf{x}^T(h(\mathbf{y\Sigma}^{-1})\mathbf{\Sigma W}^T\mathbf{W} - \mathbf{y}) = \mathbf{x}^T(\hat{\mathbf{y}} - \mathbf{y}). \tag{F.1}$$

We can take a look at what happens at the stationary point. Taking Eq. F.1, and assuming that $\mathbf{W}^T\mathbf{W} = \mathbf{I}$, we have

$$\Delta\mathbf{W} = \mathbf{x}^T\mathbf{y} - \mathbf{x}^T h(\mathbf{y\Sigma}^{-1})\mathbf{\Sigma W}^T\mathbf{W} \tag{F.2}$$

$$\mathbf{0} = \mathbb{E}(\mathbf{W}^T\mathbf{x}^T\mathbf{y} - \mathbf{W}^T\mathbf{x}^T h(\mathbf{y\Sigma}^{-1})\mathbf{\Sigma W}^T\mathbf{W}) \tag{F.3}$$

$$\mathbf{0} = \mathbf{\Sigma}^2 - \mathbb{E}(\mathbf{y}^T h(\mathbf{y\Sigma}^{-1}))\mathbf{\Sigma W}^T\mathbf{W} \tag{F.4}$$

$$\mathbf{\Sigma}^2 = \mathbb{E}(\mathbf{y}^T h(\mathbf{y\Sigma}^{-1}))\mathbf{\Sigma I} \tag{F.5}$$

$$\mathbf{\Sigma} = \mathbb{E}(\mathbf{y}^T h(\mathbf{y\Sigma}^{-1})). \tag{F.6}$$

Given that $\mathbb{E}(\mathbf{y}^T h(\mathbf{y\Sigma}^{-1}))$ is expected to be diagonal, we end up with $\sigma_i = \mathbb{E}(y_i h(y_i/\sigma_i))$. To be able to satisfy the above, we would need $h$ to adapt such that $\sqrt{\mathrm{var}(h(\mathbf{y\Sigma}^{-1}))} \approx \mathbf{1}$. We can do this by using a function that adjusts the scale, such as $h(x) = a\tanh(x/a)$. For instance, if $x \sim \mathcal{N}(0,1)$, we have $\sqrt{\mathrm{var}(tanh(x))} \approx 0.62$ and $\sqrt{\mathrm{var}(3\tanh(\frac{x}{3}))} \approx 0.90$. So by setting a value of at least $a = 3$, we get closer back to unit variance.

Alternatively, we can consider an asymmetric nonlinear function such as $a\tanh(x)$, where $a = 1/\sqrt{\mathrm{var}(tanh(x))}$. If the standardized $x$ is roughly standard normal, then $a \approx 1.6$.

We also have another option instead of scaling: we can compensate by adding an additional loss term to the encoder contribution as follows:

$$\mathcal{L}(\mathbf{W}) = \mathbb{E}(||\mathbf{x} - h(\mathbf{xW\Sigma}^{-1})\mathbf{\Sigma}[\mathbf{W}^T]_{sg}||_2^2) \tag{F.7}$$

$$+ \alpha\mathbb{E}(||\mathbf{x} - \mathbf{xW}[\mathbf{W}^T]_{sg}||_2^2) \tag{F.8}$$

$$h(x) = \tanh(x) \tag{F.9}$$

$$h'(x) = 1 \tag{F.10}$$

This results in

$$\Delta\mathbf{W} = \alpha(\mathbf{x}^T\mathbf{y} - \mathbf{x}^T\mathbf{y}\mathbf{W}^T\mathbf{W}) + \mathbf{x}^T\mathbf{y} - \mathbf{x}^T h(\mathbf{y}\boldsymbol{\Sigma}^{-1})\boldsymbol{\Sigma}\mathbf{W}^T\mathbf{W} \tag{F.11}$$

$$\mathbf{W}^T\mathbf{W} = (\alpha\boldsymbol{\Sigma}^2 + \boldsymbol{\Sigma}^2)(\alpha\boldsymbol{\Sigma}^2 + \mathbb{E}(\mathbf{y}^T h(\mathbf{y}\boldsymbol{\Sigma}^{-1}))\boldsymbol{\Sigma})^{-1} \tag{F.12}$$

$$= (\mathbf{I} + \alpha^{-1}\mathbf{I})(\mathbf{I} + \alpha^{-1}\mathbb{E}(\mathbf{y}^T h(\mathbf{y}\boldsymbol{\Sigma}^{-1}))\boldsymbol{\Sigma}^{-1})^{-1} \tag{F.13}$$

$$\xrightarrow[\alpha\to\infty]{} \mathbf{I} \tag{F.14}$$

And so if we set $\alpha$ large enough, we can compensate for the scale. In practice $\alpha \geq 2$ is enough to induce an effect.

### F.1.2 Unmodified derivative

If the derivative of $h$ is not modified, and assuming $h_a = a\tanh(x/a)$, $h_a'(x) = 1 - h_a^2(x)/a^2$, the stationary point yields

$$\boldsymbol{\Sigma}^2 = \mathbb{E}(\mathbf{y}^T h_a(\mathbf{y}\boldsymbol{\Sigma}^{-1})\boldsymbol{\Sigma}) - \frac{1}{a}\mathbb{E}((\mathbf{y}^T\mathbf{y} - \mathbf{y}^T h_a(\mathbf{y}\boldsymbol{\Sigma}^{-1})\boldsymbol{\Sigma})h_a^2(\mathbf{y}\boldsymbol{\Sigma}^{-1})). \tag{F.15}$$

We can see that the larger $a$ is, the more the equality will be satisfied. The choice of $a$ will generally depend on the type of distribution, which can be seen more clearly in Fig. 3.

### F.2 Gradient of trainable $\boldsymbol{\Sigma}$

From the loss in Eq. 3.3, we have

$$\frac{\partial\ell}{\partial\boldsymbol{\Sigma}_e} = -\mathrm{diag}(\mathrm{diag}^{-1}(\mathbf{W}_e^T\mathbf{x}^T(\hat{\mathbf{x}} - \mathbf{x})\mathbf{W}_d\boldsymbol{\Sigma}_d \odot h'(\mathbf{z})\boldsymbol{\Sigma}_e^{-2})) \tag{F.16}$$

$$\frac{\partial\ell}{\partial\boldsymbol{\Sigma}_d} = \mathrm{diag}((\hat{\mathbf{x}} - \mathbf{x})\mathbf{W}_d \odot h(\mathbf{z})) \tag{F.17}$$

where diag creates a diagonal matrix from a vector and $\mathrm{diag}^{-1}$ takes the diagonal part of the matrix as vector. If we tie the weights but keep the contributions separate, we obtain

$$\frac{\partial\ell}{\partial\boldsymbol{\Sigma}_e} = -\mathrm{diag}(\mathrm{diag}^{-1}(\mathbf{y}^T(\hat{\mathbf{y}} - \mathbf{y}) \odot h'(\mathbf{y}\boldsymbol{\Sigma}^{-1})\boldsymbol{\Sigma}^{-1})) \tag{F.18}$$

$$\frac{\partial\ell}{\partial\boldsymbol{\Sigma}_d} = \mathrm{diag}((\hat{\mathbf{y}} - \mathbf{y}) \odot h(\mathbf{y}\boldsymbol{\Sigma}^{-1})) \tag{F.19}$$

Given $\boldsymbol{\Sigma} = \mathrm{diag}(\boldsymbol{\sigma})$, we can write

$$\frac{\partial\mathcal{L}}{\partial\boldsymbol{\sigma}_e} = -\mathbf{y}(\hat{\mathbf{y}} - \mathbf{y})h'(\mathbf{y}\boldsymbol{\sigma}^{-1})\boldsymbol{\sigma}^{-1} \tag{F.20}$$

$$\frac{\partial\ell}{\partial\boldsymbol{\sigma}_d} = (\hat{\mathbf{y}} - \mathbf{y})h(\mathbf{y}\boldsymbol{\sigma}^{-1}) \tag{F.21}$$

$$\frac{\partial\ell}{\partial\boldsymbol{\sigma}} = -\mathbf{y}(\hat{\mathbf{y}} - \mathbf{y})h'(\mathbf{y}\boldsymbol{\sigma}^{-1})\boldsymbol{\sigma}^{-1} + (\hat{\mathbf{y}} - \mathbf{y})h(\mathbf{y}\boldsymbol{\sigma}^{-1}) \tag{F.22}$$

Let us consider $h = \tanh$, which results in

$$\frac{\partial \ell}{\partial \boldsymbol{\sigma}} = -\mathbf{y}(\hat{\mathbf{y}} - \mathbf{y})(1 - h^2(\mathbf{y}\boldsymbol{\sigma}^{-1}))\boldsymbol{\sigma}^{-1} + (\hat{\mathbf{y}} - \mathbf{y})h(\mathbf{y}\boldsymbol{\sigma}^{-1}) \tag{F.23}$$

$$= (\hat{\mathbf{y}} - \mathbf{y})(-\mathbf{y}\boldsymbol{\sigma}^{-1} + \mathbf{y}\boldsymbol{\sigma}^{-1}h^2(\mathbf{y}\boldsymbol{\sigma}^{-1}) + h(\mathbf{y}\boldsymbol{\sigma}^{-1})) \tag{F.24}$$

$$= (\hat{\mathbf{y}} - \mathbf{y})(-\mathbf{y}\boldsymbol{\sigma}^{-1} + f(\mathbf{y}\boldsymbol{\sigma}^{-1})) \tag{F.25}$$

$$= \boldsymbol{\sigma}(\mathbf{y}^2\boldsymbol{\sigma}^{-2} - \mathbf{y}\boldsymbol{\sigma}^{-1}f(\mathbf{y}\boldsymbol{\sigma}^{-1}) - \hat{\mathbf{y}}\mathbf{y}\boldsymbol{\sigma}^{-2} + \hat{\mathbf{y}}\boldsymbol{\sigma}^{-1}f(\mathbf{y}\boldsymbol{\sigma}^{-1})) \tag{F.26}$$

$$= \boldsymbol{\sigma}(\mathbf{z}^2 - \mathbf{z}f(\mathbf{z}) - \hat{\mathbf{z}}\mathbf{z} + \hat{\mathbf{z}}f(\mathbf{z})), \tag{F.27}$$

where $f(z) = zh^2(z) + h(z)$, $\mathbf{z} = \mathbf{y}\boldsymbol{\sigma}^{-1}$, and $\hat{\mathbf{z}} = \hat{\mathbf{y}}\boldsymbol{\sigma}^{-1}$. We can plot $f(z)z$, $z^2$ and $f(z)z - z^2$

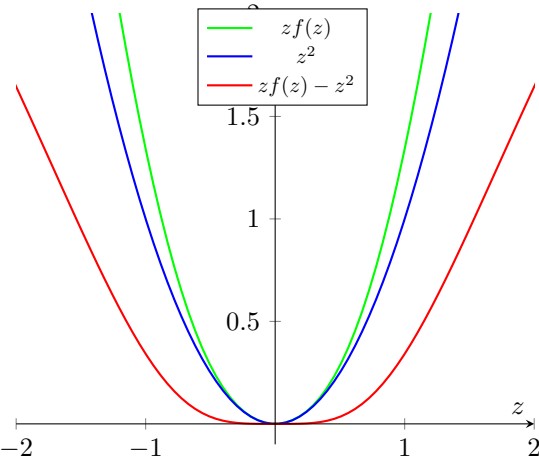

Given that $\mathbf{y}$ is centred, we have $\mathbb{E}(\mathbf{y}^2) = \text{var}(\mathbf{y})$, and so if we take the expectation for the gradient, we will obtain

$$\mathbb{E}(\mathbf{z}^2\boldsymbol{\sigma}) = \mathbb{E}(\mathbf{y}^2\boldsymbol{\sigma}^{-1}) = \text{var}(\mathbf{y})\boldsymbol{\sigma}^{-1}. \tag{F.28}$$

For $|z| < 1$, we also have $zf(z) \approx z^2$. And so we can see that $\frac{\partial \ell}{\partial \boldsymbol{\sigma}}$ takes the form of terms that are roughly $\propto \text{var}(\mathbf{y})\boldsymbol{\sigma}^{-1}$, but there is no guarantee that $\boldsymbol{\Sigma}$ would stop growing unless the reconstruction differences are zero.

### F.3 Reconstruction loss in latent space and relationship to blind deconvolution

When performing the reconstruction in latent space, we need to add something that maintains the orthogonality of $\mathbf{W}$. If we multiply both sides of the difference in Eq. 3.11 by $[\mathbf{W}]_{sg}$ and include a symmetric orthonormality regularizer, we obtain

$$\mathcal{L}(\mathbf{W}, \boldsymbol{\Sigma}) = \mathbb{E}(||\mathbf{x}[\mathbf{W}]_{sg} - h(\frac{\mathbf{x}\mathbf{W}}{\boldsymbol{\sigma}})\boldsymbol{\sigma}[\mathbf{W}^T\mathbf{W}]_{sg}||_2^2) + \beta||\mathbf{I} - \mathbf{W}^T\mathbf{W}||_F^2. \tag{F.29}$$

Although we can assume that $[\mathbf{W}^T\mathbf{W}]_{sg} = \mathbf{I}$ is maintained by an orthonormality regularizer, and might be tempted to simply remove it, we find that keeping it helps make it more stable, especially during the initial period of training where $\mathbf{W}$ might be transitioning via a non-orthogonal matrix. Given that $[\mathbf{W}^T\mathbf{W}]_{sg}$ is square, we can also use its lower triangular form $\triangle[\mathbf{W}^T\mathbf{W}]_{sg}$ to obtain an automatic ordering:

$$\mathcal{L}(\mathbf{W}, \boldsymbol{\Sigma}) = \mathbb{E}(||\mathbf{x}[\mathbf{W}]_{sg} - h(\frac{\mathbf{x}\mathbf{W}}{\boldsymbol{\sigma}})\boldsymbol{\sigma}\triangle[\mathbf{W}^T\mathbf{W}]_{sg}||_2^2) + \beta||\mathbf{I} - \mathbf{W}^T\mathbf{W}||_F^2. \tag{F.30}$$

If indeed we do assume that $[\mathbf{W}^T\mathbf{W}]_{sg} = \mathbf{I}$, we find that it generally requires a much stronger orthonormality regularizer $\beta > 1$, with an asymmetric regularizer being better than the symmetric, and it works better when $\boldsymbol{\Sigma}$ is trainable rather than estimated from data.

$$\mathcal{L}(\mathbf{W}, \boldsymbol{\Sigma}) = \mathbb{E}(||\mathbf{x}[\mathbf{W}]_{sg} - h(\frac{\mathbf{xW}}{\boldsymbol{\sigma}})\boldsymbol{\sigma}||_2^2) + \beta||\mathbf{I} - \mathbf{W}^T\mathbf{W}||_F^2. \tag{F.31}$$

Instead of $\beta||\mathbf{I} - \mathbf{W}^T\mathbf{W}||_F^2$ we can also use the asymmetric version $\beta||\searrow\mathbf{W}^T[\mathbf{W}]_{sg}||_F^2$ (see Appendix B.2).

Alternatively, we can use the linear reconstruction loss $\mathbb{E}(||\mathbf{xW}[\mathbf{W}^T]_{sg} - \mathbf{x}||_2^2)$ for maintaining the orthogonality:

$$\mathcal{L}(\mathbf{W}, \boldsymbol{\Sigma}) = \mathbb{E}(||\mathbf{x}[\mathbf{W}]_{sg} - h(\mathbf{xW}\boldsymbol{\Sigma})\boldsymbol{\Sigma}||_2^2) + \mathbb{E}(||\mathbf{xW}[\mathbf{W}^T]_{sg} - \mathbf{x}||_2^2). \tag{F.32}$$

Putting this in context of blind deconvolution (Lambert, 1996; Haykin, 1996), we can note the first term in Eq. F.29,

$$\mathbb{E}(||h(\frac{\mathbf{y}}{\boldsymbol{\sigma}})\boldsymbol{\sigma} - [\mathbf{y}]_{sg}||_2^2), \tag{F.33}$$

can be seen as a modified form of the Bussgang criterion

$$\mathbb{E}(||h(\mathbf{y}) - \mathbf{y}||_2^2. \tag{F.34}$$

### F.4 Connection with $L_1$ sparsity

ICA is closely related to sparse coding (Olshausen & Field, 1996; Bell & Sejnowski, 1997). This is because maximizing sparsity can be seen as a method for maximizing non-Gaussianity (Hyvärinen & Oja, 2000)– which is a particular ICA method. Reconstruction ICA (RICA) (Le et al., 2011) is a method that combines $L_1$ sparsity with a linear reconstruction loss, and it was initially proposed for learning overcomplete sparse features, in contrast to conventional ICA which does not model overcompleteness. RICA is, in fact, simply performing nonlinear PCA: it induces a latent reconstruction term in the weight update, and, by tying the weights of the encoder and decoder, it enforces $\mathbf{W}$ to have orthogonal columns. RICA is thus a special case of the more general ICA model which can learn an arbitrary transformation rather than just an orthogonal transformation. The RICA loss is

$$\ell(\mathbf{W}) = ||\mathbf{x} - \mathbf{xWW}^T||_2^2 + \beta\sum|\mathbf{xw_j^T}|, \tag{F.35}$$

where $\mathbf{w}_j^T$ is a column of $\mathbf{W}$. This works equally well if we use the subspace variant, i.e.:

$$\ell(\mathbf{W}) = ||\mathbf{x} - \mathbf{x}[\mathbf{W}]_{sg}\mathbf{W}^T||_2^2 + \beta\sum|\mathbf{xw_j^T}|, \tag{F.36}$$

If we compute the gradient contributions from Eq. *F*.36, we obtain

$$\frac{\partial\ell}{\partial\mathbf{W}_e} = \beta\mathbf{x}^T\text{sign}(\mathbf{y}) \tag{F.37}$$

$$\frac{\partial\ell}{\partial\mathbf{W}_d} = (\hat{\mathbf{x}} - \mathbf{x})^T\mathbf{y}, \tag{F.38}$$

which results in the combined gradient

$$\frac{\partial \ell}{\partial \mathbf{W}} = (\hat{\mathbf{x}} - \mathbf{x})^T \mathbf{y} + \beta \mathbf{x}^T \text{sign}(\mathbf{y}) \tag{F.39}$$

$$= \hat{\mathbf{x}}^T \mathbf{y} + \mathbf{x}^T (\beta \text{sign}(\mathbf{y}) - \mathbf{y}). \tag{F.40}$$

The main relevant part in Eq. F.40 is

$$\mathbf{x}^T (\beta \text{sign}(\mathbf{y}) - \mathbf{y}), \tag{F.41}$$

and this is similar to the latent reconstruction term in the nonlinear PCA gradient, except we now have a dependency on the value of $\beta$. No suggestion for the best value of $\beta$ is given by the authors (Le et al., 2011); however, previous works on sparse coding (Olshausen & Field, 1996) have set $\beta$ to be proportional to the standard deviation of the input. We can perhaps gain some insight why by looking more closely at the influence on Eq. F.40 of the norm of the input. For that we extract $||\mathbf{x}||_2$ to get

$$\beta \text{sign}(\mathbf{y}) - \mathbf{y} = ||\mathbf{x}||_2 (\beta \frac{\text{sign}(\mathbf{y})}{||\mathbf{x}||_2} - \frac{\mathbf{x}\mathbf{W}}{||\mathbf{x}||_2}). \tag{F.42}$$

We now see that if $||\mathbf{x}||_2$ increases, then the norm of $\frac{\text{sign}(\mathbf{y})}{||\mathbf{x}||_2}$ will decrease, while that of $\frac{\mathbf{x}\mathbf{W}}{||\mathbf{x}||_2}$ will remain constant. Indeed, we have

$$\frac{||\text{sign}(\mathbf{y})||_2}{||\mathbf{x}||_2} \leq \frac{\sqrt{k}}{||\mathbf{x}||_2} \xrightarrow[||\mathbf{x}||_2 \to \infty]{} 0 \tag{F.43}$$

and

$$\frac{||\mathbf{x}\mathbf{W}||_2}{||\mathbf{x}||_2} \approx 1, \tag{F.44}$$

because, from the reconstruction term, we have

$$||\mathbf{x}\mathbf{W}||_2^2 = \mathbf{x}\mathbf{W}\mathbf{W}^T \mathbf{x}^T = \hat{\mathbf{x}}\mathbf{x}^T \approx \mathbf{x}\mathbf{x}^T = ||\mathbf{x}||_2^2. \tag{F.45}$$

And so to make the relative difference $\beta \text{sign}(\mathbf{y}) - \mathbf{y}$ invariant to the norm of the input, we can set $\beta$ to be proportional to it, or more simply proportional to the standard deviation of the input as it is already centred, i.e. $\beta = \beta_0 \mathbb{E}(||\mathbf{x}||_2)$. As to what the value of $\beta_0$ should be, we can plot the functions $f_{\beta_0}(y) = \beta_0 \text{sign}(y) - y$ and $f_{\beta_0}(y) = \beta_0 \tanh(y) - y$:

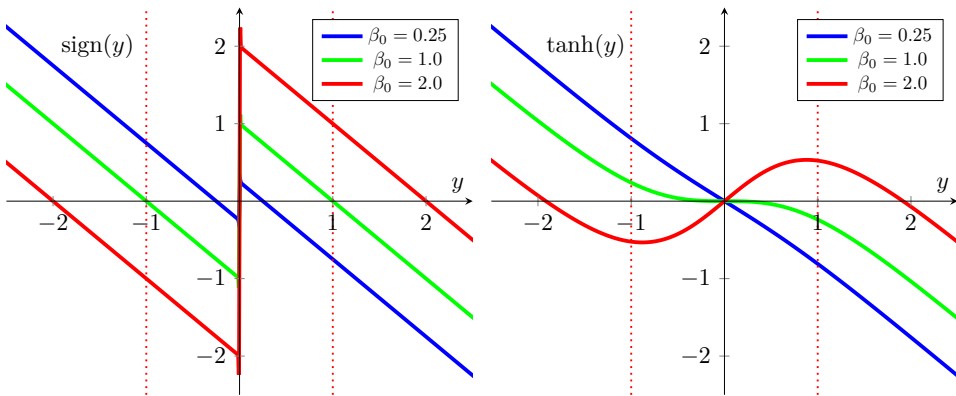

For tanh, if $\beta_0 \leq 1$, then it has a single inflection point, while if $\beta_0 > 1$, then it has two inflection points. A reasonable range for $\beta_0$ appears to be $]0.0, 1.0]$. This range is in line with (Olshausen & Field, 1996) where $\beta_0$ was set to 0.14.

The additional term in the gradient does not maintain unit norm exactly; this is because, if we follow a similar analysis as in Appendix A.3.2, we have for $\Delta \mathbf{W} = \mathbf{0}$

$$\mathbf{W}^T\mathbf{W} = \mathbf{I} - \beta\mathbb{E}(\mathbf{y}^T\text{sign}(\mathbf{y}))\mathbf{\Sigma}^{-2}, \tag{F.46}$$

if using the subspace RICA loss, or

$$\mathbf{W}^T\mathbf{W} = \mathbf{I} - \frac{\beta}{2}\mathbb{E}(\mathbf{y}^T\text{sign}(\mathbf{y}))\mathbf{\Sigma}^{-2}, \tag{F.47}$$

if using the original RICA loss. Therefore, the norm of the columns is less than one.

Another thing we can note is that $\lim_{a\to\infty}\tanh(ax) = \text{sign}(x)$, so in Eq. F.40 we could potentially replace sign with tanh to get

$$\frac{\partial\ell}{\partial\mathbf{W}} = \hat{\mathbf{x}}^T\mathbf{y} + \mathbf{x}^T(\beta\tanh(\mathbf{y}) - \mathbf{y}). \tag{F.48}$$

We know that tanh is the derivative of $\log\cosh$, which is none other than the function used by FastICA for the negentropy approximation (Hyvarinen, 1999). This can be expressed as the following loss function:

$$\mathcal{L}(\mathbf{W}) = ||\mathbf{x} - \mathbf{x}[\mathbf{W}]_{sg}\mathbf{W}^T||_2^2 + \beta\sum\log\cosh(\mathbf{x}\mathbf{w_j^T}). \tag{F.49}$$

### F.5 Skew-symmetric update

Similar to the derivation of the EASI algorithm (see Appendix D.2), we can combine the gradient obtained from a linear reconstruction loss with the term $\mathbf{W}\mathbf{y}^Th(\mathbf{y})$. However, to avoid a similar problem as in the previous section of the gradient update not maintaining unit norm columns, we can simply remove the diagonal part of $\mathbf{y}^Th(\mathbf{y})$, resulting in the weight update

$$\Delta\mathbf{W} \propto \mathbf{x}^T\mathbf{y} - \mathbf{W}\mathbf{y}^T\mathbf{y} - \beta\mathbf{W}(\mathbf{y}^Th(\mathbf{y}) - \text{diag}(h(\mathbf{y})\odot\mathbf{y})). \tag{F.50}$$

Or, similar to EASI, we can use it in its skew-symmetric form:

$$\Delta\mathbf{W} \propto \mathbf{x}^T\mathbf{y} - \mathbf{W}\mathbf{y}^T\mathbf{y} - \beta\mathbf{W}(\mathbf{y}^Th(\mathbf{y}) - h(\mathbf{y})^T\mathbf{y})), \tag{F.51}$$

which we can also write as the loss function

$$\ell(\mathbf{W}) = \frac{1}{2}||\mathbf{x} - \mathbf{x}[\mathbf{W}]_{sg}\mathbf{W}^T||_2^2 + \beta\mathbf{1}\mathbf{W}\odot[\mathbf{W}(\mathbf{y}^Th(\mathbf{y}) - h(\mathbf{y})^T\mathbf{y})))]_{sg}\mathbf{1}^T. \tag{F.52}$$

As justified by the previous section, we can set $\beta = \mathbb{E}(||\mathbf{x}||_2)$, or we can simply use $\mathbf{\Sigma}$ to compensate:

$$\Delta\mathbf{W} \propto \mathbf{x}^T\mathbf{y} - \mathbf{W}\mathbf{y}^T\mathbf{y} - \mathbf{W}(\mathbf{y}^Th(\mathbf{y}\mathbf{\Sigma}^{-1}) - \text{diag}(\mathbf{y}\odot h(\mathbf{y}\mathbf{\Sigma}^{-1})))\mathbf{\Sigma}. \tag{F.53}$$

In all of these variants, we can have $h = \text{sign}$ or $h = \tanh$.

### F.6 Modified decoder contribution

The gradient updates of the encoder and decoder (as derived in 3.1) are

$$\frac{\partial \ell}{\partial \mathbf{W}_e} = \mathbf{x}^T(\hat{\mathbf{y}} - \mathbf{y}) \odot h'(\mathbf{z}) \tag{F.54}$$

$$\frac{\partial \ell}{\partial \mathbf{W}_d} = (\hat{\mathbf{x}} - \mathbf{x})^T h(\mathbf{z})\boldsymbol{\Sigma}. \tag{F.55}$$

The main issue is that, when both are used in the tied case, the decoder contribution overpowers the encoder contribution. This is evident in the linear case, where, when $\mathbf{W}$ is semi-orthogonal, the encoder contribution is zero. Indeed, let $\mathbf{W} \in \mathbb{R}^{p \times k}$ with $p > k$, recall that in the linear case (Appendix A.2.1) we have

$$\frac{\partial \ell}{\partial \mathbf{W}_e} = \mathbf{x}^T\mathbf{y}(\mathbf{W}^T\mathbf{W} - \mathbf{I}) \tag{F.56}$$

$$\frac{\partial \ell}{\partial \mathbf{W}_d} = (\mathbf{W}\mathbf{W}^T - \mathbf{I})\mathbf{x}^T\mathbf{y} \tag{F.57}$$

If we consider their Frobineus norms, we can write

$$||\frac{\partial \ell}{\partial \mathbf{W}_e}||_F = 0 \tag{F.58}$$

$$\lambda_{\min}(\mathbf{x}^T\mathbf{y})\sqrt{p - k} \le ||\frac{\partial \ell}{\partial \mathbf{W}_d}||_F, \tag{F.59}$$

where the lower bound on the decoder contribution can be derived from the fact that for $\mathbf{A} \in \mathbb{R}^{p \times k}$ and $\mathbf{B} \in \mathbb{R}^{k \times n}$, we have (Fang et al., 1994)

$$\lambda_{\min}(\mathbf{A})\|\mathbf{B}\|_F \le \|\mathbf{A}\mathbf{B}\|_F \le \lambda_{\max}(\mathbf{A})\|\mathbf{B}\|_F, \tag{F.60}$$

where $\lambda_{\min}(\mathbf{A})$ and $\lambda_{\max}(\mathbf{A})$ refer to the smallest and largest eigenvalues of $\mathbf{A}$, respectively, and

$$||\mathbf{W}\mathbf{W}^T - \mathbf{I}||_F^2 = tr((\mathbf{W}\mathbf{W}^T - \mathbf{I})^T(\mathbf{W}\mathbf{W}^T - \mathbf{I})) \tag{F.61}$$

$$= tr(\mathbf{W}\mathbf{W}^T\mathbf{W}\mathbf{W}^T - 2\mathbf{W}\mathbf{W}^T + \mathbf{I}) \tag{F.62}$$

$$= tr(\mathbf{I} - \mathbf{W}\mathbf{W}^T) = p - k. \tag{F.63}$$

In the nonlinear case, the gradient updates are approximately close to the linear update (especially if $h(z) = a\tanh(z/a)$ with $a > 3$), except that the encoder contribution is no longer zero, and it is still overpowered by the decoder contribution. And so if omit the latter, we can better see the effect of the former.

An alternative would be to modify the relative scale so that the decoder contribution does not overpower the encoder contribution. The variance of the components plays a role in this, given that this is not a problem in the case of conventional nonlinear PCA with whitened input, where all the components have unit variance. This suggests three options: (1) scale the encoder contribution by $\boldsymbol{\Sigma}$; (2) drop $\boldsymbol{\Sigma}$ from the decoder contribution and, optionally, $h'(\mathbf{z})$ from the encoder contribution; (3) scale down the decoder contribution by a constant, which can simply be the inverse of the spectral, Frobineus, or nuclear norm of $\boldsymbol{\Sigma}$.

For the third option we can write it as a loss:

$$\mathcal{L}(\mathbf{W}) = \mathbb{E}(||\mathbf{x} - h(\mathbf{x}\mathbf{W}\boldsymbol{\Sigma}^{-1})\boldsymbol{\Sigma}[\mathbf{W}^T]_{sg}||_2^2 + \frac{1}{||\boldsymbol{\Sigma}||_2}||\mathbf{x} - [h(\mathbf{x}\mathbf{W}\boldsymbol{\Sigma}^{-1})\boldsymbol{\Sigma}]_{sg}\mathbf{W}^T||_2^2). \tag{F.64}$$

### F.7 Ordering the components based on index position

Here we look at a few ways for ordering the components automatically based on index position.

#### F.7.1 Regularizer

One way to do this is via a Gram-Schmidt-like regularizer (Wang et al., 1995) by adding

$$J(\mathbf{W}) = ||\triangleright(\mathbf{W}^T[\mathbf{W}]_{sg})||_F^2, \tag{F.65}$$

where $\triangleright$ refers to the lower triangular matrix without the diagonal element.

#### F.7.2 Triangular weight update

In the GHA (see Appendix A.3.3), we can order the components by index position by taking the lower (or upper) triangular part of the term $\mathbf{W}\mathbf{y}^T\mathbf{y}$, a term which originates from the linear decoder contribution. Similarly to the GHA, we can also include a triangular term taken from the nonlinear decoder contribution, which is

$$\mathbf{W}(h(\mathbf{z})\boldsymbol{\Sigma})^T h(\mathbf{z})\boldsymbol{\Sigma}, \tag{F.66}$$

where $\mathbf{z} = \mathbf{y}\boldsymbol{\Sigma}^{-1}$. However, as we have seen in Section F.6, we need to scale this term appropriately lest it overpowers the encoder contribution. We can consider variations where we drop $\boldsymbol{\Sigma}$ from either left or right, or make the approximation of $y \approx h(\mathbf{z})\boldsymbol{\Sigma}$:

$$\mathbf{W}\searrow ((h(\mathbf{z})\boldsymbol{\Sigma})^T h(\mathbf{z})) \tag{F.67}$$
$$\text{or } \mathbf{W}\searrow (h(\mathbf{z})^T h(\mathbf{z})\boldsymbol{\Sigma}) \tag{F.68}$$
$$\text{or } \mathbf{W}\searrow (h(\mathbf{z})^T \mathbf{y}) \tag{F.69}$$
$$\text{or } \mathbf{W}\searrow (\mathbf{y}^T h(\mathbf{z})) \tag{F.70}$$
$$\text{or } \mathbf{W}\searrow (\mathbf{z}^T \mathbf{y}) \tag{F.71}$$
$$\text{or } \mathbf{W}\searrow (\mathbf{y}^T \mathbf{z}) \tag{F.72}$$

If $\boldsymbol{\Sigma}$ is non-trainable and $h = \tanh$, then $\boldsymbol{\Sigma} = a\hat{\boldsymbol{\Sigma}}$ where $a \geq 3$ and $\hat{\boldsymbol{\Sigma}}$ is the estimated standard deviation of $\mathbf{y}$.

#### F.7.3 Weighted latent reconstruction

Another option is similar to the weighted subspace algorithm (see A.3.2) where we insert a linearly-spaced $\boldsymbol{\Lambda}$ into the weight update derived in Eq. F.1:

$$\Delta\mathbf{W} \propto \mathbf{x}^T\mathbf{y}\boldsymbol{\Lambda} - \mathbf{x}^T h(\mathbf{y}\boldsymbol{\Sigma}^{-1})\boldsymbol{\Sigma}\boldsymbol{\Lambda}\mathbf{W}^T\mathbf{W}, \tag{F.73}$$

where $\boldsymbol{\Lambda} = \text{diag}(\lambda_1, ..., \lambda_k)$ such that $1 \geq \lambda_1 > ... > \lambda_k > 0$.

As loss function we can have:

$$\mathcal{L}(\mathbf{W}) = \mathbb{E}(||[\mathbf{y}]_{sg}\boldsymbol{\Lambda} - h(\mathbf{y}\boldsymbol{\Sigma}^{-1})\boldsymbol{\Sigma}\boldsymbol{\Lambda}[\mathbf{W}^T\mathbf{W}]_{sg}||_2^2). \tag{F.74}$$

### F.7.4 Embedded projective deflation

Let us consider the loss

$$\ell(\mathbf{W}_e, \mathbf{W}_d, \mathbf{\Sigma}_e, \mathbf{\Sigma}_d) = \frac{1}{2}||\mathbf{x} - h(\mathbf{x}\mathbf{W}_{e1}P(\mathbf{W}_e)\mathbf{\Sigma}_e^{-1})\mathbf{\Sigma}_d\mathbf{W}_d^T||_2^2, \tag{F.75}$$

where

$$P(\mathbf{W}_e) = \alpha\mathbf{I} + \beta\mathbf{W}_{e2}^T\mathbf{W}_{e3}, \tag{F.76}$$

and $\mathbf{W}_e = \mathbf{W}_{e1} = \mathbf{W}_{e2} = \mathbf{W}_{e3}$ for elucidating the contribution of each part to the gradient. Taking the gradient of the loss with respect to the weights, we have

$$\frac{\partial\ell}{\partial\mathbf{W}_{e1}} = \mathbf{x}^T(((\hat{\mathbf{x}} - \mathbf{x})\mathbf{W}_d) \odot h'(\mathbf{y}\mathbf{\Sigma}^{-1}))P(\mathbf{W}_e) \tag{F.77}$$

$$\frac{\partial\ell}{\partial\mathbf{W}_{e2}} = \beta\mathbf{W}_e((\hat{\mathbf{x}} - \mathbf{x})\mathbf{W}_d \odot h'(\mathbf{y}\mathbf{\Sigma}^{-1}))^T\mathbf{y} \tag{F.78}$$

$$\frac{\partial\ell}{\partial\mathbf{W}_{e3}} = \beta\mathbf{W}_e\mathbf{y}^T((\hat{\mathbf{x}} - \mathbf{x})\mathbf{W}_d \odot h'(\mathbf{y}\mathbf{\Sigma}^{-1})), \tag{F.79}$$

Setting $\mathbf{W}_e = \mathbf{W}_d = \mathbf{W}$ and $\mathbf{\Sigma}_e = \mathbf{\Sigma}_d = \mathbf{\Sigma}$, we obtain

$$\frac{\partial\ell}{\partial\mathbf{W}_{e1}} = \mathbf{x}^T((\hat{\mathbf{y}} - \mathbf{y}) \odot h'(\mathbf{y}\mathbf{\Sigma}^{-1}))P(\mathbf{W}) \tag{F.80}$$

$$\frac{\partial\ell}{\partial\mathbf{W}_{e2}} = \beta\mathbf{W}((\hat{\mathbf{y}} - \mathbf{y}) \odot h'(\mathbf{y}\mathbf{\Sigma}^{-1}))^T\mathbf{y} \tag{F.81}$$

$$\frac{\partial\ell}{\partial\mathbf{W}_{e3}} = \beta\mathbf{W}\mathbf{y}^T((\hat{\mathbf{y}} - \mathbf{y}) \odot h'(\mathbf{y}\mathbf{\Sigma}^{-1})) \tag{F.82}$$

Let $\mathbf{y}_\delta = (\hat{\mathbf{y}} - \mathbf{y}) \odot h'(\mathbf{y}\mathbf{\Sigma}^{-1})$, then we can write

$$\frac{\partial\ell}{\partial\mathbf{W}_{e1}} = \mathbf{x}^T\mathbf{y}_\delta P(\mathbf{W}) \tag{F.83}$$

$$\frac{\partial\ell}{\partial\mathbf{W}_{e2}} = \beta\mathbf{W}\mathbf{y}_\delta^T\mathbf{y} \tag{F.84}$$

$$\frac{\partial\ell}{\partial\mathbf{W}_{e3}} = \beta\mathbf{W}\mathbf{y}^T\mathbf{y}_\delta, \tag{F.85}$$

which results in the overall gradient

$$\frac{\partial\ell}{\partial\mathbf{W}_e} = \mathbf{x}^T\mathbf{y}_\delta P(\mathbf{W}) + \beta\mathbf{W}(\mathbf{y}_\delta^T\mathbf{y} + \mathbf{y}^T\mathbf{y}_\delta). \tag{F.86}$$

If we set $P(\mathbf{W}) = \mathbf{I} - \diagdown\mathbf{W}^T[\mathbf{W}]_{sg}$, we obtain

$$\frac{\partial\ell}{\partial\mathbf{W}_e} = \mathbf{x}^T\mathbf{y}_\delta P(\mathbf{W}) - \mathbf{W}\overline{\diagdown}\ (\mathbf{y}_\delta^T\mathbf{y}). \tag{F.87}$$

This introduces an asymmetric term $\mathbf{W}\overline{\diagdown}\ (\mathbf{y}_\delta^T\mathbf{y})$, similar to GHA, resulting in an ordering of components by index position.

### F.7.5 Nested dropout

Similarly to the linear case (Appendix A.3.5), let $\mathbf{m}_{1|j} \in \{0,1\}^k$ be a vector such that $m_i = \begin{cases} 1 & \text{if } i < j \\ 0 & \text{otherwise} \end{cases}$, then we can order components using the loss

$$\mathcal{L}(\mathbf{W}) = \mathbb{E}(||\mathbf{x} - h(\mathbf{x}\mathbf{W}\mathbf{\Sigma}^{-1}) \odot \mathbf{m}_{1|j}\mathbf{\Sigma}[\mathbf{W}^T]_{sg}||_2^2 + ||\mathbf{I} - \mathbf{W}^T\mathbf{W}||_F^2. \tag{F.88}$$

One thing to note about nested dropout is that the larger the index, the less frequently the component receives a gradient update, so training has to run for longer. It can also benefit from the addition of an orthonormality regularizer to provide a gradient signal to the components receiving less frequent updates from the reconstruction.

### F.8 Non-centred nonlinear PCA

When using a zero-centred function like tanh, its input should also be zero-centred, so, to obtain a non-centred version of nonlinear PCA, we can simply subtract the mean, and then add it back after passing through the nonlinear function:

$$\mathcal{L}(\mathbf{W}, \boldsymbol{\sigma}) = \mathbb{E}(||\mathbf{x} - (h(\frac{\mathbf{x}\mathbf{W} - \bar{\boldsymbol{\mu}}_y}{\boldsymbol{\sigma}})\boldsymbol{\sigma} + \bar{\boldsymbol{\mu}}_y)[\mathbf{W}^T]_{sg}||_2^2). \tag{F.89}$$

We are simply standardizing (or normalizing to zero mean and unit variance) the components before the nonlinearity and then undoing the standardization after the nonlinearity. This could also be seen as applying a batch normalization layer (Ioffe & Szegedy, 2015) and then undoing it after the nonlinearity.

We can find an upper bound of the loss by writing

$$\mathcal{L}(\mathbf{W}, \boldsymbol{\sigma}) = \mathbb{E}(||\mathbf{x} - (h(\frac{\mathbf{x}\mathbf{W} - \bar{\boldsymbol{\mu}}_\mathbf{y}}{\boldsymbol{\sigma}})\boldsymbol{\sigma} + \bar{\boldsymbol{\mu}}_\mathbf{y})[\mathbf{W}^T]_{sg}||_2^2) \tag{F.90}$$

$$= \mathbb{E}(||\mathbf{x} - \bar{\boldsymbol{\mu}}_\mathbf{x} + \bar{\boldsymbol{\mu}}_\mathbf{x} - (h(\frac{\mathbf{x}\mathbf{W} - \bar{\boldsymbol{\mu}}_\mathbf{y}}{\boldsymbol{\sigma}})\boldsymbol{\sigma} + \bar{\boldsymbol{\mu}}_\mathbf{y})[\mathbf{W}^T]_{sg}||_2^2) \tag{F.91}$$

$$\leq \mathbb{E}(||\mathbf{x} - \bar{\boldsymbol{\mu}}_\mathbf{x} - (h(\frac{\mathbf{x}\mathbf{W} - \bar{\boldsymbol{\mu}}_\mathbf{y}}{\boldsymbol{\sigma}})\boldsymbol{\sigma})[\mathbf{W}^T]_{sg}||_2^2) + \mathbb{E}(||\bar{\boldsymbol{\mu}}_\mathbf{x} - \bar{\boldsymbol{\mu}}_\mathbf{y}[\mathbf{W}^T]_{sg}||_2^2). \tag{F.92}$$

The upper bound consists of the centred nonlinear PCA loss and a linear mean reconstruction loss. Due to the stop gradient operator, the latter has no effect when $\mathbf{W}$ is orthogonal (see Appendix A.2.1). We can instead change it to

$$\mathbb{E}(||\bar{\boldsymbol{\mu}}_\mathbf{x} - \bar{\boldsymbol{\mu}}_\mathbf{x}\mathbf{W}\mathbf{W}^T||_2^2), \tag{F.93}$$

resulting into the following non-centred loss:

$$\mathcal{L}(\mathbf{W}, \boldsymbol{\sigma}) = \mathbb{E}(||\mathbf{x} - \bar{\boldsymbol{\mu}}_\mathbf{x} - (h(\frac{\mathbf{x}\mathbf{W} - \bar{\boldsymbol{\mu}}_\mathbf{y}}{\boldsymbol{\sigma}})\boldsymbol{\sigma})[\mathbf{W}^T]_{sg}||_2^2) + \mathbb{E}(||\bar{\boldsymbol{\mu}}_\mathbf{x} - \bar{\boldsymbol{\mu}}_\mathbf{x}\mathbf{W}\mathbf{W}^T||_2^2). \tag{F.94}$$

## G  Additional Experiments

### G.1  Nonlinear PCA

#### G.1.1  Highlighting the inability of linear PCA to separate equal variance components

A straightforward way to see how linear PCA is not able to separate components with equal variance is to ZCA-whiten the inputs. This involves transforming the inputs with $\mathbf{W}_L \mathbf{\Sigma}^{-1} \mathbf{W}_L^T$ which effectively transforms all the components into unit-variance components before projecting them back into the input space. Figure 7 shows that linear PCA is not able to relearn the original transformation without a rotational indeterminacy over the entire space; this is not the case with nonlinear PCA.

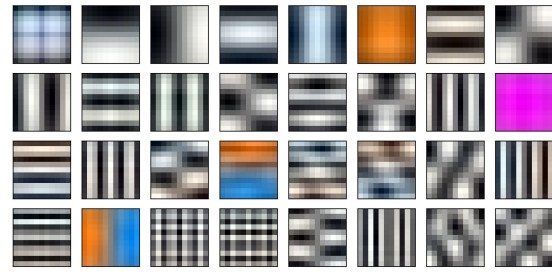

(a) Linear PCA filters on unwhitened data

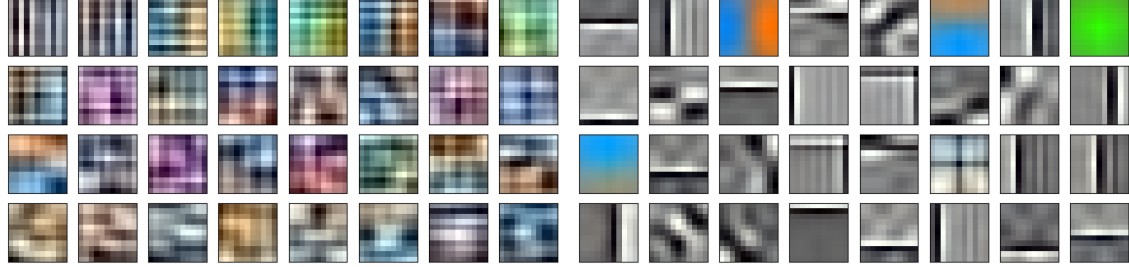

(b) Linear PCA filters on ZCA-whitened data  (c) Nonlinear PCA filters on ZCA-whitened data

Figure 7: PCA filters on ZCA-whitened data. Linear PCA is not able to recover the principal eigenvectors since all the variances are equal.

#### G.1.2  Resolving the linear PCA rotational indeterminacy

Here we attempt to recover the rotational indeterminacy $\mathbf{R}$ from $\mathbf{W}_\mathrm{L} = \mathbf{W}_\mathrm{N} \mathbf{R}$. For that we use gradient descent to recover $\mathbf{R}$ using the loss $||\mathbf{W}_\mathrm{L} - \mathbf{W}_\mathrm{N}\mathbf{R}||_2^2 + ||\mathbf{R}^T \mathbf{R} - \mathbf{I}||$. Figure 8 displays the first 64 linear and nonlinear PCA filters, and Fig. 9 shows the matches for a given linear PCA filter where the corresponding value in $|R_{ij}| > 0.2$.

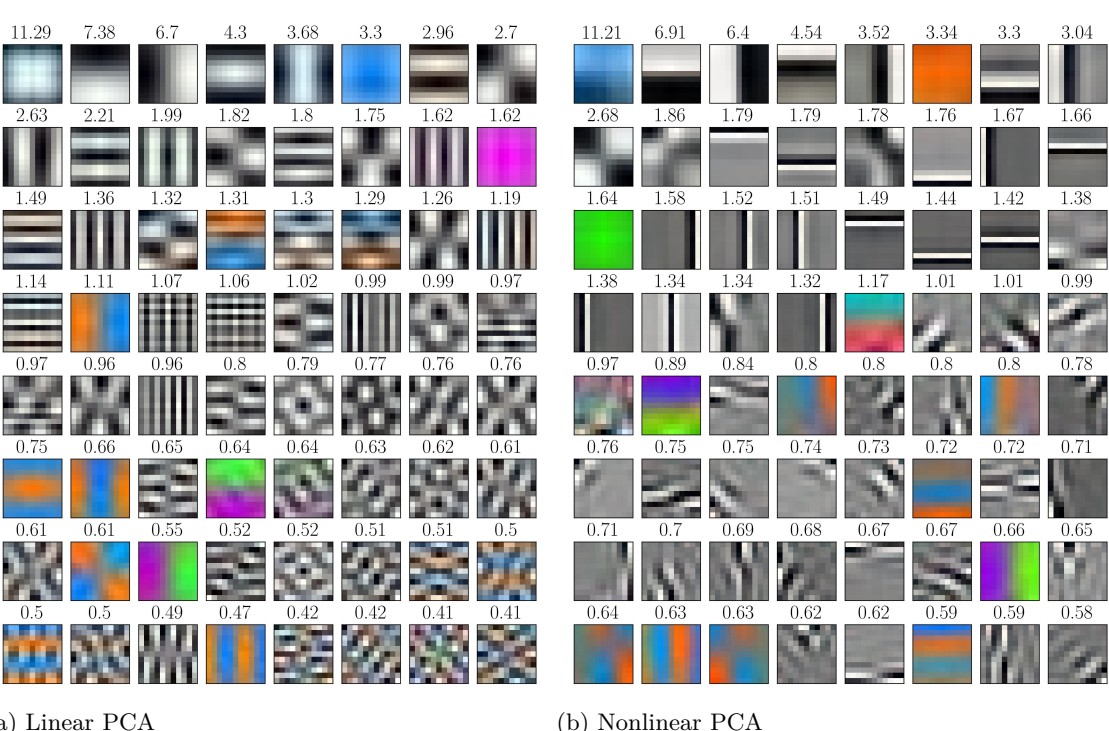

(a) Linear PCA   (b) Nonlinear PCA

Figure 8: PCA filters ordered by their standard deviations

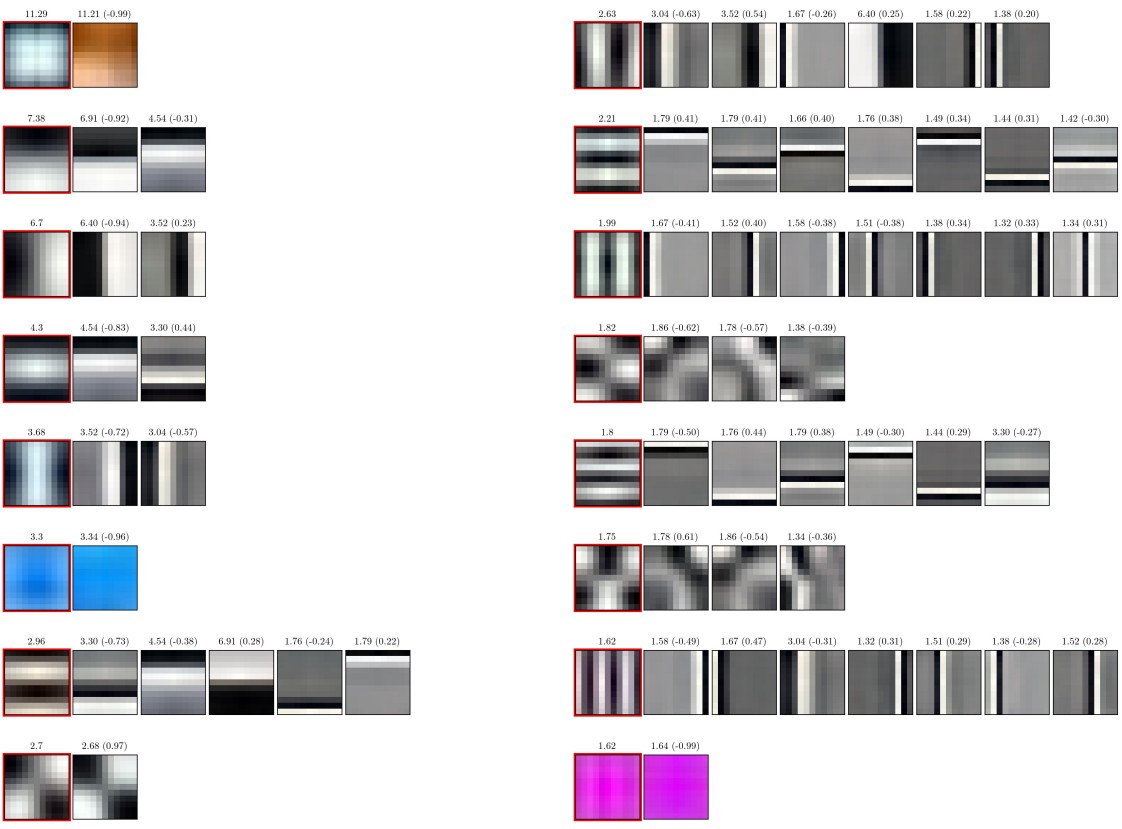

Figure 9: The filters with the red bounding box are the top 16 linear PCA filters. The filters adjacent to each one of the filters in red are the nonlinear PCA filters that could be used as linear combination to generate the linear PCA filters. We can note that the filters generally have similar variances, but there are a few outliers which are undoubtedly due to the unconstrained fitting of the rotational indeterminacy matrix **R**.

### G.1.3 Trainable $\Sigma$

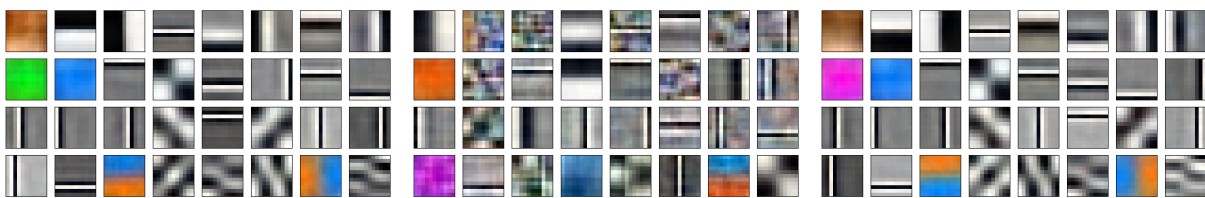

(a) Epoch 3 - without regularization (b) Epoch 10 - without regularization (c) Epoch 10 - with $1e^{-3}$ $L_2$ regularization

Figure 10: Without any regularization, $\sigma$ will tend to keep increasing and result in the degeneration of the filters. This does no occur when using the estimated standard deviations for $\sigma$.

### G.1.4 Scaling of the input

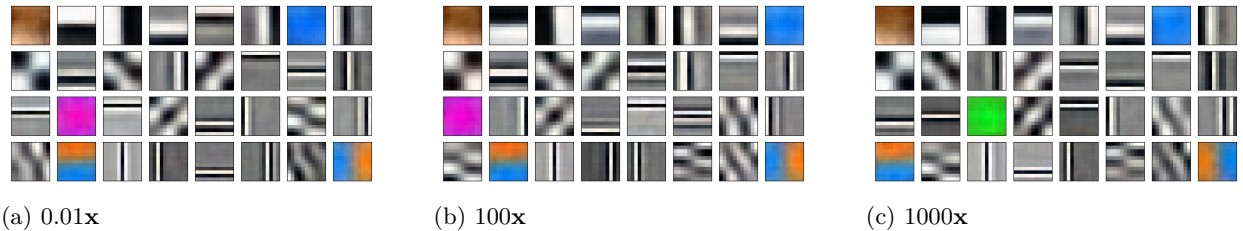

(a) $0.01\mathbf{x}$            (b) $100\mathbf{x}$            (c) $1000\mathbf{x}$

Figure 11: When using nonlinear PCA with estimated standard deviations, it automatically adapts to the scale of the input.

### G.1.5 Scale of tanh

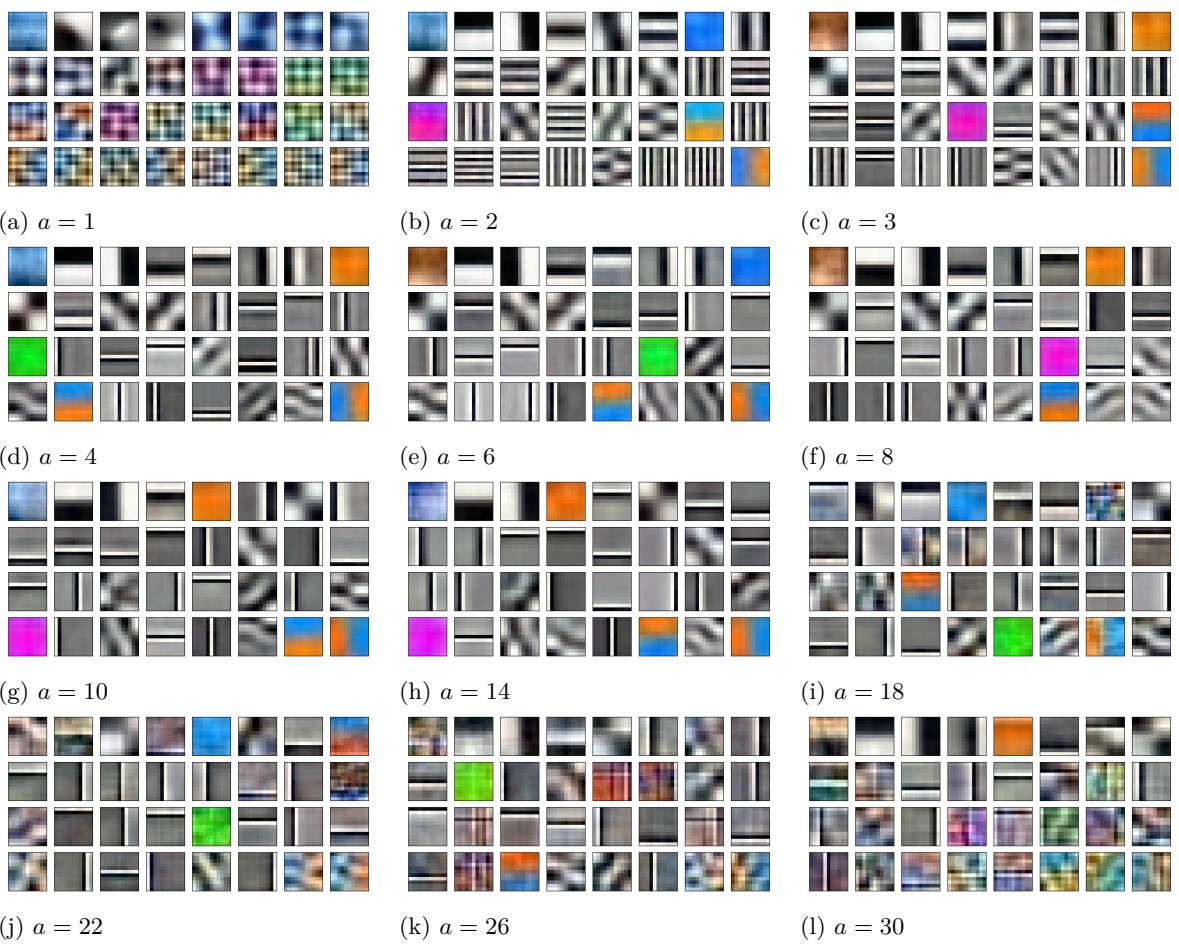

(a) $a = 1$     (b) $a = 2$     (c) $a = 3$

(d) $a = 4$     (e) $a = 6$     (f) $a = 8$

(g) $a = 10$     (h) $a = 14$     (i) $a = 18$

(j) $a = 22$     (k) $a = 26$     (l) $a = 30$

Figure 12: Showing the effect of varying the scalar $a$ in $a \tanh(x/a)$ on the obtained filters. We see that we start getting better defined filters from $a = 2$, but quite a few are still entangled/superimposed. We get better separation from at least $a = 3$. The filters start degenerating $a > 14$.

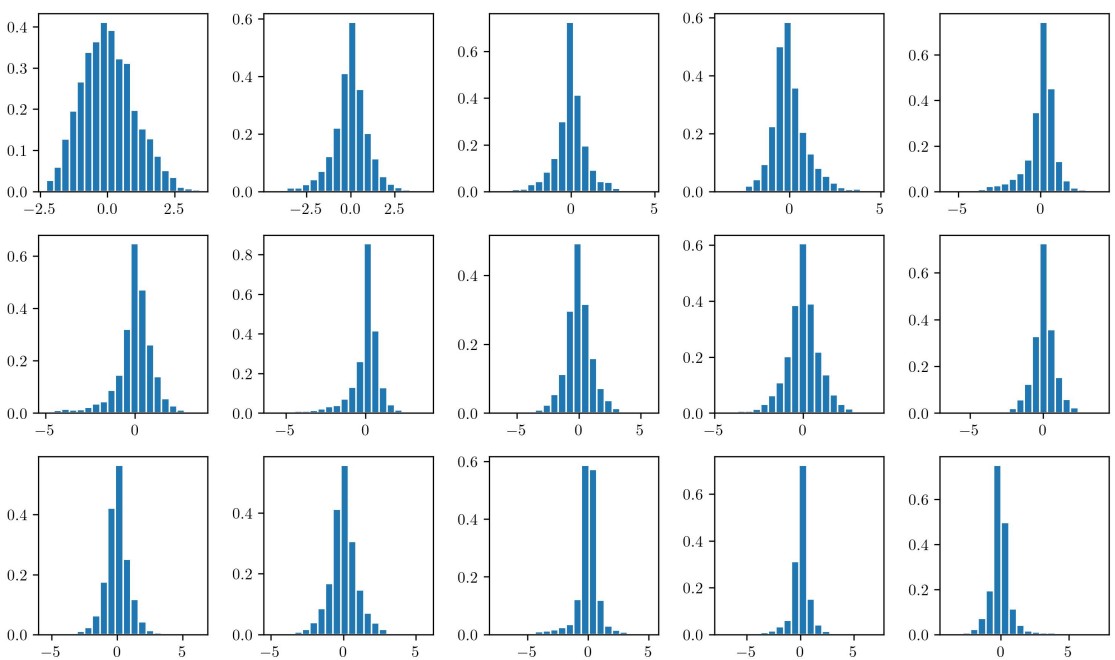

Figure 13: The distributions of the first 15 filters ordered by their variances. We see that they mostly tend to be super-Gaussian distributions, which is why a larger value of $a$ in $a\tanh(x/a)$ was needed.

### G.1.6 Latent space reconstruction

Here we look at the reconstruction loss in latent space and look the filters obtained using the losses described in Section F.3, which we repeat here for convenience:

$$\mathcal{L}(\mathbf{W}, \mathbf{\Sigma}) = \mathbb{E}(||\mathbf{x}[\mathbf{W}]_{sg} - h(\frac{\mathbf{x}\mathbf{W}}{\sigma})\sigma[\mathbf{W}^T\mathbf{W}]_{sg}||_2^2) + \beta||\mathbf{I} - \mathbf{W}^T\mathbf{W}||_F^2 \tag{F.29}$$

$$\mathcal{L}(\mathbf{W}, \mathbf{\Sigma}) = \mathbb{E}(||\mathbf{x}[\mathbf{W}]_{sg} - h(\frac{\mathbf{x}\mathbf{W}}{\sigma})\sigma_\searrow[\mathbf{W}^T\mathbf{W}]_{sg}||_2^2) + \beta||\mathbf{I} - \mathbf{W}^T\mathbf{W}||_F^2 \tag{F.30}$$

$$\mathcal{L}(\mathbf{W}, \mathbf{\Sigma}) = \mathbb{E}(||\mathbf{x}[\mathbf{W}]_{sg} - h(\frac{\mathbf{x}\mathbf{W}}{\sigma})\sigma||_2^2) + \beta||\mathbf{I} - \mathbf{W}^T\mathbf{W}||_F^2. \tag{F.31}$$

$$\mathcal{L}(\mathbf{W}, \mathbf{\Sigma}) = \mathbb{E}(||\mathbf{x}[\mathbf{W}]_{sg} - h(\mathbf{x}\mathbf{W}\mathbf{\Sigma})\mathbf{\Sigma}||_2^2) + \mathbb{E}(||\mathbf{x}\mathbf{W}[\mathbf{W}^T]_{sg} - \mathbf{x}||_2^2). \tag{F.32}$$

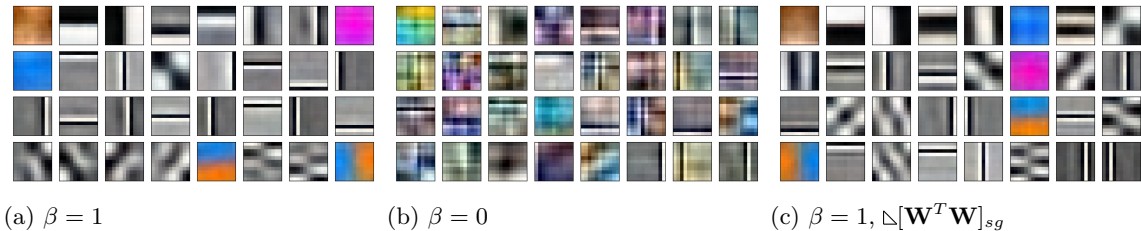

(a) $\beta = 1$        (b) $\beta = 0$        (c) $\beta = 1, \searrow[\mathbf{W}^T\mathbf{W}]_{sg}$

Figure 14: Using Eq. F.29, we see that we do not require a strong gain on the orthonormality regularizer.

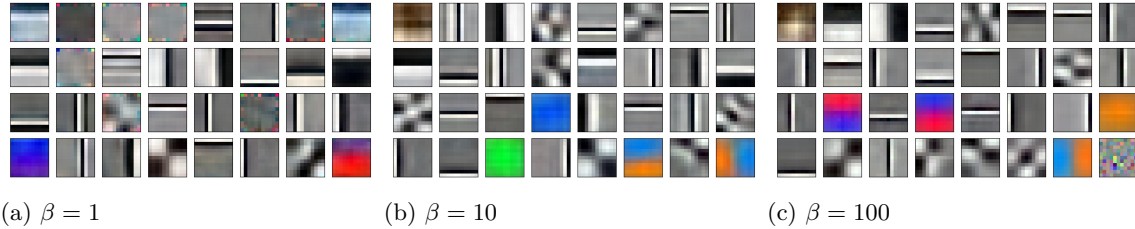

(a) $\beta = 1$        (b) $\beta = 10$        (c) $\beta = 100$

Figure 15: Using Eq. F.31, we see that it benefits from having a larger gain on the orthonormality regularizer.

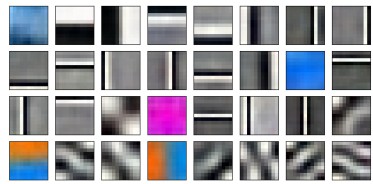

Figure 16: Using Eq. F.32, we see that adding the encoder contribution from the linear reconstruction maintains the orthogonality of the components.

### G.1.7 Asymmetric nonlinear function with modified derivative

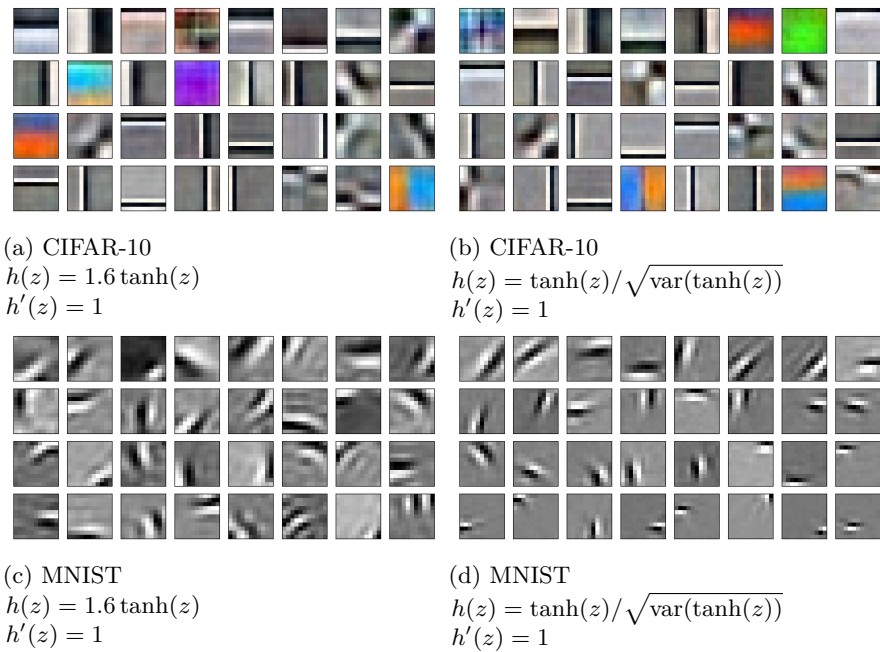

(a) CIFAR-10
$h(z) = 1.6 \tanh(z)$
$h'(z) = 1$

(b) CIFAR-10
$h(z) = \tanh(z)/\sqrt{\mathrm{var}(\tanh(z))}$
$h'(z) = 1$

(c) MNIST
$h(z) = 1.6 \tanh(z)$
$h'(z) = 1$

(d) MNIST
$h(z) = \tanh(z)/\sqrt{\mathrm{var}(\tanh(z))}$
$h'(z) = 1$

Figure 17: Filters obtained on CIFAR-10 and MNIST with an asymmetric activation function where $a$ is either a constant or adaptive.

### G.1.8 First 64 filters

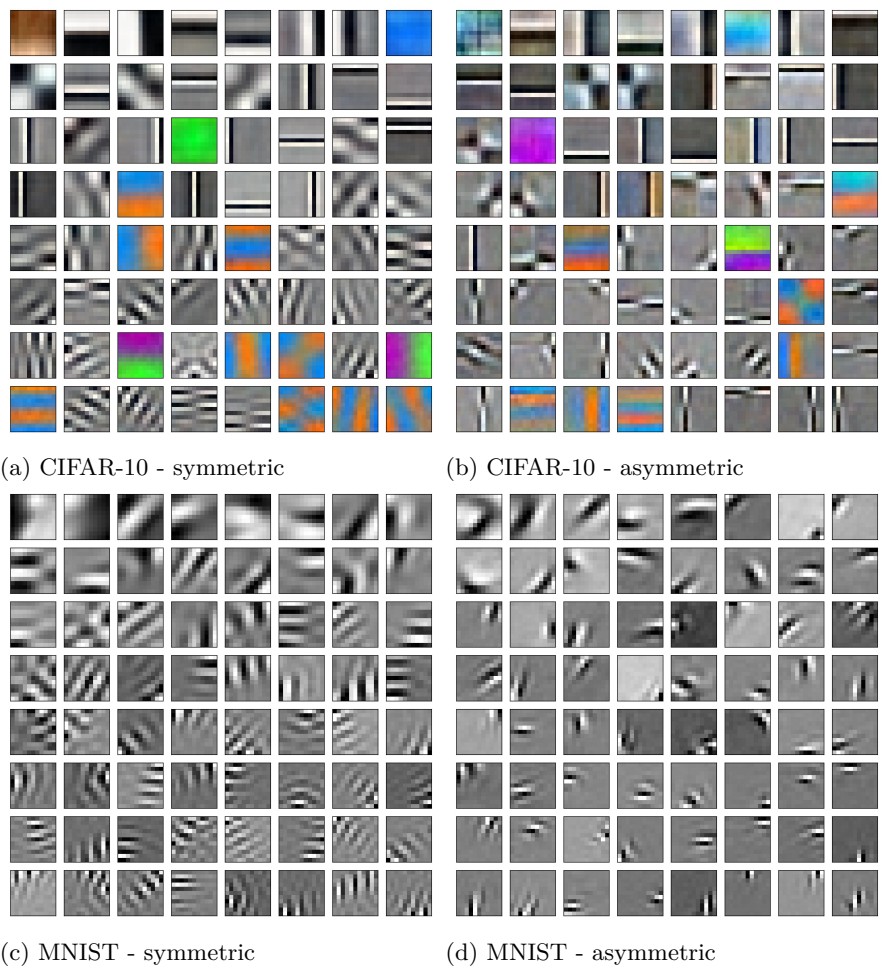

(a) CIFAR-10 - symmetric      (b) CIFAR-10 - asymmetric

(c) MNIST - symmetric      (d) MNIST - asymmetric

Figure 18: The first 64 filters, sorted by variance, obtained on CIFAR-10 and MNIST with either a symmetric activation function ($h(z) = 4\tanh(z/4)$) or an adaptive asymmetric activation function ($h(z) = \tanh(z)/\sqrt{\mathrm{var}(\tanh(z))}, h'(z) = 1$). We can note that the obtained filters seem to be more localized in the asymmetric case compared to the symmetric case.

## G.2   $L_1$ sparsity - RICA

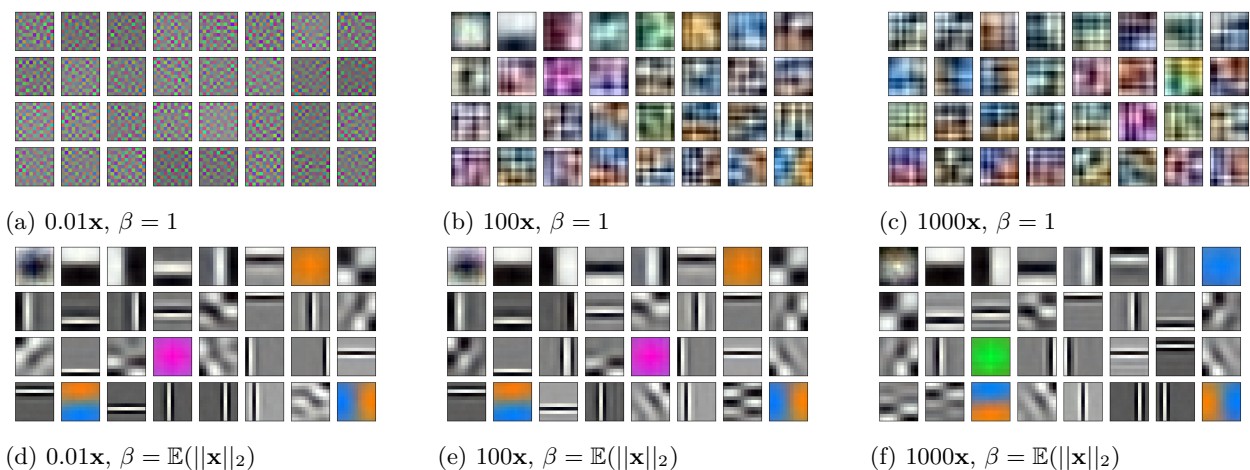

(a) 0.01**x**, $\beta = 1$        (b) 100**x**, $\beta = 1$        (c) 1000**x**, $\beta = 1$

(d) 0.01**x**, $\beta = \mathbb{E}(||\mathbf{x}||_2)$    (e) 100**x**, $\beta = \mathbb{E}(||\mathbf{x}||_2)$    (f) 1000**x**, $\beta = \mathbb{E}(||\mathbf{x}||_2)$

Figure 19: When using RICA F.4, the strength of the sparsity regularizer needs to be adjusted to adapt to the scale of the input. Here we show the effect of the input scale on the obtained filters between adaptive and nonadaptive $\beta$. (a-c) have $\beta = 1$, while (d-f) have $\beta = \mathbb{E}(||\mathbf{x}||_2)$. We see that making $\beta$ proportional to $\mathbb{E}(||\mathbf{x}||_2)$ makes it invariant to the scale of the input.

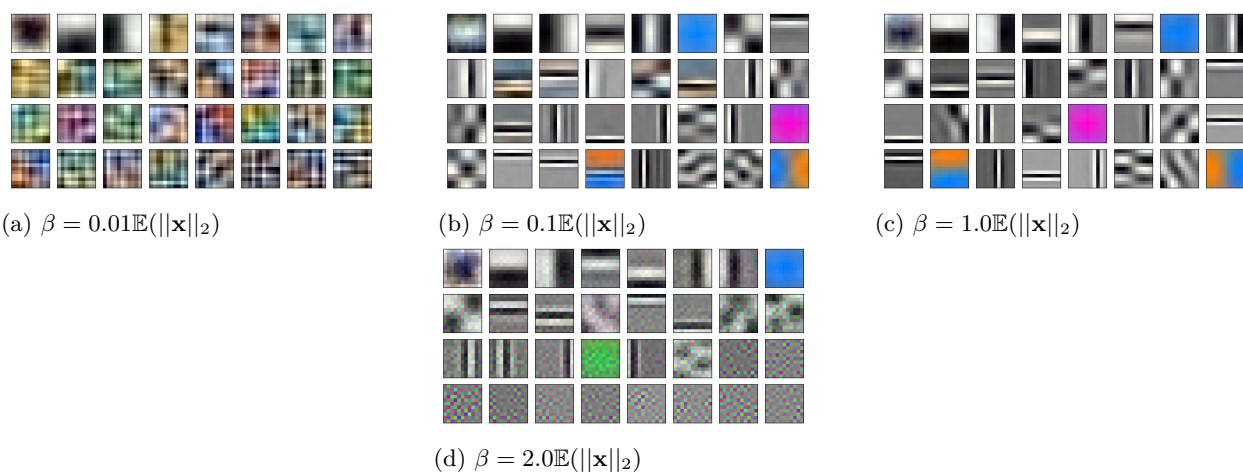

(a) $\beta = 0.01\mathbb{E}(||\mathbf{x}||_2)$     (b) $\beta = 0.1\mathbb{E}(||\mathbf{x}||_2)$     (c) $\beta = 1.0\mathbb{E}(||\mathbf{x}||_2)$

(d) $\beta = 2.0\mathbb{E}(||\mathbf{x}||_2)$

Figure 20: Filters obtained on CIFAR-10 using RICA F.4 with varying $L_1$ sparsity regularization intensity. Unit weight normalization was used (see Appendix C).

### G.3 Linear PCA

Here we summarize variations of linear PCA methods.

| Method | Filters |
| --- | --- |
| 1: PCA via SVD (A.1) | |

2: Linear autoencoder (A.2.1)

$$\mathcal{L}(\mathbf{W}) = \mathbb{E}(||\mathbf{x} - \mathbf{x}\mathbf{W}\mathbf{W}^T||_2^2)$$

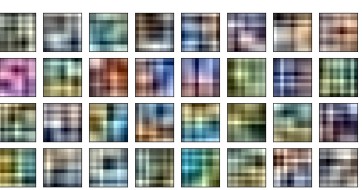

3: Subspace learning algorithm (A.2.2)

$$\Delta\mathbf{W} \propto \mathbf{x}^T\mathbf{y} - \mathbf{W}\mathbf{y}^T\mathbf{y}$$

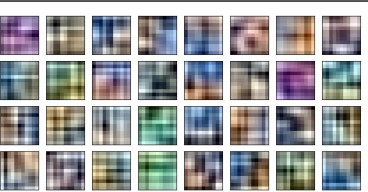

4: Weighted subspace learning algorithm (WSLA) - variant 1 (A.3.1)

$$\Delta\mathbf{W} \propto \mathbf{x}^T\mathbf{y} - \mathbf{W}\mathbf{y}^T\mathbf{y}\mathbf{\Lambda}^{-1}$$
$$1 \geq \lambda_1 > ... > \lambda_k > 0$$

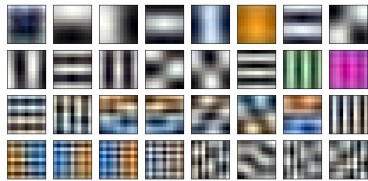

5: WSLA - variant 2 (A.3.1)

$$\Delta\mathbf{W} \propto \mathbf{x}^T\mathbf{y}\mathbf{\Lambda} - \mathbf{W}\mathbf{y}^T\mathbf{y}$$
$$1 \geq \lambda_1 > ... > \lambda_k > 0$$

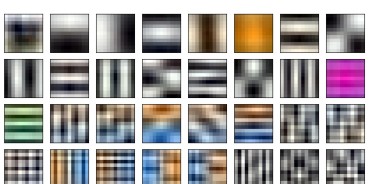

6: WSLA - variant 3 (A.3.1)

$$\Delta\mathbf{W} \propto \mathbf{x}^T\mathbf{y} - \mathbf{W}(\mathbf{y}\mathbf{\Lambda}^{\frac{1}{2}})^T\mathbf{y}\mathbf{\Lambda}^{-\frac{1}{2}}$$
$$1 \geq \lambda_1 > ... > \lambda_k > 0$$

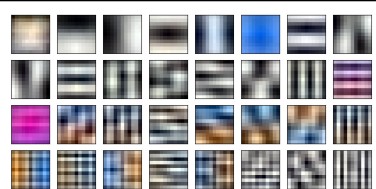

7: Weighted subspace algorithm with unit norm (A.3.2)

$$\mathcal{L}(\mathbf{W}) = \mathbb{E}(||\mathbf{x}[\mathbf{W}]_{sg}\mathbf{W}^T - \mathbf{x}||_2^2$$
$$+ ||\mathbf{W}\hat{\mathbf{\Sigma}}\mathbf{\Lambda}^{\frac{1}{2}}||_2^2 - ||\mathbf{W}\hat{\mathbf{\Sigma}}||_2^2$$
$$- ||\mathbf{x}\mathbf{W}\mathbf{\Lambda}^{\frac{1}{2}}||_2^2 + ||\mathbf{x}\mathbf{W}||_2^2)$$

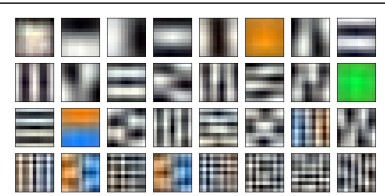

8: Generalized Hebbian algorithm (GHA) (A.3.3)

$$\mathbf{y} = \mathbf{xW}$$
$$\mathbf{\Delta W} \propto \mathbf{x}^T\mathbf{y} - \mathbf{W}_{\vartriangle}(\mathbf{y}^T\mathbf{y})$$

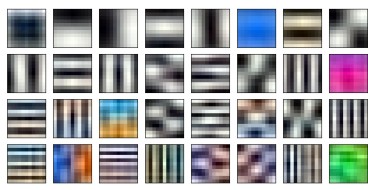

9: GHA with encoder contribution (A.3.4)

$$\Delta \mathbf{W} \propto \quad \mathbf{x}^T\mathbf{y} - \mathbf{W}_{\vartriangle}(\mathbf{y}^T\mathbf{y})$$
$$- \mathbf{x}^T\mathbf{y}(_{\vartriangle}(\mathbf{W}^T\mathbf{W}) - \mathbf{I})$$

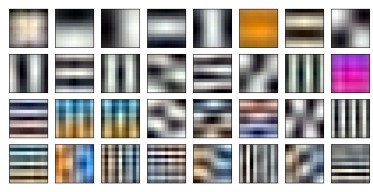

10: GHA + subspace learning algorithm (A.3.4)

$$\Delta \mathbf{W} \propto \quad \mathbf{x}^T\mathbf{y} - \mathbf{W}_{\vartriangle}(\mathbf{y}^T\mathbf{y})$$
$$- \mathbf{W}(\mathbf{y}^T\mathbf{y} - \mathrm{diag}(\mathbf{y}^2))$$

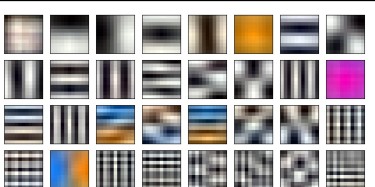

11: Reconstruction + GHA (A.3.4)

$$\mathcal{L}(\mathbf{W}) = \mathbb{E}(||\mathbf{x} - \mathbf{xWW}^T||_2^2$$
$$+ \mathbf{1W} \odot [\mathbf{W}_{\vartriangle}(\mathbf{y}^T\mathbf{y})]_{sg}\mathbf{1}^T)$$

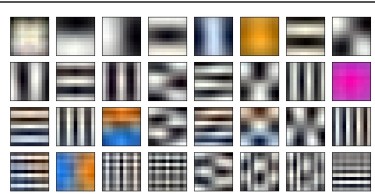

12: Nested dropout (A.3.5)

$$\mathcal{L}(\mathbf{W}) = \mathbb{E}(||(\mathbf{xW} \odot \mathbf{m}_{1|j})\mathbf{W}^T - \mathbf{x}||_2^2)$$

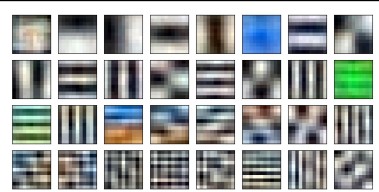

13: Variance maximizer with weighted regularizer (A.3.6)

$$J(\mathbf{W}) = \mathbb{E}(||\mathbf{x} - \mathbf{xWW}^T||_2^2 - ||\mathbf{xW\Lambda}^{\frac{1}{2}}||_2^2)$$
$$1 \geq \lambda_1 > ... > \lambda_k > 0$$

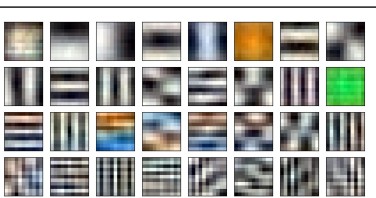

14: Variance maximizer regularizer with nested dropout (A.3.6)

$$\mathcal{L}(\mathbf{W}) = \mathbb{E}(||\mathbf{x} - \mathbf{x}\mathbf{W}\mathbf{W}^T||_2^2$$
$$- ||\mathbf{x}\mathbf{W} \cdot \mathbf{m}_{1|j}||_2^2)$$

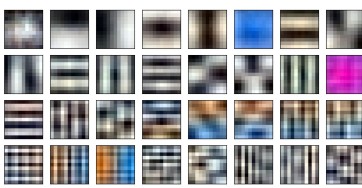

## G.4 Nonlinear PCA

Here we summarize variations of nonlinear PCA methods.

| Method | Filters |
|---|---|

1: Conventional without whitening (2.6)

$$\mathcal{L}(\mathbf{W}) = \mathbb{E}(||\mathbf{x} - h(\mathbf{x}\mathbf{W})\mathbf{W}^T||_2^2)$$

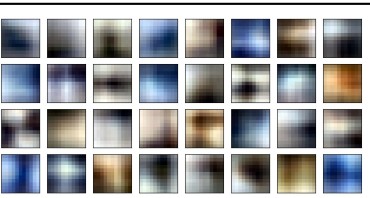

2: Trainable $\boldsymbol{\sigma}$ - without stop gradient (3)

$$\mathcal{L}(\mathbf{W}, \boldsymbol{\Sigma}) = \mathbb{E}(||\mathbf{x} - h(\mathbf{x}\mathbf{W}\boldsymbol{\Sigma}^{-1})\boldsymbol{\Sigma}\mathbf{W}^T||_2^2)$$

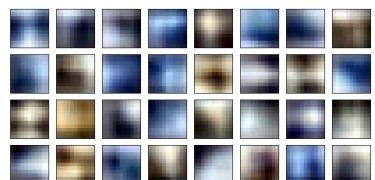

3: Trainable $\boldsymbol{\sigma}$ (Eq. 3.11)

$$\mathcal{L}(\mathbf{W}, \boldsymbol{\Sigma}) = \mathbb{E}(||\mathbf{x} - h(\mathbf{x}\mathbf{W}\boldsymbol{\Sigma}^{-1})\boldsymbol{\Sigma}[\mathbf{W}^T]_{sg}||_2^2)$$

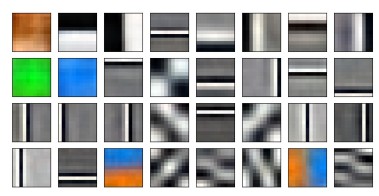

4: Trainable $\boldsymbol{\sigma}$ - latent reconstruction with reconstruction orthogonal regularizer (F.3)

$$\mathcal{L}(\mathbf{W}, \boldsymbol{\Sigma}) = \mathbb{E}(||\mathbf{x}[\mathbf{W}]_{sg} - h(\mathbf{x}\mathbf{W}\boldsymbol{\Sigma})\boldsymbol{\Sigma}||_2^2$$
$$+ \mathbb{E}(||\mathbf{x}\mathbf{W}[\mathbf{W}^T]_{sg} - \mathbf{x}||_2^2)$$

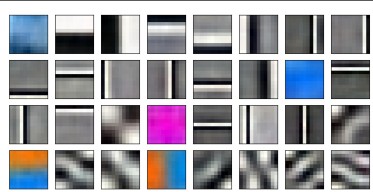

5: Trainable $\boldsymbol{\sigma}$ - latent reconstruction with symmetric orthogonal regularizer (F.3)

$$\mathcal{L}(\mathbf{W}, \boldsymbol{\Sigma}) = \mathbb{E}(||\mathbf{x}[\mathbf{W}]_{sg} - h(\mathbf{x}\mathbf{W}\boldsymbol{\Sigma})\boldsymbol{\Sigma}||_2^2)$$
$$+ \beta||\mathbf{I} - \mathbf{W}^T\mathbf{W}||_F^2$$
$$\beta = 10$$

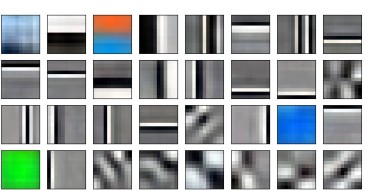

6: Trainable $\boldsymbol{\sigma}$ - latent reconstruction with asymmetric orthogonal regularizer (F.3)

$$\mathcal{L}(\mathbf{W}, \boldsymbol{\Sigma}) = \mathbb{E}(||\mathbf{x}[\mathbf{W}]_{sg} - h(\mathbf{x}\mathbf{W}\boldsymbol{\Sigma})\boldsymbol{\Sigma}||_2^2)$$
$$+ \beta||(\boldsymbol{\diagdown}(\mathbf{W}^T[\mathbf{W}]_{sg}))||_F^2$$
$$\beta = 10$$

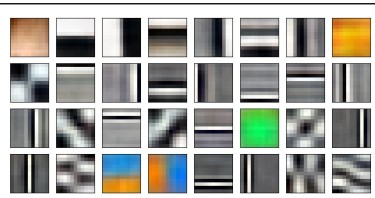

7: Trainable $\boldsymbol{\sigma}$ - latent reconstruction with $[\mathbf{W}^T\mathbf{W}]_{sg}$ and symmetric orthogonal regularizer (F.3)

$$\mathcal{L}(\mathbf{W}, \boldsymbol{\Sigma}) = \mathbb{E}(||\mathbf{x}[\mathbf{W}]_{sg} - h(\frac{\mathbf{x}\mathbf{W}}{\boldsymbol{\sigma}})\boldsymbol{\sigma}[\mathbf{W}^T\mathbf{W}]_{sg}||_2^2)$$
$$+ ||\mathbf{I} - \mathbf{W}^T\mathbf{W}||_F^2$$

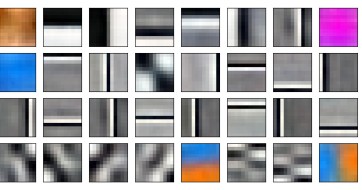

8: EMA $\boldsymbol{\sigma}$ - tanh (3)

$$\mathcal{L}(\mathbf{W}) = \mathbb{E}(||\mathbf{x} - h(\mathbf{x}\mathbf{W}\boldsymbol{\Sigma}^{-1})\boldsymbol{\Sigma}[\mathbf{W}^T]_{sg}||_2^2)$$
$$h(x) = \tanh(x)$$

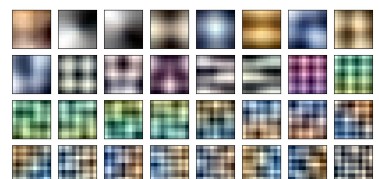

9: EMA $\boldsymbol{\sigma}$ - scaled tanh (3.4)

$$\mathcal{L}(\mathbf{W}) = \mathbb{E}(||\mathbf{x} - h(\mathbf{x}\mathbf{W}\boldsymbol{\Sigma}^{-1})\boldsymbol{\Sigma}[\mathbf{W}^T]_{sg}||_2^2)$$
$$h(x) = 4\tanh(\frac{x}{4})$$

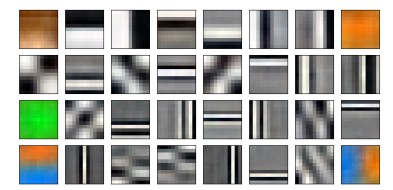

10: EMA $\boldsymbol{\sigma}$ - scaled tanh with modified derivative (Eq. F.1)

$$\mathcal{L}(\mathbf{W}) = \mathbb{E}(||\mathbf{x} - h(\mathbf{x}\mathbf{W}\boldsymbol{\Sigma}^{-1})\boldsymbol{\Sigma}[\mathbf{W}^T]_{sg}||_2^2)$$
$$h(x) = 4\tanh(\frac{x}{4})$$
$$h'(x) = 1$$

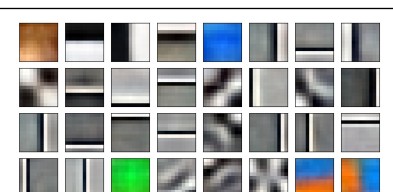

11: EMA $\boldsymbol{\sigma}$ - with hard tanh approximation (3.4)

$$\mathcal{L}(\mathbf{W}) = \mathbb{E}(||\mathbf{x} - h(\mathbf{xW\Sigma}^{-1})\mathbf{\Sigma}[\mathbf{W}^T]_{sg}||_2^2)$$
$$h(x) = max(-2, min(2, 0))$$
$$h'(x) = 1$$

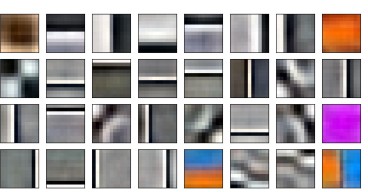

12: Asymmetric activation - adaptive (F.1.1)

$$\mathcal{L}(\mathbf{W}) = \mathbb{E}(||\mathbf{x} - h(\mathbf{xW\Sigma}^{-1})\mathbf{\Sigma}[\mathbf{W}^T]_{sg}||_2^2)$$
$$h(\mathbf{x}) = \mathbf{a}\tanh(\mathbf{x})$$
$$h'(x) = 1$$
$$\mathbf{a} = 1/[\sqrt{\text{var}(h(\mathbf{z}))}]_{sg}$$

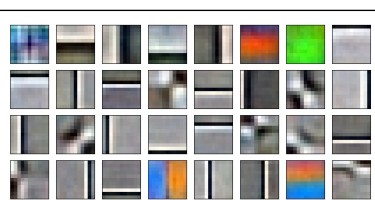

13: Asymmetric activation - constant (F.1.1)

$$\mathcal{L}(\mathbf{W}) = \mathbb{E}(||\mathbf{x} - h(\mathbf{xW\Sigma}^{-1})\mathbf{\Sigma}[\mathbf{W}^T]_{sg}||_2^2)$$
$$h(\mathbf{x}) = 1.6\tanh(\mathbf{x})$$
$$h'(x) = 1$$

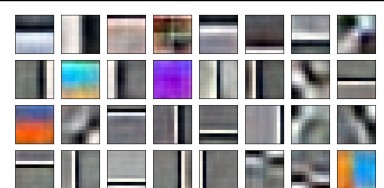

14: EMA $\boldsymbol{\sigma}$ - latent reconstruction with symmetric orthogonal regularizer - scaled tanh (F.3)

$$\mathcal{L}(\mathbf{W}, \mathbf{\Sigma}) = \mathbb{E}(||\mathbf{x}[\mathbf{W}]_{sg} - h(\mathbf{xW\Sigma})\mathbf{\Sigma}[\mathbf{W}^T\mathbf{W}]_{sg}||_2^2)$$
$$+ ||\mathbf{I} - \mathbf{W}^T\mathbf{W}||_F^2$$
$$h(x) = 4\tanh(\frac{x}{4})$$

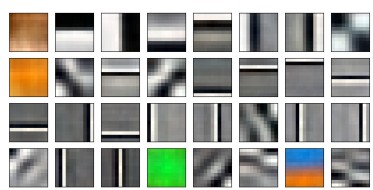

15: EMA $\boldsymbol{\sigma}$ - trainable scaled tanh (3.4)

$$\mathcal{L}(\mathbf{W}, a) = \mathbb{E}(||\mathbf{x} - h_a(\mathbf{xW\Sigma}^{-1})\mathbf{\Sigma}[\mathbf{W}^T]_{sg}||_2^2)$$
$$h_a(x) = a\tanh(\frac{x}{a})$$

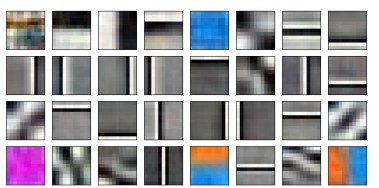

16: EMA $\boldsymbol{\sigma}$ - baked-in GS. No need to order the components as it is done automatically. (3.6)

$$\mathcal{L}(\mathbf{W}) = \mathbb{E}(||\mathbf{x} - h(\mathbf{xW}P(\mathbf{W})\mathbf{\Sigma}^{-1})\mathbf{\Sigma}[\mathbf{W}^T]_{sg}||_2^2)$$
$$P(\mathbf{W}) = (I - \unlhd(\mathbf{W}^T[\mathbf{W}]_{sg})$$

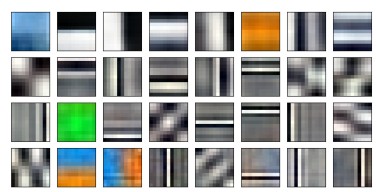

17: EMA $\boldsymbol{\sigma}$ - scaled tanh with nested dropout (F.7.5)

$$\mathcal{L}(\mathbf{W}) = \mathbb{E}(||\mathbf{x} - h(\mathbf{x}\mathbf{W}\boldsymbol{\Sigma}^{-1}) \odot \mathbf{m}_{1|j}\boldsymbol{\Sigma}[\mathbf{W}^T]_{sg}||_2^2)$$
$$+ ||\mathbf{W}^T\mathbf{W} - \mathbf{I}||_F^2$$
$$h(x) = 4\tanh(\frac{x}{4})$$

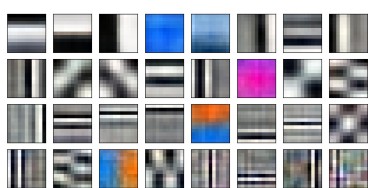

18: EMA $\boldsymbol{\sigma}$ - modified tanh derivative - with linear reconstruction (F.1.1)

$$\mathcal{L}(\mathbf{W}) = \mathbb{E}(||\mathbf{x} - h(\mathbf{x}\mathbf{W}\boldsymbol{\Sigma}^{-1})\boldsymbol{\Sigma}[\mathbf{W}^T]_{sg}||_2^2)$$
$$+ \alpha\mathbb{E}(||\mathbf{x} - \mathbf{x}\mathbf{W}[\mathbf{W}^T]_{sg}||_2^2)$$
$$h(x) = \tanh(x)$$
$$h'(x) = 1$$
$$\alpha \geq 2$$

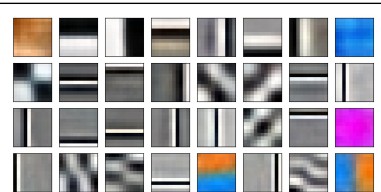

19: RICA (F.4)

$$\mathcal{L}(\mathbf{W}) = \mathbb{E}(||\mathbf{x} - \mathbf{x}\mathbf{W}\mathbf{W}^T||_2^2 + \beta\sum|\mathbf{x}\mathbf{w_j^T}|)$$
$$\beta = \mathbb{E}(||\mathbf{x}||_2)$$

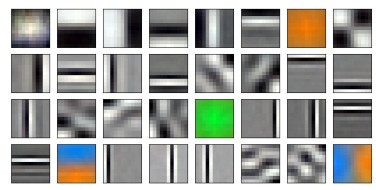

20: Reconstruction with log cosh regularizer (F.4)

$$\mathcal{L}(\mathbf{W}) = \mathbb{E}(||\mathbf{x} - \mathbf{x}\mathbf{W}\mathbf{W}^T||_2^2 + \beta\sum\log\cosh(\mathbf{x}\mathbf{w_j^T}))$$

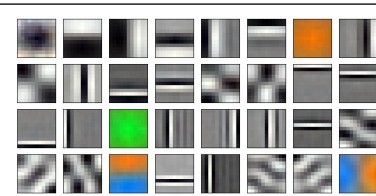

21: Reconstruction skew symmetric $\mathbf{y}^T h(\mathbf{y})$ (F.5)

$$\mathcal{L}(\mathbf{W}) = \mathbb{E}(||\mathbf{x} - \mathbf{x}\mathbf{W}\mathbf{W}^T||_2^2$$
$$+ \beta\mathbf{1}\mathbf{W} \odot [\mathbf{W}(\mathbf{y}^T h(\mathbf{y}) - h(\mathbf{y})^T\mathbf{y}))]_{sg}\mathbf{1}^T)$$
$$h = \tanh$$
$$\beta = \mathbb{E}(||\mathbf{x}||_2)$$

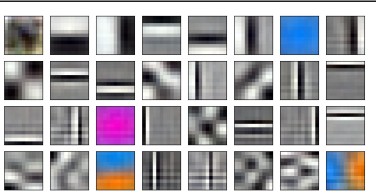

22: Reconstruction with symmetric $\mathbf{y}^T h(\mathbf{y})$ without diagonal (F.5)

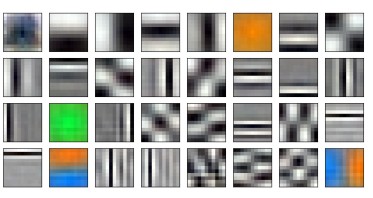

$$\mathcal{L}(\mathbf{W}) = \mathbb{E}(||\mathbf{x} - \mathbf{x}\mathbf{W}\mathbf{W}^T||_2^2$$
$$+ \beta \mathbf{1}\mathbf{W} \odot [\mathbf{W}(\mathbf{y}^T h(\mathbf{y}) - \mathrm{diag}(h(\mathbf{y}) \odot \mathbf{y})))]_{sg}\mathbf{1}^T)$$
$$h = \tanh$$
$$\beta = \mathbb{E}(||\mathbf{x}||_2)$$

23: Reconstruction with symmetric $\mathbf{y}^T h(\mathbf{y}/\boldsymbol{\sigma})\boldsymbol{\sigma}$ without diagonal (F.5)

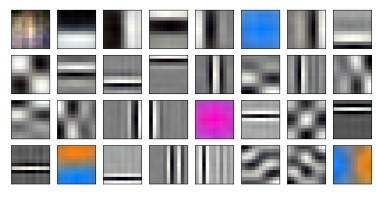

$$\mathcal{L}(\mathbf{W}) = \mathbb{E}(||\mathbf{x} - \mathbf{x}\mathbf{W}\mathbf{W}^T||_2^2$$
$$+ \mathbf{1}\mathbf{W} \odot [\mathbf{W}(\mathbf{y}^T h(\mathbf{y}\boldsymbol{\Sigma}^{-1})\boldsymbol{\Sigma}$$
$$- \mathrm{diag}(h(\mathbf{y}\boldsymbol{\Sigma}^{-1})\boldsymbol{\Sigma} \odot \mathbf{y})))]_{sg}\mathbf{1}^T)$$
$$h = \mathrm{sign}$$

24: EMA $\boldsymbol{\sigma}$ - scaled tanh with triangular update (F.7.2)

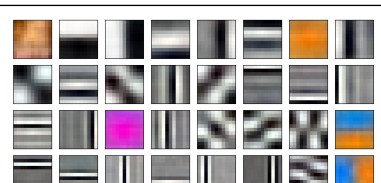

$$\mathcal{L}(\mathbf{W}) = \mathbb{E}(||\mathbf{x} - h(\mathbf{x}\mathbf{W}\boldsymbol{\Sigma}^{-1})\boldsymbol{\Sigma}[\mathbf{W}^T]_{sg}||_2^2$$
$$+ \mathbf{1}\mathbf{W} \odot [\mathbf{W} \searrow (\mathbf{y}^T \mathbf{y}\boldsymbol{\Sigma}^{-1})]_{sg}\mathbf{1}^T)$$
$$h(x) = 6\tanh(\frac{x}{6})$$

## G.5  Time Series

### G.5.1  Orthogonal - same variance

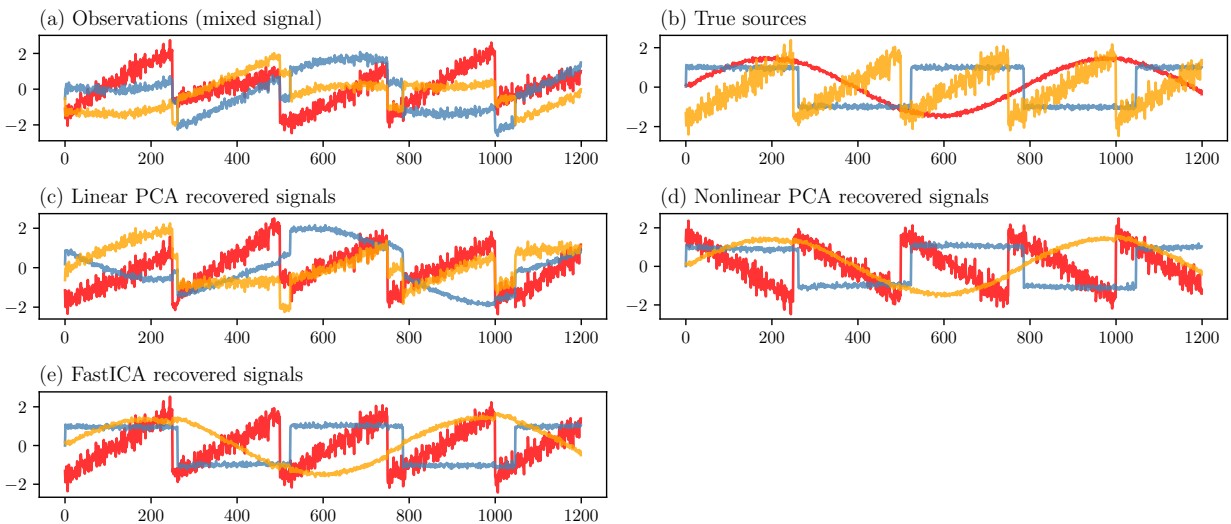

(a) Observations (mixed signal)

(b) True sources

(c) Linear PCA recovered signals

(d) Nonlinear PCA recovered signals

(e) FastICA recovered signals

Figure 21: Three signals (sinusoidal, square, and sawtooth) that were mixed with an **orthogonal** mixing matrix. Linear PCA was unable to separate the signals as they all had the same variance.

### G.5.2   Orthogonal - all distinct variances

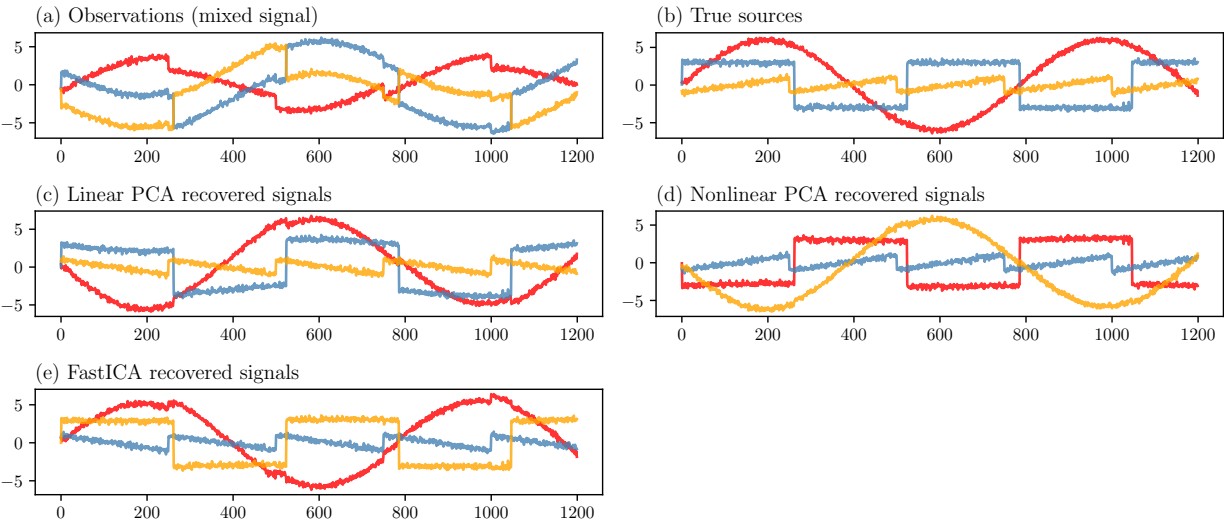

Figure 22: Three signals (sinusoidal, square, and sawtooth) with distinct variances that were mixed with an **orthogonal** mixing matrix. All three managed to separate the signals.

### G.6   Sub- and Super- Gaussian 2D points

Here we look at the effect of applying linear PCA, nonlinear PCA, and linear ICA on 2D points ($n = 1000$) sampled from either a uniform distribution (sub-Gaussian) or a Laplace distribution (super-Gaussian). Figures 23 and 24 summarize the results.

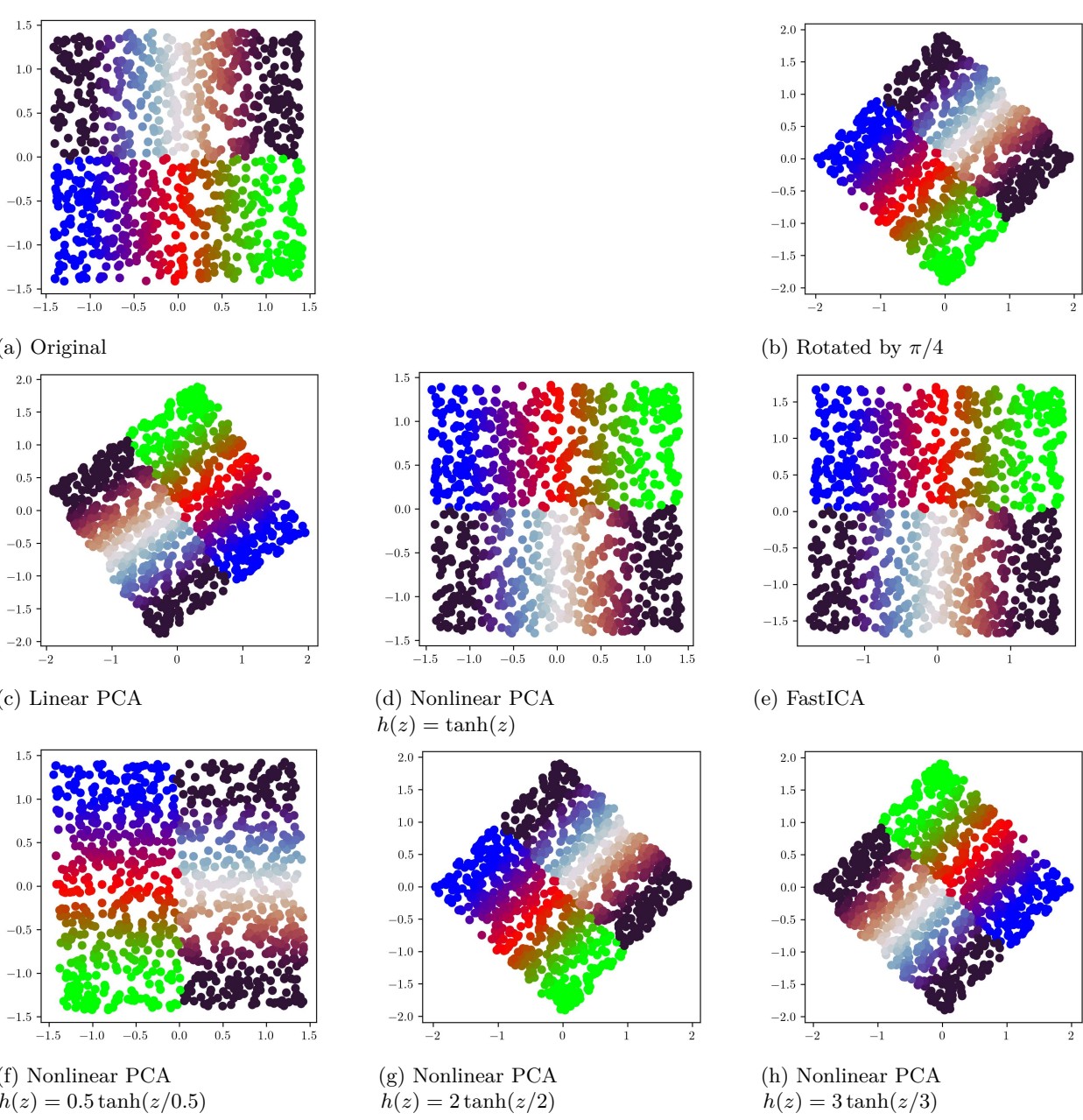

(a) Original

(b) Rotated by $\pi/4$

(c) Linear PCA

(d) Nonlinear PCA
$h(z) = \tanh(z)$

(e) FastICA

(f) Nonlinear PCA
$h(z) = 0.5\tanh(z/0.5)$

(g) Nonlinear PCA
$h(z) = 2\tanh(z/2)$

(h) Nonlinear PCA
$h(z) = 3\tanh(z/3)$

Figure 23: Uniform distribution (sub-Gaussian) with equal variance. The original data (a) was rotated by $\pi/4$ (b), then we attempted to recover the original data from the rotated data. We see that linear PCA (c) failed to recover the original data. Both nonlinear PCA (d) and FastICA (e) managed to recover the original data (up to sign and permutational indeterminacies, given the equal variance). We also see that $h(z) = a\tanh(z/a)$ with $a \leq 1$ worked best for the uniform distribution with nonlinear PCA.

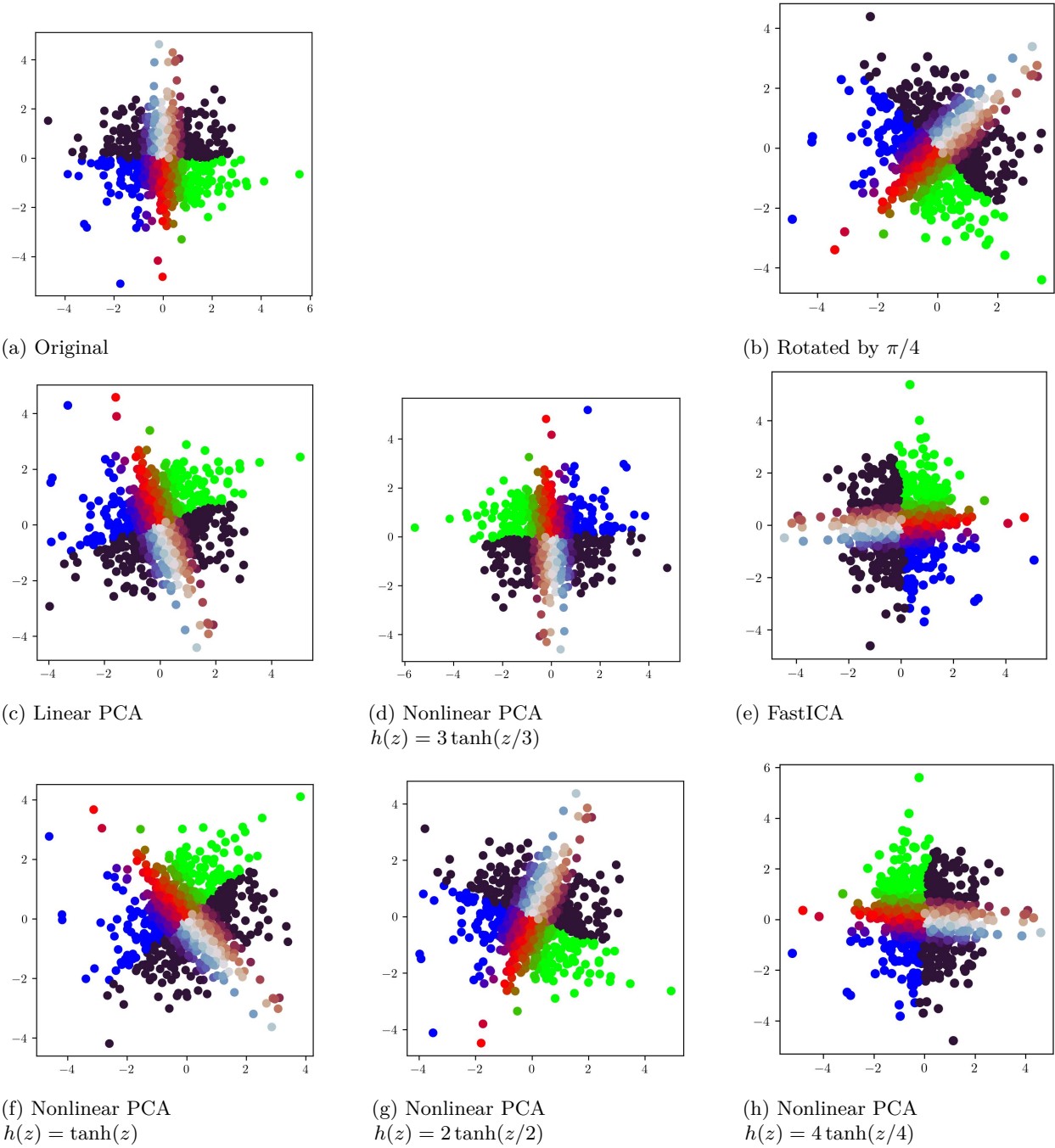

Figure 24: Laplace distribution (super-Gaussian) with equal variance. The original data (a) was rotated by $\pi/4$ (b), then we attempted to recover the original data from the rotated data. We see that linear PCA (c) failed to recover the original data. Both nonlinear PCA (d) and FastICA (e) managed to recover the original data (up to sign and permutational indeterminacies, given the equal variance). We also see that $h(z) = a \tanh(z/a)$ with $a \geq 3$ work best for the Laplace distribution with nonlinear PCA.

# H  Training details

**Patches**   We used the Adam optimizer (Kingma & Ba, 2014) with a learning rate either of 0.01 or 0.001, $\beta_1 = 0.9$ and $\beta_2 = 0.999$. We used a batch size of 128. For the patches, each epoch consisted of 4K iterations, and, unless stated otherwise, we trained for a total of three epochs. With the majority of the nonlinear PCA methods, we used a projective unit norm constraint (see Appendix C). The unit norm constraint with the Adam optimizer generally behaved well on unwhitened inputs, but not as well on whitened inputs. With whitened inputs, we found that it was better to either use the vanilla SGD optimizer with a unit-norm constraint, or the Adam optimizer with a unit-norm or orthogonality regularizer. This potentially might be due to the adaptive point-wise scaling that the Adam optimizer performs, but we did not investigate this further.

**Time signals**   We used the vanilla SGD optimizer with a learning rate of 0.01 and momentum 0.9. We used a batch size of 100 and trained for a total of 200 epochs and picked the weights with the lowest reconstruction loss. When using the Adam optimizer, we noted that it seemed better to also use differentiable weight normalization in addition to the projective unit norm constraint.

**2D points**   We used the vanilla SGD optimizer with a learning rate of 0.01 and momentum 0.9. We used a batch size of 100 and trained for a total of 100 epochs.

# I  Block rotation matrix RS = SR

$$\mathbf{SR} = \begin{pmatrix} s_1\mathbf{I}_{2\times2} & \mathbf{0} \\ \mathbf{0} & s_2 \end{pmatrix} \begin{pmatrix} \mathbf{R}_{2\times2} & \mathbf{0} \\ \mathbf{0} & 1 \end{pmatrix} = \begin{pmatrix} s_1\mathbf{R}_{2\times2} & \mathbf{0} \\ \mathbf{0} & s_2 \end{pmatrix} = \begin{pmatrix} \mathbf{R}_{2\times2} & \mathbf{0} \\ \mathbf{0} & 1 \end{pmatrix} \begin{pmatrix} s_1\mathbf{I}_{2\times2} & \mathbf{0} \\ \mathbf{0} & s_2 \end{pmatrix} = \mathbf{RS} \tag{I.1}$$

# J  Exponential moving average (EMA)

$$\bar{\boldsymbol{\mu}} = \frac{1}{b}\sum \mathbf{y}_i \tag{J.1}$$

$$\bar{\boldsymbol{\sigma}}^2 = \frac{1}{b}\sum (\mathbf{y}_i - \bar{\boldsymbol{\mu}})^2 \tag{J.2}$$

$$\hat{\boldsymbol{\mu}} = \alpha\hat{\boldsymbol{\mu}} + (1-\alpha)[\bar{\boldsymbol{\mu}}]_{sg} \tag{J.3}$$

$$\hat{\boldsymbol{\sigma}}^2 = \alpha\hat{\boldsymbol{\sigma}}^2 + (1-\alpha)[\bar{\boldsymbol{\sigma}}^2]_{sg} \tag{J.4}$$

$$\text{Batch} \quad n_{\mathrm{B}}(\mathbf{y}) = \frac{\mathbf{y} - \bar{\boldsymbol{\mu}}}{\sqrt{\bar{\boldsymbol{\sigma}}^2 + \epsilon}} \tag{J.5}$$

$$\text{EMA} \quad n_{\mathrm{E}}(\mathbf{y}) = \frac{\mathbf{y} - \hat{\boldsymbol{\mu}}}{\sqrt{\hat{\boldsymbol{\sigma}}^2 + \epsilon}} \tag{J.6}$$

where $[\ \ ]_{sg}$ is the stop gradient operator, $\alpha$ is the momentum, and $b$ is the batch size.

