# OpenReview forum: "$\sigma$-PCA: a building block for neural learning of identifiable linear transformations"
_TMLR — Accepted by TMLR_

### Review · Reviewer_3yA9 · 2023-12-01

**Summary Of Contributions:**

This paper presents a simple modification of the PCA loss function and optimization algorithm which solves for both the objectives of PCA and non-linear PCA, solving for both variance reduction and statistical independent maximization. It provides extensive analysis of the resulting algorithm on a number of synthetic tasks, and code to reproduce it.

**Audience:**

Yes

**Broader Impact Concerns:**

No concerns.

**Claims And Evidence:**

Yes

**Requested Changes:**

None mandated. Consider whether a real-world validation on a downstream task can be fit within the scope of this (already lengthy) paper.

**Strengths And Weaknesses:**

Strengths:
- The paper is extremely well written and limpid.
- It is also one of the clearest expositions of PCA, non-linear PCA and ICA I've come across.
- Reproducibility is high thanks to the notebooks, which I hope will be posted online.
- The contribution is simple, consequential and well-motivated.

Weaknesses:
- The paper takes for granted that the goal of learning a unique rotation in equal variance subspaces of PCA is actually worthwhile. It's not intuitive to the naive reader why this is desirable and the connections to identifyability and statistical independence only come across later as if they were self-evident.
- The paper proposes an algorithm that is essentially mid-way between PCA -- a robust, well-established technique, useful and widely used in real-world contexts, and ICA -- the poster child for methods that look great theoretically, give great results on synthetic data, but largely fail at any real world applications due to their sensitivity to even mild violations of their underlying assumptions. This begs the question of which category \sigma-PCA falls into. It would be really strengthening the paper if it provided evidence of the method improving performance on a real-world downstream task, not artificially constructed and not merely producing prettier filters.

Unknowns:
- I have not looked at the PCA literature in 10 years+ and would not be in a position to evaluate novelty.

---

> ### Author Response · Authors · 2024-02-12
>
> Thank you for having taken the time to read our paper, and we appreciate that you've found the exposition on the topic to be one of the clearest. The notebooks will be posted online.
>
> W1: We will be updating the paper in the next few days to flesh out and explicitly state the connections with identifiability.
>
> W2: As you've rightly noted, the contribution in itself is simple. Nonlinear PCA, in our formulation, just goes a step further than linear PCA in that it eliminates the rotational indeterminacy. In terms of potential applications, it is restricted to any application where one might still resort to using linear PCA.
> The more interesting application of this work, we think, is the extension of the method to deeper networks for learning nonlinear transformations.  We initially started treating both linear and nonlinear transformations in the same paper, but it became even more lengthier than it currently is, and so we decided that it would be best to separate them both, for, after all, they are different topics. In our follow up paper, which we aim to finish in the next 2-3 months, we develop the extension to nonlinear transformations and apply it to more complicated datasets, while emphasising the connections to learning disentangled/identifiable representations.

---

### Review · Reviewer_SXR6 · 2023-12-27

**Summary Of Contributions:**

This paper introduces σ-PCA, a neural model that unifies linear and nonlinear principal component analysis (PCA) within the framework of single-layer autoencoders. The model is notable for addressing both rotation and scale, emphasizing variances. It provides a comprehensive view, bridging the gap between linear and nonlinear PCA methodologies.

**Audience:**

Yes

**Claims And Evidence:**

Yes

**Requested Changes:**

- It is suggested to add discussions about the connections of the proposed models with modern deep learning models.
- It is suggested to analyze the scalability of the proposed algorithm.
- It is interesting to explore the robustness especially adversarial robustness of the proposed $\sigma$-PCA.

**Strengths And Weaknesses:**

### Strengths

- *A new model*: This paper introduces σ-PCA, a novel neural model that integrates linear and nonlinear PCA through single-layer autoencoders.  A distinctive aspect of σ-PCA is its ability to handle both rotation and scale, focusing on variances. This approach effectively narrows the gap between linear and nonlinear PCA.
- *Good properties*: σ-PCA can execute a semi-orthogonal transformation akin to linear PCA, facilitating dimensionality reduction and variance-based ordering. However, it surpasses linear PCA by avoiding the issue of rotational indeterminacy, marking a significant advancement in PCA methodology.

### Weaknesses
- Connections with modern deep learning progress: I appreciate the new model. However, it is unclear whether the proposed model can be utilized in designing or analyzing modern deep learning models like Transformers and CNNs.
- Comprehensive analysis of the algorithm: The detailed computational cost of the proposed algorithm is not provided, and thus the scaling ability of the proposed algorithm is unclear.

---

> ### Author Response · Authors · 2024-02-12
>
> We appreciate you taking the time to read our paper and you finding that it presents an advancement in PCA methodology.
>
> W1: We decided that it would be appropriate to limit the scope of this paper to learning linear transformations only, something we have tried to highlight in the introduction, but will try to further emphasise it in our amends in the next few day.
> We will have a lengthy follow up paper, coming out in the next 2-3 months, dealing with nonlinear transformations and so making lots of connections with deep learning.
>
> W2: In terms of scalability, the method is certainly more scalable than any of the modern deep learning methods, for, after all, it is just a single-layer autoencoder, which can be trained efficiently with gradient descent. It can be implemented trivially with any deep learning framework. In the supplementary material, we've included an implementation in tensorflow.
> It would certainly be interesting to explore adversarial robustness, but we suspect it would be the same as with linear PCA and linear ICA.  We have not looked whether the adversarial robustness of such linear methods was explored in the literature, but it would certainly be a good avenue for future research.

---

### Review · Reviewer_5HSt · 2024-02-08

**Summary Of Contributions:**

This paper identifies the reason why conventional nonlinear PCA cannot recover the first rotation in the transformation (which is crucial for dimensionality reduction), and accordingly develops a solution by incorporating a (learnable) diagonal scaling term in the objective function of conventional nonlinear PCA. This results in an "end-to-end" nonlinear PCA procedure, without the need of first whitening (i.e., applying linear PCA rotation and standardizing) the data. The paper further discusses about the connection between linear PCA, nonlinear PCA, and linear ICA, and provide empirical studies to validate the proposed method.

**Audience:**

Yes

**Claims And Evidence:**

Yes

**Requested Changes:**

- As discussed above, what is the benefit of the proposed "end-to-end" nonlinear PCA method, as compared to the existing two-step procedure (whitening + existing nonlinear ICA method)? Does the former have better computational complexity, run time, empirical performance (or sample complexity), or even interpretability?
- The paper (e.g. conclusion) states that the method has "added benefit of identifiability". Although it might be clear to the authors, it would be great to give an explicit statement (e.g. proposition/theorem) to formulate it. E.g. what are the conditions needed for Eq. 23 to achieve identifiability and what is the indeterminacy involved.
- For Eq. 23, it may be worth discussing whether the global minimum can be reached with the proposed procedure.

Some minor points:
- It seems that most of the indeterminacies in Sec. 2.1 also hold in factor analysis, which is known to share similarities with PCA and ICA. It is worth making this connection to enrich the paper.
- Most of the PCA methods discussed in the introduction and related works are quite old. The paper could benefit from also briefly discussing more recent nonlinear PCA methods (such as https://arxiv.org/abs/1402.0119 and https://www.sciencedirect.com/science/article/pii/S089360802300669X).
- In Sec. 2.4, introduction, or related works, it is stated that "The only case where. . all distinct" or "solely based on ... non-Gaussian". This could be rephrased because Gaussian components may still be identified in some cases (e.g. https://arxiv.org/abs/1804.00408, https://openreview.net/forum?id=kJIibP5bq2, https://ieeexplore.ieee.org/document/374166), and may be relevant to the identifiability of the proposed method.
- The motivation of Sec. 3.1 does not seem very clear. It would be great to give a high-level overview in the beginning of Sec. 3.1 to explain the motivation.

**Strengths And Weaknesses:**

Strengths
- The paper is well written. The introduction and background sections provide a clear discussion of the related area. The connection between linear and nonlinear PCA, as well as linear ICA, is well discussed and insightful.
- The proposed method is a simple and effective improvement over existing nonlinear PCA method.

Weaknesses
- While I understand that the existing nonlinear PCA method relies on a whitening step, and that the proposed method is an "end-to-end" (or single-step) procedure, the benefit of the latter is not well discussed and does not seem clear. Specifically, does such an end-to-end procedure have a better computational complexity (or run time), or does it lead to better empirical performance (or sample complexity)?
- It would be great to give a proposition/theorem that explicitly formulates the identifiability conditions and results for the method (see below for more details).

---

> ### Author Response · Authors · 2024-02-12
>
> Thank you for taking the time to read our paper, and we appreciate you finding the connections insightful.
>
> With whitening, the overall transformation with conventional nonlinear PCA ends up being $W = (U\Sigma^{-1})V$. Even though $W$ and $V$ are orthogonal, because of the scale, the overall transformation $W$ is not necessarily orthogonal. As part of linear ICA, conventional nonlinear PCA could only be used to learn $V$, and linear PCA could only be used to learn $U$. The problem, then, is that it was not possible to learn an orthogonal transformation directly that maximises independence. And so conventional nonlinear PCA could only be used after linear PCA was applied as a preprocessing step (whitening). With our modification, nonlinear PCA can learn an orthogonal transformation directly that maximises independence, so nonlinear  PCA becomes on par with linear PCA rather than being a method that is only dependent on it, except that it can go a step further than linear PCA and eliminate the rotational indeterminacy when variances are similar.
>
> If the task is to learn an orthogonal transformation that maximises independence, our formulation of nonlinear PCA would fit an orthogonal matrix directly as $W = (U\Sigma^{-1})$, whereas linear ICA would fit $W' = (U'\Sigma^{-1})V$, without a guarantee that $W'$ will be orthogonal (unless the linear ICA algorithm is modified with additional constraints to enforce the orthogonality of the overall transformation). Although the difference can be said to be minor, for this specific task, nonlinear PCA is parameterised just right, whereas linear ICA is over-parameterised.
>
> One unfortunate thing about the term "nonlinear PCA" is that it has been used to refer to a wide variety of methods covering both linear and nonlinear transformations, and so the term nonlinear has been used to refer to either function or transformation or both. This is in contrast to the ICA literature, where the terms linear and nonlinear refer only to the transformation, despite linear ICA using a nonlinear function. In our work, in the context of PCA, nonlinear refers to the function, as we are simply building on prior, albeit old, work where the term was used in such a manner.
> In the two nonlinear PCA papers you reference, nonlinear refers to the transformation, and so they are outside the scope of this paper, which is solely linear transformations. As we have remarked in our response to the other reviewers, there will be a follow up paper in the next few months dealing with the extension of the method to nonlinear transformations.
>
> In the next few days we will do the following:
>
> - formulate explicitly the identifiability conditions;
> - add in the appendix the connection with factor analysis;
> - comment on whether global minimum can be reached;
> - update the motivation for sec 3.1;
> - rephrase our sentence on identifiability of Gaussian components.

---

> > ### Author Response · Authors · 2024-02-22
> >
> > We have updated our paper, and in particular we have
> >
> > - added a statement to explicitly define when a transformation is identifiable;
> > - added an additional clarification on rotational indeterminacies;
> > - added in the appendix a section on factor analysis and probabilistic PCA;
> > - added an additional section on the link with linear ICA as well as notes on the convergence of nonlinear PCA;
> > - rephrased the introductory sentence of sec 3.1 to better show its aim;
> > - rephrased our sentence on the identifiability of Gaussian components by restricting it to criterions based on variance or independence of components in the data distribution.

---

### Decision · Action_Editor_LCNi · 2024-05-23

**Recommendation:** Accept with minor revision

**Comment:**

The paper well studies relationship between, linear PCA, nonlinear PCA, and linear ICA, presenting an interesting nonlinear PCA formulation. Analysis on this relation, in the perspective of SVD of linear transformation is nice. However, the emphasis on trying to relate PCA, nonlinear PCA, and ICA, and developing a method that can simultaneously find the maximal variance subspace while also learning independent components, requires a lot to take in, since each of the mentioned problems has different objectives and underlying assumptions.
BSS is  the common problem formulation that ties them together? Make it clear what is a common problem. If BSS is a common problem, then  clearly explains what is the benefit of the proposed formulation in solve BSS.
Since the paper has interesting studies, so I do not want to kill this work.

**Audience:**

PCA and nonlinear PCA are relatively old topics, but the authors did a good job in bringing up an interesting formulation involving the scaling diagonal matrix. Relationship between linear PCA, nonlinear PCA, and linear ICA is well summarized in Section 3.2, which may be interesting for TMLR audience.

**Claims And Evidence:**

This paper claims that it presents a unified neural model for linear and nonlinear PCA, referred to as "\sigma-PCA". However, it is NOT clear what the goal of this paper is. This paper has been re-assigned to me since both authors and previous AE requested for re-assignment, after the review was complete. The previous AE also pointed this out, saying "There is emphasis on trying to relate PCA, nonlinear PCA, and ICA, and developing a method that can simultaneously find the maximal variance subspace while also learning independent components. That is a lot to take in. The reason is that each of the mentioned problems has different objectives and underlying assumptions. What is a common problem formulation that ties them together? Is it a blind source separation problem? If so, what are the underlying assumptions? Similarly, why should the same algorithm work for these seemingly different tasks that are only solvable under different sets of assumptions? The very premise of the paper is odd to me." In fact, I also agree with this. I am sure that the paper contains interesting contributions, but it would be nice to make it clear what the goal of this paper is. It seems that the new nonlinear PCA formulation has been applied to the blind source separation problems. The experiments were done on a few rather toy blind source separation problems. Many different methods and algorithms for ICA or BSS have been developed for last two decades. The nonlinear PCA is one heuristic approach to BSS. Thus, why such a new nonlinear PCA formulation is important to solve BSS? All reviewers agree that the paper has interesting contributions on nonlinear PCA but they have little confidence. I would like to suggest the authors to narrow down the scope of this paper, in order to better present the main contribution.

---

> ### Author Response · Authors · 2024-06-17
> **We really appreciate your time and effort for taking over as action editor for our submission.**
>
> We had framed our paper from the point of view of a unified neural model of linear and nonlinear PCA because, in a sense, it seemed appropriate to give credit to all the prior works of neural PCA that led to the conception of conventional nonlinear PCA, a linear ICA method that could only work on whitened inputs. But the majority of those works are, at this point, three decades old, and are, perhaps, not fresh in the minds of the current research community, and so, in hindsight, we understand that it might be hard to see the relevance of presenting a unified model when it is not exactly clear from the title what nonlinear PCA is.
>
> The main goal of this paper is to solve the following problem: eliminating the subspace rotational indeterminacy from the canonical linear PCA solution -- this, indeed, is a blind source separation problem.
>
> The subspace rotational indeterminacy exists because linear PCA cannot separate components when they have the same variance.
> For separating components, when variance cannot be used as a criterion, then the alternative is use independence. Maximising independence is at the heart of blind source separation problems. Blind source separation/linear ICA methods tend to require whitening, which in itself is not exactly an issue, but it solves a subtly different problem -- something that we have tried to emphasise in our paper.
>
> As part of our paper, we have aimed to show that a simple modification to the conventional nonlinear PCA algorithm which involves taking the variances into account, and placing an emphasis on using the encoder contribution part of the gradient, leads to an algorithm that can eliminate the subspace rotational indeterminacy from the canonical linear PCA solution  -- without whitening the inputs. We have then shown that by simply looking at the weight update, there are two clear terms that appear: one that maximises variance, and another that maximises independence.  It, therefore, naturally combines aspects of linear PCA and linear ICA. With our modification, nonlinear PCA becomes on par with linear PCA in that it can learn a semi-orthogonal transformation to reduce dimensionality, whereas previously it was restricted to learning orthogonal transformation on whitened input -- this motivated the statement of a unified model.
>
>
> Linear PCA, nonlinear PCA, and linear ICA are all methods for learning linear transformations, and what ties them all together are the concepts of orthogonality, Gaussianity, variance, and independence. Using those concepts, we have attempted to explain at length how they all tie together when considered from the point of view of the SVD of the linear ICA transformation.
>
>
> The relevance of all this work may appear, at this point, moot, but we believe that it provides a better conceptual understanding as to what a tied single-layer autoencoder with a nonlinear function is doing, and, in particular, how autoencoders can be used to perform PCA and ICA. Although, owning to the length of the paper, we have delegated to the appendix the fact that single-layer spare autoencoders appear to perform nonlinear PCA, we believe that it might help conceptually explain what exactly such sparse autoencoders are doing given the recent trend of using them to find interpretable features from language models, e.g. [2].
>
> In light of the suggestions we have made the following changes:
>
> - We have rewritten the abstract to emphasise the problem we are solving, namely that of eliminating the subspace rotational indeterminacy from the canonical linear PCA solution, i.e. making the linear PCA solution identifiable.
> - We have also made a few minor changes in the paper to emphasise the above point as well as adding additional clarification on why nonlinear PCA, as part of $\sigma$-PCA, maximises both variance and independence, and how it reduces to existing linear ICA on unit-variance inputs [1].
> - We have changed the title of the paper to:  $\sigma$-PCA: a building block for neural learning of identifiable linear transformations
>
> The justification for the title change is the following: (1) It emphasises that the scope is linear transformations, and that it specifically deals with transformations that are identifiable;
> (2) it can be said that it is a building block, because, as part of our formulation, nonlinear PCA can directly learn an identifiable semi-orthogonal transformation, which is simply the canonical linear PCA solution without the subspace rotational indeterminacy, and it can be used to perform linear ICA, when applied twice, to learn arbitrary unit-variance linear transformations.
>
> [1] Hyvärinen, Aapo, and Erkki Oja. "Independent component analysis by general nonlinear Hebbian-like learning rules." signal processing 64.3 (1998): 301-313.
>
> [2] Cunningham, Hoagy, et al. "Sparse autoencoders find highly interpretable features in language models." arXiv preprint arXiv:2309.08600 (2023).